# Multi-criteria group shilling attacks

**Tugba Turkoglu Kaya**◯¤*

Computer Engineering Department, Ardahan University, Camlicatak, Ardahan, Turkey

¤ Current address: Department of Computer Engineering, Ardahan University, Ardahan, Turkey
* tugbaturkoglu@ardahan.edu.tr

## Abstract

The rapid advancement of technology has enabled the collection of detailed, multi-dimensional user data, paving the way for multi-criteria recommendation systems that consider diverse aspects of user preferences. While traditional recommendation systems aim to satisfy individual users, group recommendation systems are designed to generate suggestions that accommodate the collective preferences of a group. However, the increasing prevalence of group interactions in digital environments has also introduced new vulnerabilities, such as group shilling attacks, where coordinated malicious users manipulate recommendation outcomes. This study conducts the first comprehensive robustness analysis of multi-criteria group recommender systems, addressing a critical research gap. A novel shilling attack strategy is proposed by adapting the group shilling model to multi-criteria settings, allowing a deeper understanding of the risks these systems face. Experimental results indicate that the proposed multi-criteria recommender system achieves notable robustness across datasets. Specifically, the average hit ratio (AvgHR) increases up to approximately 12% on the YM20 dataset and reaches around 15% on the YM10 dataset. Furthermore, among the target item selection strategies, the MUP-NNZ method consistently demonstrates superior resistance to profile injection attacks, confirming its effectiveness in maintaining recommendation accuracy under adversarial conditions.

## 1 Introduction

The rapid advancement of science and technology in recent years has enabled the collection of large-scale and detailed user data through online platforms. User comments, ratings, and feedback now form extensive datasets that reflect individual preferences and behaviors. Recommender systems analyze these heterogeneous data sources to help users make informed decisions efficiently, enhancing user satisfaction by reducing the time spent on product selection. To further improve recommendation accuracy, many modern systems allow users to provide more detailed, multi-dimensional feedback instead of a single overall score. For instance, major

**Data availability statement:** All MATLAB source code used to generate the results in this manuscript, together with the processed datasets, have been deposited in the public GitHub repository: https://github.com/tugba7203/mcrs-shilling-attack-codes.

**Funding:** This study was supported by Ardahan University Scientific Research Project Commission, Turkey under the grant no: 2025-002. The funders had no role in study design, data collection and analysis, decision to publish, or preparation of the manuscript. No authors received a salary from the funders.

**Competing interests:** The authors declare that they have no known competing financial interests or personal relationships that could have appeared to influence the work reported in this paper.

hotel booking platforms enable users to rate criteria such as cleanliness, breakfast, and staff friendliness; Amazon.com offers rating dimensions for video games; and eBay allows users to evaluate sellers based on multiple aspects. Similarly, Yahoo!Movies lets users rate movies across sub-criteria such as acting, direction, story, and visuals. These systems, known as *multi-criteria recommender systems (MCRS)*, enable a more comprehensive understanding of user preferences and contribute to higher recommendation accuracy [1].

Traditional recommender systems—whether single- or multi-criteria—provide personalized recommendations for items users may like but have not yet experienced [2]. However, many real-world activities are shared experiences, such as dining with colleagues, watching movies with family, or exercising with friends. In these contexts, the focus shifts from *individual* to *collective preferences*. To address this need, *group recommender systems (GRS)* have been developed as an advanced extension of traditional recommenders. GRS aim to generate recommendations that satisfy the collective interests of multiple users by aggregating their individual evaluations—whether single- or multi-criteria—thereby improving satisfaction and decision-making efficiency in group settings.

Companies utilizing group recommendation systems, as well as other recommendation system methods (including multi-criteria recommendation systems), must ensure that they provide high-quality recommendations to avoid disappointing their customers. Recommending inappropriate or irrelevant products may lead users to switch to competing e-commerce platforms, resulting in customer loss and a decline in the company's competitive standing. Additionally, malicious individuals or businesses can exploit these systems to manipulate recommendations for their own benefit. By targeting specific products, they may attempt to artificially increase or decrease their popularity. To achieve this, they can introduce fake user profiles into the system. Such activities are commonly referred to as "shilling attacks" or "profile injection attacks" [3].

Shilling attacks in recommender systems are typically classified into two categories: *push* attacks, which aim to artificially promote the popularity of targeted items, and *nuke* attacks, which attempt to diminish their visibility or perceived value. While traditional shilling attacks are often executed through the independent injection of fake user profiles, a more sophisticated and dangerous variant emerges when multiple malicious actors operate in a coordinated and covert manner. This strategy, widely known as a *group shilling attack*, poses a substantially greater threat to the integrity and trustworthiness of recommender systems compared to conventional individual-based attacks [4].

Despite the significance of this threat, the existing body of research reveals a critical gap: the vast majority of prior studies on shilling attacks have focused on single-criteria recommender systems. Even within the domain of multi-criteria recommenders, recent works [5,6] have concentrated primarily on individual-level attack detection rather than on group-level robustness analysis. To the best of our knowledge, no prior research has systematically investigated group shilling attacks within multi-criteria group recommender systems (MC-GRS), where recommendations are based on multiple user preference dimensions and collective decision-making

mechanisms. This absence represents a serious limitation in the literature, given that MC-GRS are increasingly deployed in real-world digital environments and are inherently more vulnerable to coordinated adversarial manipulation.

Motivated by this gap, the present study undertakes the first comprehensive robustness analysis of multi-criteria group recommender systems under group shilling attacks. To this end, we propose a novel attack strategy specifically tailored to multi-criteria group settings, exposing critical vulnerabilities and quantifying the extent to which such systems can be manipulated. Furthermore, we develop a systematic evaluation framework to rigorously assess the impact of these attacks, thereby offering actionable insights for designing more resilient, accurate, and trustworthy recommender systems. By bridging this crucial gap, our work not only advances the theoretical understanding of adversarial threats in recommender systems but also contributes practical guidelines for safeguarding next-generation recommendation technologies against coordinated manipulation.

The contributions of the study are listed below;

- This study provides the first comprehensive robustness analysis of multi-criteria group recommender systems, thereby addressing a significant and previously unexplored gap in the existing body of literature. The analysis offers a holistic perspective on the reliability and security of such systems under adversarial conditions.
- The research makes a substantial contribution to the understanding of group shilling attacks, particularly within the context of multi-criteria recommendation settings, which have been largely neglected in prior studies. It delivers in-depth insights into the ways in which group interactions may be exploited and manipulated, thus filling an important void in current knowledge.
- A novel shilling attack strategy is introduced by specifically adapting the traditional group shilling model to multi-criteria group recommender systems. This adaptation reveals critical system vulnerabilities and, at the same time, provides valuable guidance for the development of more resilient recommendation frameworks and defensive mechanisms.

This study consists of 8 sections. While the literature review related to the GRS is included in Sect 2, information about the methods used throughout the study is given in Sect 3. While the proposed method is presented in Sect 4, data set, testing methodology and performance metrics are introduced in Sect 5. In Sect 6 the implementation and results of the proposed methods and approaches are given, in the last section, the general conclusions and suggestions from the study are presented.

## 2 Related work

With the rapid development of the Internet and the emergence of e-commerce sites, recommendation systems create personalized product lists using user-product feedback and harmonize similarities between users. In addition, individuals by nature tend to act together by interacting with people they know for many activities. In this mechanism, called group recommendation systems, the target audience is a community of users who come together for various reasons and have to act together, rather than individual users. However, recommendation systems and group recommendation systems are vulnerable to attacks by malicious users, such as the insertion of fake profiles, so-called Shilling Attacks. The vulnerability of recommender systems to shilling attacks has led to many recent studies focusing on the concept of trust in the proposal presented from different perspectives. Riedl and Lam [3], who first gives the term shilling, suggested two basic attack models, Random and Average attack models. Although random attack (called RandomBot [3,7] can be applied easily, it is not a very effective model [3,7], while average attack (called AverageBot [3,8] is difficult to implement because it is high knowledge. The other attack studies are [7–15].

In traditional shilling attacks, malicious users work individually to add fake profiles in order to manipulate recommendation (including group recommendation systems) systems into recommending or not recommending certain products. In fact, a group of attackers can work together to affect the services of recommendation systems by adding fake profiles to

the system. These systems, referred to as group shilling attacks, are more harmful than traditional shilling attacks, and it seems that much work has been done on the detection of group shilling attacks rather than their impact.

This concept was first mentioned by Su et al. [4]. Researchers have noted the impact of aggressive user groups with the same shilling behavior. However, in the proposed study, Wang et al. [16], who demonstrated that high diversity should be taken into account when creating shilling groups, expanded the group shilling attack creation algorithm. The detection of proposed group shilling attacks was carried out by Wang et al. [17].The researcher present a shilling group detection method based on graph convolutional network. Another study, [18] propose a graph embedding-based method to detect group shilling attacks in collaborative recommender systems. Other studies carried out on group shilling attack and detection are [19–23].

In the context of multi-criteria recommendation, Kaya and Kaleli [6] developed a novel top-*n* recommendation framework based on a new neighborhood selection process (NSP) utilizing entropy and association rule mining (ARM), effectively improving accuracy through the analysis of user and item characteristics. Turkoglu Kaya et al. [5] further contributed by proposing a classification-based shilling attack detection model specifically designed for multi-criteria recommender systems, focusing on identifying fake profiles through user–item interaction patterns and popularity-driven features. These studies significantly improved robustness and personalization in multi-criteria settings but primarily addressed individual-level recommendation scenarios.

Meanwhile, robustness in deep learning–based recommendation has also attracted significant attention. Wang et al. [24] proposed a *Distributionally Robust Graph-based Recommender System (DR-GNN)*, incorporating Distributional Robust Optimization (DRO) to mitigate performance degradation under distributional shifts, ensuring stability in graph neural network (GNN) recommenders. Similarly, Boratto et al. [25] examined robustness and fairness trade-offs in GNN-based recommenders under edge-level perturbations, revealing how fairness between providers and consumers is affected by adversarial noise. These works highlight the growing importance of robustness under structural and fairness-oriented perturbations.

To bridge these robustness-oriented advances with existing attack and detection research, Table 1 summarizes key studies addressing shilling and group shilling attacks in recommender systems.

Despite these advancements, none of the existing studies have systematically investigated robustness in *multi-criteria group recommender systems (MC-GRS)* under coordinated group shilling attacks. Prior research has primarily focused on either robustness in single-user multi-criteria recommenders or perturbation-resilient graph-based systems. In contrast, the present study provides the first comprehensive robustness analysis of MC-GRS under adversarial manipulation, introducing a novel group-oriented attack model and an evaluation framework to quantify its impact. This approach bridges the gap between multi-criteria and group-level robustness, contributing new insights into the design of resilient recommendation architectures.

When we look at the literature review, it is seen that studies on group recommendation systems are not included, and moreover, the focus is more on the detection of group shilling attacks rather than the impact of these attacks. For this purpose, the robustness of group recommendation systems and the impact of these group shilling attacks on multi-criteria systems will be examined in order to fill the gap in the literature.

## 3 Preliminaries

In the section, necessary information about the methods used in the study is given.

### 3.1 Group recommender systems

Group recommendation systems (GRS) are mechanisms that analyze the characteristics and tendencies of a user community that acts together by sharing the same environment, rather than individual recommendations, and produces group recommendations that will satisfy them at the maximum level [30]. In these systems, the first step in generating group

**Table 1**. Comparative summary of existing studies on shilling and group shilling attack detection in recommender systems.

| Ref. | Attack Type | Main Findings / Results | Advantages | Strengths |
|------|-------------|-------------------------|------------|-----------|
| [19] | Group Shilling | Effective detection of group attacks on Netflix and Amazon datasets using bisecting K-means clustering. | Segments suspicious candidate groups based on behavioral similarity; improves detection precision. | Demonstrates solid clustering performance and scalability on real-world datasets. |
| [18] | Group Shilling | Proposed a graph embedding + K-means++ based framework achieving high detection accuracy. | Captures latent user relations via graph topology; identifies collusive behaviors. | Integrates graph representation learning for deeper structural understanding of user interactions. |
| [16] | Group Shilling (Attack Model) | Introduced a generative and diverse attack model on MovieLens dataset that can evade classical detection. | Generates realistic and diverse fake profiles; enhances realism of evaluations. | Provides a benchmark model for testing robustness of defense mechanisms. |
| [4] | Group Shilling | Proposed a similarity-spreading algorithm to locate collusive users in e-commerce systems. | First conceptual model for group shilling; foundational for later detection methods. | Established the initial framework for group-level detection in recommender systems. |
| [26] | Shilling (Multi-target, GAN-based) | Proposed a multi-target black-box attack framework with $> 80\%$ success rate using GAN profiles. | Produces realistic profiles; supports multi-target attacks. | Introduces GAN-based adversarial modeling for complex, multi-target attacks. |
| [27] | Shilling + Adversarial | Unified diffusion-based purification and adaptive training; outperformed SOTA on three datasets. | Handles both known and unknown adversarial perturbations; robust across data types. | Combines adversarial defense and self-adaptive learning into a unified framework. |
| [28] | Group Shilling | Proposed a credibility and time-series-based detection model; strong benchmark performance. | Integrates temporal patterns and user credibility; detects coordinated attacks. | Effectively models dynamic behavioral patterns and rating credibility. |
| [29] | Shilling (LLM-based) | Introduced SemanticShield, an LLM-driven auditing framework with semantic and behavioral layers. | Exploits semantic item features and LLM reasoning; strong generalization ability. | Leverages large language models for semantic-level reasoning and explainable detection. |

suggestions is to identify user communities containing similar individuals. In group creation mechanisms where 4 different approaches were determined, the increase in the number of users in the system over time and the constant change in user preferences showed that the automatic group definition method is more successful and appropriate in these scenarios [31,32]. The reason why automatically defined group recommendation systems are successful is that they include users who have similar tastes and are in harmony with each other, rather than individuals who come together randomly, and it can be easier to please the suggestions offered to these groups of people [33]. The most common method used to identify similar and compatible user groups is to divide users into groups using a clustering algorithm without any restrictions [34]. In clustering algorithms based on users' evaluations of products, the k-means method is frequently used in automatic group identification due to its efficiency and applicability.

After creating user groups, products or services that will satisfy the individuals in the group to the maximum extent must be offered. For this purpose, it is generally produced by taking into account group profiles, which are created by combining the rating values of group members for products/services using methods called aggregation techniques and reflecting the group's preferences in relevant products/services [34]. Combining techniques, which have a very important role in the production of group suggestions, directly affect the quality of group suggestions [35]. Many different combining techniques have been developed in the literature in line with the need, and Average (Avg) [36–39], Average without Misery (AwM) [40], Multiplicative (Mul) [41], and Additive Utilitarian (AU) [34,42], Approval Voting (AV) based on rating frequency [34,43], Most satisfaction or least satisfaction with the highest and lowest ratings (Most Pleasure-MP; Least Misery-LM ) [34,41,44–46] and Most Respected Person (MRP) [47] are techniques that take into account influential group members.

## 3.2 Group shilling attack

The concept of group shilling attacks in recommendation systems was proposed by Su et al. [4] and two scenarios for such attacks were presented. Based on this concept, Wang et al. [16] presented a new group shilling attack model to increase the diversity within the group and keep the similarity relationships between each pair of attackers to a minimum. The method they propose makes attack detection difficult as well as successful attacks.

In the developed attack model, two new versions have been proposed, denoted as $GSAGen_s$ (strict version) and $GSAGen_l$ (loose version), respectively. While the attackers in the shilling group created with the $GSAGen_s$ model do not have common rated products other than the target product, in the $GSAGen_l$ model, the attackers in the shilling group have a common rated target item and each pair of attackers have rated at most two filler items in common, and these filler items are rated by at most two attackers. According to the description of both models, it can be said that $GSAGen_s$ have stricter conditions. $GSAGen_s$ and $GSAGen_l$ can be divided into two types: $GSAGen_sRan$ and $GSAGen_sAvg$ ($GSAGen_lRan$ and $GSAGen_lAvg$), which are formed by the random attack model and the average attack model, respectively. Fig 1 is shown briefly the group attack models [19].

In Fig 1, $GSAGen_s$ has more string conditions in creating group attack profiles, so the size of attack groups created by the group attack model is limited. Therefore, we will use the loose version ($GSAGen_l$) to create group attack profiles in the article to ensure the effect of group shilling attacks.

## 4 Multi-criteria group shilling attack

In the attack on multi-criteria group recommendation systems, which contain versatile user data, the attacker's aim is to manipulate the products in the group recommendation list and ensure that fake products are included in the list instead of recommendations that will please the group. Therefore, in the group attack methodology to be created for this purpose, we assume that the attacker wants to inject shill attack groups into the system and that fake users in these groups rate target products that real users do not like or are unpopular. Thus, the attack intent is push, as the aim is to include the wrong products in the list. However, one of the important points here is how to distribute the fake profiles homogeneously to each group and make a successful attack for each group in the attack on the group suggestion system. We mentioned in Sect 3.2 that the group shillings designed to be compatible with this purpose are $GSAGen_s$ and $GSAGen_l$. It is seen in literature reviews that the $GSAGen_l$ attack model is more suitable for studies since the $GSAGen_s$ attack model has very

| Version | Type | $I_S$ | | $I_F$ | | $I_T$ |
|---|---|---|---|---|---|---|
| | | Items | Rating | Items | Rating | Rating |
| $GSAGen_s$ | $GSAGen_s$ Ran | | Not used | Each item in this set is randomly chosen and only rated by one attacker | system's mean | $r_{max}/r_{min}$ |
| | $GSAGen_s$ Avg | | Not used | Each item in this set is randomly chosen and only rated by one attacker | each item's mean | $r_{max}/r_{min}$ |
| $GSAGen_l$ | $GSAGen_l$ Ran | | Not used | Each item in this set is randomly chosen and rated by two attackers at most | system's mean | $r_{max}/r_{min}$ |
| | $GSAGen_l$ Avg | | Not used | Each item in this set is randomly chosen and rated by two attackers at most | each item's mean | $r_{max}/r_{min}$ |

**Fig 1**. **General scheme of the group shilling attack.**

strict rules and limits the user groups to be injected into the system. Therefore, in this study, the *GSAGen$_I$* model is chosen to create a attack profile called multi-criteria group shilling attack (MC-GSA). Another critical point is that for MC-GSA, we need to decide how to design the attack and determine attack-specific parametric values in the system consisting of multiple criteria.

The overall framework of the proposed method, which performs the robustness analysis of multi-criteria group recommender systems, is illustrated in Fig 2. In this section, we first explain the design of the *GSAGen$_I$* attack model and describe how the parametric values specific to the multi-criteria setting are determined.

As shown in Fig 2, the framework outlines the complete workflow of the proposed MC-GSA model, including the generation of attack profiles, clustering of users into groups, and the construction of group-based recommendation lists. To enhance the clarity and readability of the diagram, a legend has been added below, defining all symbols and notations used in the figure.

**Legend:**

$u_i$ : Normal users in the training dataset
$a_i$ : Attack users in the shilling groups
$r_{max}$ : Maximum rating value assigned to target items
$\gamma_{ij}$ : Rating values for filler items
$G_n$ : Formed user groups after the clustering process
*Top-n List* : Final recommended items for each group

After introducing the overall structure, the operation of the proposed model can be described in four main steps, each representing a distinct stage of the robustness analysis process.

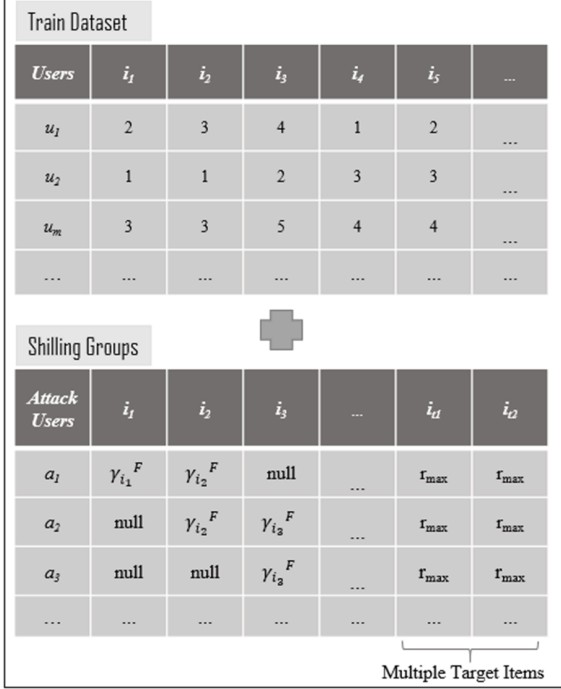
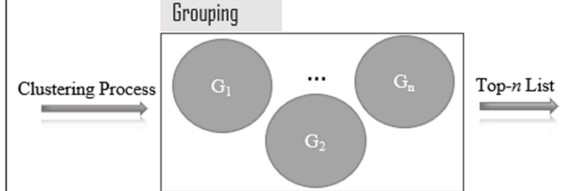

**Fig 2**. **The framework of MC-GSA.**

- **Training Dataset:** The initial dataset consists of genuine users ($u_1, u_2, ..., u_m$) who provide ratings for multiple items ($i_1, i_2, ..., i_k$) across different criteria. This dataset reflects the normal operation of the recommender system.
- **Shilling Groups:** To simulate the attack, synthetic user profiles ($a_1, a_2, ...$) are generated and injected into the dataset. These attack profiles apply specific strategies, such as assigning maximum ratings ($r_{max}$) to target items or using pre-defined filler values ($\gamma^F$), in order to bias the recommendation outcome.
- **Clustering Process:** The combined dataset of genuine and attack users is processed through a clustering algorithm. Users are grouped into clusters ($G_1, G_2, ..., G_n$) according to similarities in their rating patterns, which reflects the group-based recommendation scenario.
- **Top-*n* Recommendation Lists:** For each group, a Top-*n* list is generated. As illustrated in the figure, some of the target items (highlighted in red) are promoted within the recommendation lists, revealing the manipulative effect of group shilling attacks on multi-criteria group recommender systems.

This framework demonstrates how adversarial user profiles can significantly distort group recommendation results and emphasizes the importance of designing robust defense mechanisms against such attacks.

## 4.1 Designing attack specific-parameters

The contents of MC-GSA profiles designed on multi-criteria group recommendation systems are shown in the Table 2. According to Table 2, the group attack user profile consists of the following in the MC-GSA:

- **Selected Items ($I_S$)** is empty for the MC-GSA ($GSAGen_I$) model.
- **Filler Items ($I_F$):** The ratings are the set of fillers provided by attacker *i* in the attack group. Fillers are very important in group shilling attacks. The item sets in traditional shilling attacks are chosen randomly. In contrast, the selection of fillers in group shilling attacks is more meticulous.
- **Target Items ($I_T$),** Target product rating value gets the maximum or minimum rating value depending on the purpose of the attack. In the literature study [48], it has been seen that the choice of target product is significant for the success of the attack. Therefore, target item selection is carried out using three different methods.
  - *Most Unpopular - Unsatisfaction Ratio (MUP-UsR):* In the system, the items with the least satisfaction ratio are selected. The items with the highest ratio of the rating value below the threshold value to the total number of ratings given for that product are selected. It can be calculated using the formula given in Eq 1:

$$UsR_i = \frac{\sum_{u \in rated_i} \mathbb{1}(r_{u,i} <= threshold)}{|U_i|} \tag{1}$$

  where, $|U_i|$ is number of rated for item *i*.
  - *Most Unpopular - Dominant Ratio (MUP-DR):* These are the items where dissatisfaction is more dominant. It is calculated as the ratio of the number of rated for the item higher than a certain threshold value to the number of rated lower than the threshold value. As a result of the calculation, the items with the lowest ratio are selected. It can be

**Table 2**. MC-GSA attack user group profile contents.

| Version | Type | $I_S$ | $I_F$ | $I_T$ |
|---|---|---|---|---|
| $GSAGen_I$ | $GSAGen_IRan$ | Empty | Each product is randomly selected and rated by at most two attacker System's mean | Multiple Targets: rmax for push attacks |
| | $GSAGen_IAvg$ | Empty | Each product is randomly selected and rated by at most two attacker Item's mean | Multiple Targets: rmax for push attacks |

calculated using the formula given in Eq 2:

$$DR_i = \sum_{u \in rated_i} \frac{\mathbb{1}(r_{u,i} > threshold)}{\mathbb{1}(r_{u,i} <= threshold)} \tag{2}$$

where, $|U_i|$ is number of rated for item $i$.
- *Most Unpopular - Number of Non Zero (MUP-NNZ):* Target items are those that have received the fewest user ratings.
- *Most Unpopular - Sum (MUP-Sum):* Target items are those that have received the fewest sum of user rating values.

Following the design of attack user groups, statistical parameters need to be addressed. In single-criteria systems, the statistical parameters required for the attack profile are produced by using all rating values. However, including more than one criterion in multi-criteria systems may allow these attack-specific features to be predicted in more than one way. For the purpose, in this study, the following three different scenarios (overall-based, criteria-based, aggregated-based) are taken into account to calculate these parametric values on the multi-criteria system:

*General-based ($S_1$):* In this scenario, which reflects the most reliable information in a general-based, multi-criteria system, the process is progressed like a single-criteria system. Therefore, the selection of parametric values and target/filler products required for the attack is made only based on overall criteria.

*Criteria-based ($S_2$):* Criterion-based scenario is intuitively the best way to produce results using all the information available with the system in an encapsulated way, as separate values for each subcriteria can help attack profiles resemble real users [49]. In this scenario, the attack process for each criterion is progressed one by one and the average is taken into account for performance evaluation.

## 4.2 Attack methodology for MC-GSA

In this section, the steps of MC-GSA methodologies performed on multi-criteria group recommendation systems are given.

When the algorithmic steps of the MC-GSA method given in Algorithm 1 are examined, the GSAGen method, which has minimal relationships with each other, is adopted when creating attack profiles, unlike traditional methods. After the creation of fake user groups, a grouping process is performed with these profiles added to the system and the attack process is started. Each of the steps such as producing predictions and preparing recommendation lists for user groups are carried out according to three different scenarios ($S_1$, $S_2$). While in the $S_1$, the process is run on the overall criterion, in the second scenario ($S_2$), the process is run for each criterion and the average of the performance evaluation is taken. In the final stage, a list of suggestions is created and the success of the attack is evaluated.

***Complexity Note.*** The overall computational complexity of the proposed MC-GSA algorithm can be expressed as $O(S K_f [(U+A)k_c td + GH(C+|g| + \log H) + I + AC])$, where $U$ is the number of users, $I$ is the number of items, $C$ is the number of criteria, $A$ is the number of injected attack profiles, $G$ is the number of groups, $|g|$ is the average group size, $k_c$ is the number of clusters, $t$ is the number of clustering iterations, and $H$ represents the total number of evaluated items ($100 + n$). The dominant computational cost arises from the clustering and prediction phases, which scale approximately linearly with the number of users and items. Therefore, the method is computationally feasible for medium-scale multi-criteria datasets and can be further optimized through parallelization or sampling for large-scale applications.

In practice, however, the runtime behaviour is also affected by the attack design: since push attacks are applied to multiple target products concurrently, the procedure generates and evaluates synthetic user profiles on-the-fly. These synthetic profiles consist of filler and target item sets (MUP-UsR, MUP-DR, MUP-NNZ, MUP-Sum), as given in Table 2.

**Algorithm 1.  MC-GSA methodology.**

**Input:** `Multi-criteria dataset` $\mathcal{D}_{n \times m}$
**Output:** `Group recommender lists; performance results`
**Function** $MC\_GSA(\mathcal{D}_{n \times m})$

 **foreach** $(S_1, S_2)$ **do**

 `Divide the dataset into probe` $P$ `and train set` $M$ `using` $k$-`fold;`
 `Identify target items in` $M$ `(see Sect 4.1);`
 `Select` $n$ `target products where` $n$ `is the number of items presented to user groups;`
 `Generate synthetic group attack profiles using target items and add them to` $M$`;`
 `Start the attack process;`
 `Cluster the dataset with the added group attack profiles;`
 `Select the first 100 products least evaluated by the users;`
 `Add` $n$ `target products to these least-rated items for attack tests;`
 `Generate estimated values for` $100+n$ `products for each group;`
 `Prepare a top-`$n$ `recommendation list for the user groups using the ordered list;`

 **end**

 `Determine the number of target products and their order in the list;`

## 5 Experiments

The section examines the impact of MC-GSA on group recommendation systems. First, the data set, experimental design and performance evaluation metrics used in the study are briefly mentioned.

### 5.1 Dataset

In the study, Yahoo! Movies (YM), a well-known multi-criteria data set in the field of RS and group RS, is used as the data set [50,51]. These researchers collected rating data from the Yahoo!Movies website with a web crawler. All considered movies are part of the single-rating dataset provided by Yahoo!Research (https://webscope.sandbox.yahoo.com). In the YM data set, users stated their preferences information based on four criteria, i.e., acting, story, direction, visuals. In addition to the individual criteria ratings, users made an overall grade for each movie to reflect their overall opinion. We utilize two subsets of the dataset, the YM10 and YM20 data sets which is converted to a 1-5 scale. The YM10 dataset includes data with at least 10 reviews for each user and movie, while the YM20 dataset sets the threshold at 20 reviews.

Information on the datasets are shown in Table 3.

### 5.2 Experiment design

Our experiments split the dataset into test and training using 10-fold, based on the attack methodology outlined in the previous sections. Attack profiles are created using two different attack models, i.e., $GSAGen_lRan$, $GSAGen_lAvg$ with the target products obtained using the training data set. The training dataset is updated including attack profiles. Then, estimated values are produced by adding $n$ such as 5, 10, and 15, target products to 100 products that the user $u$ is not rated from the test dataset due to the small sizes of the subsets within the dataset, the sizes of the excluded products are also designated as small. From the list of $100+n$ products, the user $u$ is presented with a list of $n$ products that they may like the most, and a performance evaluation is made.

**Table 3**. Information about YM sub-datasets.

| Dataset | #Users | #Items | #Ratings |
|---------|--------|--------|----------|
| YM20 | 429 | 491 | 18.504 |
| YM10 | 1827 | 1471 | 48.026 |

The parameter selections required for the schema are given below.

**Attack user profiles**: The synthetic user profiles to be added to the system consist of $I_f$s and $I_t$s according to the MC-GSA model. How to select these products has been described in the previous sections. Since the aim of the attack is to include the target products in top-$n$ (push attack), the rating value of the target items is determined as $r_{max}$ (=5). How many synthetic profiles will be added to the system varies for each data set. The previous studies have shown that choosing this ratio between 5% and 10% is very effective [52,53]. However, in this study, we set the attack size as 0.1 since the change in attack size affects the cost/benefit of the attack; a value greater than 10% is not considered appropriate in real-world applications [54]. Therefore, in this study, the number of attack users is chosen as 183, 43 (10%) for YM10 and YM20, respectively.

**Filler item selection**: Two different methods used for the selection of filler products are given in the previous section (Sect 3.2). While the number of filler items in the synthetic user profile for the YM10 dataset is 15 (1% of items in the dataset), 74 (5%), these values are 4 and 21 for the YM20 dataset, respectively.

**Target item selection**: It has been stated in the previous sections that the target products are selected by four different methods (MUP-UsR, MUP-DR, MUP-NNZ and MUP-Sum). In the first method, MUP-UsR, which contains high information, the products with the lowest satisfaction rates in the system are selected. Specifically, the products with the highest ratio of scores below the threshold value to the total number of scores given for that product are chosen. The other method, MUP-DR, which contains low information, focuses on products where dissatisfaction is particularly dominant. Similarly, in other methods with low information (MUP-NNZ, MUP-Sum), the number of evaluations and scores for all products is taken into account.

The number of target products to be selected from the first 50 products determined according to these three methods will be as much as the number of $n$ products in the top-$n$ recommendation list to be presented to the user. Therefore, 5, 10, and 15 multiple-target items are randomly selected from 50 products in this study.

**Test variations**: In this study, we used 2 filler item selection methods, 2 filler item sizes, 4 target product selection types, 3 target item size, 1 attack size on 2 data sets (YM10, YM20) and 2 scenarious. Selection methods and the created attack user profiles are carried out for each criterion in the multi-criteria data set by following the approach steps adopted by Adomavicius [1] and also used overall criterion. All calculations and experimental results are created using Matlab R2022a.

## 5.3 Evaluation metrics

Robustness metrics have been developed to evaluate these GRS and RS attacks [52,53]. For example, Average Hit Ratio (Average Hit Ratio, AvgHR) calculates the percentage of users whose target products are included in the top-$n$ list, and Weighted Rank Value (WRV), which is a proposed new metric by researchers [5] that shows the order in which the target products are included in the top-$n$.

**AvgHR**:It is the rate of the target products in the top-$n$ list created for the user's groups. In the metric, the number of multiple target products included in the list is divided by the list size to find the hit rate of the user [5]. A high value in the AvgHR metric calculated using Eq 3 indicates that the attack is highly effective.

$$HitRatio_u = \frac{\sum_{i \in IT_T} H_{ui}}{n}$$

$$AvgHR = \frac{\sum_{i \in U_T} HitRatio_u}{|U_T|}$$

(3)

Here, $IT_T$ and $U_T$ are the set of target products and users, respectively. $n$ is the size of the recommendation list presented to the user.

**WRV**: In the new metric proposed by Kaya and Kaleli [5], the order in which the target products are presented in the manipulated recommendation lists is discussed. Here, the approach has been adopted that the target products' ranking at the top can have a significant impact on the success of the attack. Therefore, the success of the attack will be shown by specifying the order in which the target products are included in top-*n*. Obtaining a high WRV calculated using the Eq 4 shows that the attack is quite effective.

$$Degree_u = \sum_{i \in IT_T} \frac{n}{Rank_{ui}}$$

(4)

$$WRV = \frac{\sum_{i \in U_T} Degree_u}{|U_T|}$$

The range of values obtained with the new metric is 0-11.4167, 0-29.2897, 0-49.7734 for top-5, top-10, and top-15, respectively.

## 6 Experimental results

In this study, the effects of group shilling attacks on multi-criteria systems are examined in two different data sets, for different parameter values and two different scenarios. The performance of the attack on this system when the user groups are presented with top-5,10,15 product recommendation lists for the YM20 data set is given in Table 4. The table examines the effects of different types of attacks on recommendation systems, *GSAGen_lRan* and *GSAGen_lAvg*, cluster sizes (1%, 5%, 10%) and target item selection strategies (MUP-DR, MUP-NNZ, MUP-Sum), on the system within the scope of two different scenarios ($S_1$ and $S_2$). Performance measurements are evaluated based on the rates of the target items exposed to the attack in the recommendation lists being included in different top-*n* ranges (Top-5, Top-10, Top-15). Table 4 reports the AvgHR results of multi-criteria group shilling attacks on the YM20 dataset for a filler size of 1%. AvgHR reflects the average number of target items that appear in the Top-*n* recommendation lists. For instance, an AvgHR value of 0.05 at Top-10 means that, on average, $10 \times 0.05 = 0.5$ target items are included in each recommendation list, i.e., approximately one fake item appears in every two lists. This provides a clear indication of how strongly the attack is able to infiltrate the recommendation outcomes.

A general examination of the results shows that as the filler size increases, AvgHR values rise significantly. This implies that injecting more attack profiles into the system strengthens the impact of the attack, thereby increasing the number of target items in the recommendation lists. For example, under *GSAGen_lRan*, the MUP-NNZ strategy increases from 0.0500 at Top-10 with 1% filler to 0.0970 at 10% filler, nearly doubling the success of the attack. When analyzing target item selection strategies, MUP-UsR yields relatively low AvgHR values. Since this strategy relies only on dissatisfaction ratios, its ability to penetrate the recommendation process is limited. For example, at 1% filler and Top-10, the AvgHR value is 0.0294, indicating only 0.29 fake items per list on average. MUP-DR performs slightly better than UsR, as it considers the balance between ratings above and below a threshold. At 5% filler and Top-10, its AvgHR value is 0.0442, corresponding to approximately 0.44 fake items per list. The strongest results are observed for MUP-NNZ. This strategy targets items with the fewest ratings, which are typically less popular and therefore easier to push into higher ranks. Under *GSAGen_lRan* with 10% filler and Top-10, the AvgHR value reaches 0.0970, meaning that nearly every recommendation list contains one attack item. This makes MUP-NNZ the most vulnerable strategy against shilling attacks. MUP-Sum (Sum of Ratings) is also considerably affected, although slightly less than NNZ. For instance, under *GSAGen_lAvg* with 5% filler at Top-10, the AvgHR value is 0.0731, implying that around 0.73 fake items are included per list. Overall, MUP-NNZ emerges as the weakest and most vulnerable strategy, while MUP-UsR and MUP-DR exhibit relatively more robust behavior. In these cases, fake items appear less frequently in the recommendation lists, particularly when the filler size is low. Furthermore, AvgHR values under *GSAGen_lAvg* are generally higher than those under *GSAGen_lRan*, indicating

**Table 4. AvgHR results of multi-criteria recommender systems on YM20 dataset for filler size 1%.**

| Attack Type | Cluster Size | Target Item Selection | $S_1$ | | | $S_2$ | | |
|---|---|---|---|---|---|---|---|---|
| | | | Top-5 | Top-10 | Top-15 | Top-5 | Top-10 | Top-15 |
| *GSAGen_lRan* | 1% | MUP-UsR | 0.0189 | 0.0294 | 0.0493 | 0.0136 | 0.0287 | 0.0554 |
| | | MUP-DR | 0.0222 | 0.0367 | 0.0600 | 0.0638 | 0.0678 | 0.0697 |
| | | MUP-NNZ | **0.0378** | **0.0500** | **0.0837** | 0.0569 | 0.0585 | 0.0613 |
| | | MUP-Sum | 0.0289 | 0.0344 | 0.0580 | **0.0880** | **0.0940** | **0.0969** |
| | 5% | MUP-UsR | 0.0110 | 0.0473 | 0.0657 | 0.0150 | 0.0463 | 0.0714 |
| | | MUP-DR | 0.0203 | 0.0442 | 0.0663 | 0.0551 | 0.0722 | 0.0834 |
| | | MUP-NNZ | **0.0426** | **0.0767** | **0.1060** | 0.0435 | 0.0576 | 0.0704 |
| | | MUP-Sum | 0.0149 | 0.0484 | 0.0642 | **0.0611** | **0.0841** | **0.1049** |
| | 10% | MUP-UsR | 0.0401 | 0.0713 | **0.1156** | 0.0380 | 0.0760 | 0.1035 |
| | | MUP-DR | 0.0409 | 0.0854 | 0.1122 | **0.0402** | **0.0815** | **0.1060** |
| | | MUP-NNZ | **0.0618** | **0.0970** | 0.0911 | 0.0322 | 0.0647 | 0.1011 |
| | | MUP-Sum | 0.0398 | 0.0709 | 0.1090 | 0.0271 | 0.0634 | 0.0978 |
| *GSAGen_lAvg* | 1% | MUP-UsR | 0.0144 | 0.0317 | 0.0487 | 0.0135 | 0.0340 | 0.0572 |
| | | MUP-DR | 0.0206 | 0.0356 | 0.0604 | **0.1179** | **0.1229** | **0.1263** |
| | | MUP-NNZ | **0.0361** | **0.0567** | **0.0870** | 0.0893 | 0.0930 | 0.0959 |
| | | MUP-Sum | 0.0233 | 0.0433 | 0.0678 | 0.0497 | 0.0527 | 0.0548 |
| | 5% | MUP-UsR | 0.0197 | 0.0385 | 0.0727 | 0.0180 | 0.0395 | 0.0673 |
| | | MUP-DR | 0.0157 | 0.0478 | 0.0661 | **0.0767** | **0.0920** | **0.1106** |
| | | MUP-NNZ | **0.0429** | **0.0758** | **0.1057** | 0.0633 | 0.0818 | 0.0960 |
| | | MUP-Sum | 0.0163 | 0.0471 | 0.0677 | 0.0350 | 0.0527 | 0.0684 |
| | 10% | MUP-UsR | **0.0459** | 0.0737 | **0.1104** | **0.0572** | **0.0824** | 0.1076 |
| | | MUP-DR | 0.0336 | **0.0780** | 0.1090 | 0.0541 | 0.0781 | **0.1085** |
| | | MUP-NNZ | 0.0443 | 0.0745 | 0.0981 | 0.0313 | 0.0638 | 0.0960 |
| | | MUP-Sum | 0.0157 | 0.0483 | 0.0804 | 0.0287 | 0.0714 | 0.1055 |

that systematic selection of target items increases attack success compared to random selection. Additionally, as the Top-*n* value increases (e.g., from Top-5 to Top-15), the likelihood of including fake items grows, since longer lists offer more opportunities for target items to appear.

In summary, the results highlight that the effectiveness of multi-criteria group shilling attacks strongly depends on both the filler size and the target item selection strategy. Attacks focusing on sparsely rated items (e.g., MUP-NNZ) demonstrate the highest vulnerability, while dissatisfaction-based strategies (MUP-UsR and MUP-DR) are more resilient. These findings emphasize the necessity of incorporating robust defense mechanisms, particularly against vulnerabilities associated with less popular items, in the design of multi-criteria group recommender systems.

Table 5 presents the AvgHR results of multi-criteria group shilling attacks on the YM20 dataset with a filler size of 5%. The findings clearly show that the impact of the attack becomes more pronounced under this setting. For instance, under *GSAGen_lRan* with the MUP-Sum strategy, the AvgHR at Top-10 reaches 0.1211, which corresponds to approximately 12% of the items in the recommendation list. In practical terms, this implies that on average one fake item is included in every Top-10 list. At Top-15, the value rises to 0.1248, meaning that nearly two manipulated items are regularly injected into the recommendations. The MUP-DR strategy also yields relatively high values. At a 1% cluster size, the Top-10 AvgHR is observed at 0.0929, which equates to around 9% of the items being target products. This ratio is almost three times higher compared to MUP-UsR under the same setting, highlighting that the DR-based selection renders the system more vulnerable to manipulation. MUP-NNZ produces moderately strong outcomes. For example, under *GSAGen_lAvg* with a 5% cluster size, the Top-10 AvgHR is 0.1067, corresponding to about 10% of the recommended items. This result indicates that sparsely rated items, when targeted, can be effectively pushed into the recommendation lists, exposing a significant weakness of the system against this type of strategy. In contrast, the MUP-UsR approach generates the lowest AvgHR values. With a 10% cluster size at Top-10, the metric reaches only 0.0667, implying that roughly 6–7% of the items are fake. This suggests that UsR-based target selection has a limited impact, thereby making the system more resistant

**Table 5**. AvgHR results of multi-criteria recommender systems on YM20 dataset for filler size 5%.

| Attack Type | Cluster Size | Target Item Selection | $S_1$ | | | $S_2$ | | |
| --- | --- | --- | --- | --- | --- | --- | --- | --- |
| | | | Top-5 | Top-10 | Top-15 | Top-5 | Top-10 | Top-15 |
| GSAGen_lRan | 1% | MUP-UsR | 0.0183 | 0.0325 | 0.0433 | 0.0151 | 0.0288 | 0.0496 |
| | | MUP-DR | 0.0927 | 0.0929 | 0.0949 | 0.0502 | 0.0525 | 0.0567 |
| | | MUP-NNZ | 0.0783 | 0.0795 | 0.0812 | 0.0900 | 0.0934 | 0.0954 |
| | | MUP-Sum | **0.1118** | **0.1211** | **0.1248** | **0.0938** | **0.0968** | **0.0989** |
| | 5% | MUP-UsR | 0.0194 | 0.0438 | 0.0803 | 0.0206 | 0.0437 | 0.0698 |
| | | MUP-DR | 0.0719 | 0.0806 | 0.1028 | 0.0419 | 0.0621 | 0.0860 |
| | | MUP-NNZ | 0.0489 | 0.0538 | 0.0699 | 0.0673 | 0.0854 | 0.0993 |
| | | MUP-Sum | **0.0766** | **0.0977** | **0.1250** | **0.0755** | **0.0903** | **0.0979** |
| | 10% | MUP-UsR | 0.0437 | **0.0836** | **0.1156** | 0.0399 | 0.0709 | 0.1024 |
| | | MUP-DR | 0.0373 | 0.0819 | 0.1136 | **0.0424** | 0.0730 | 0.1049 |
| | | MUP-NNZ | 0.0336 | 0.0401 | 0.0726 | 0.0326 | 0.0529 | 0.0964 |
| | | MUP-Sum | **0.0646** | 0.0638 | 0.1034 | 0.0361 | **0.0767** | **0.1059** |
| GSAGen_lAvg | 1% | MUP-UsR | 0.0217 | 0.0289 | 0.0556 | 0.0153 | 0.0298 | 0.0577 |
| | | MUP-DR | 0.0812 | 0.0832 | 0.0860 | 0.0774 | 0.0828 | 0.0884 |
| | | MUP-NNZ | **0.1267** | **0.1309** | **0.1327** | **0.1089** | **0.1135** | **0.1177** |
| | | MUP-Sum | 0.0864 | 0.0925 | 0.0950 | 0.0959 | 0.1005 | 0.1026 |
| | 5% | MUP-UsR | 0.0146 | 0.0341 | 0.0744 | 0.0129 | 0.0431 | 0.0726 |
| | | MUP-DR | 0.0662 | 0.0782 | 0.0885 | 0.0679 | 0.0956 | 0.1125 |
| | | MUP-NNZ | **0.0909** | **0.1167** | **0.1258** | **0.0849** | **0.1081** | **0.1289** |
| | | MUP-Sum | 0.0799 | 0.0913 | 0.1055 | 0.0670 | 0.0847 | 0.0973 |
| | 10% | MUP-UsR | **0.0437** | 0.0667 | 0.1081 | **0.0497** | 0.0760 | 0.1075 |
| | | MUP-DR | 0.0310 | **0.0844** | 0.1140 | 0.0428 | **0.0815** | **0.1103** |
| | | MUP-NNZ | 0.0262 | 0.0565 | 0.0759 | 0.0261 | 0.0609 | 0.0936 |
| | | MUP-Sum | 0.0396 | 0.0714 | **0.1179** | 0.0273 | 0.0758 | 0.1010 |

to such attacks compared to other strategies. A general trend is also observed regarding the effect of cluster size: as the cluster size increases, AvgHR values consistently rise. For instance, under GSAGen_lRan at Top-10, MUP-UsR increases from approximately 3% to over 8% as the cluster size grows. Similarly, MUP-Sum increases from 9% at 5% cluster size to nearly 12% at 10% cluster size. This indicates that larger groups provide greater leverage for injected attack profiles, thereby amplifying the attack's overall effectiveness.

In summary, when the filler size is set to 5%, the most significant impact is observed for MUP-Sum and MUP-DR, which both achieve AvgHR values in the range of 10–12%. MUP-NNZ remains moderately strong but stable, while MUP-UsR shows the lowest ratios, marking it as the most robust selection strategy against group shilling attacks. These results demonstrate that the choice of target item selection strategy critically influences the attack success, with sum- and dominance-based approaches posing the highest threat to multi-criteria group recommender systems.

A comparative analysis of Tables 4 and 5 reveals that increasing the filler size from 1% to 5% substantially amplifies the success of the attacks. At a filler size of 1%, AvgHR values remain relatively low, with Top-10 lists typically containing only 3–5% of fake items. For example, under MUP-UsR, AvgHR values are observed around 0.029–0.038, indicating that, on average, 0.3–0.4 target items are present in each list of ten. Similarly, MUP-DR yields values in the 4–6% range, while MUP-NNZ produces slightly higher ratios of approximately 5%. Interestingly, MUP-Sum already shows stronger results under GSAGen_lRan, achieving 0.094 at Top-10, corresponding to nearly one fake item in every list. When the filler size is raised to 5%, the impact of the attacks intensifies considerably, with AvgHR values nearly doubling to reach the 8–12% range. In particular, MUP-Sum and MUP-DR exhibit the most significant increases, both achieving Top-10 values exceeding 0.10. For instance, under GSAGen_lRan, MUP-Sum reaches 0.1211 at Top-10, which means that approximately 12% of the items in the recommendation lists are fake. Similarly, under GSAGen_lAvg, both MUP-DR and MUP-NNZ surpass the 10% threshold, highlighting the heightened vulnerability of the system under higher filler conditions. In summary, raising the filler size from 1% to 5% nearly doubles the effectiveness of the attacks. While the influence of shilling profiles remains

relatively limited at 1%, with only sporadic infiltration of recommendation lists, a filler size of 5% results in approximately one out of every ten recommended items being fake. This finding emphasizes that filler size is a critical factor in determining attack success, and further demonstrates that strategies such as MUP-Sum and MUP-DR pose the greatest threat to multi-criteria group recommender systems under higher filler conditions.

A comparative examination of Tables 6 and 7 indicates that increasing the filler size from 1% to 5% substantially amplifies the effectiveness of group shilling attacks. Under a filler size of 1%, AvgHR values remain relatively moderate, with the proportion of fake items in the Top-10
lists typically ranging between 6% and 12%. In contrast, when the filler size is set to 5%, these ratios increase to between 9% and 15%, clearly demonstrating that the greater the number of filler ratings injected into the system, the stronger the influence of the attack on recommendation outcomes. For the MUP-UsR strategy, AvgHR values at 1% filler remain around 0.0643 (approximately 6.4%), which corresponds to fewer than one fake item in a typical Top-10 recommendation list. However, with a filler size of 5%, this range increases to 0.0841–0.1185, meaning that 8–11% of the recommended items are manipulated. Although UsR yields the lowest values overall, its effectiveness still grows with larger filler sizes, indicating that even this relatively robust strategy becomes increasingly vulnerable. The MUP-DR method shows AvgHR values of about 0.0838 (8.4%) under 1% filler, but these values climb to 0.1037–0.1210 (10–12%) under 5% filler. This growth highlights the sensitivity of the DR strategy to filler size, as larger proportions of injected profiles significantly increase the number of fake items appearing in the lists. The MUP-NNZ strategy produces the highest AvgHR values across both settings. At 1% filler, Top-10 results reach 0.1231 (around 12%), while under 5% filler the values rise further to 0.1262–0.1385 (13–14%). This demonstrates that sparsely rated items are the most vulnerable targets, as they can be effectively pushed into recommendation lists regardless of filler size, making MUP-NNZ the weakest selection strategy. The MUP-Sum strategy records AvgHR values in the range of 0.0791–0.1072 at 1% filler. When the filler size is increased to 5%, the values rise to 0.0852–0.1194, representing a 2–3% increase in

**Table 6**. AvgHR results of multi-criteria recommender systems on YM10 dataset for filler size 1%.

| Attack Type | Cluster Size | Target Item Selection | $S_1$ | | | $S_2$ | | |
|---|---|---|---|---|---|---|---|---|
| | | | Top-5 | Top-10 | Top-15 | Top-5 | Top-10 | Top-15 |
| GSAGen_lRan | 1% | MUP-UsR | 0.0265 | 0.0643 | 0.0827 | 0.0586 | 0.0715 | 0.1174 |
| | | MUP-DR | 0.0818 | 0.0838 | 0.0845 | 0.1116 | 0.1161 | 0.1207 |
| | | MUP-NNZ | **0.1209** | **0.1231** | **0.1249** | **0.1853** | **0.1903** | **0.1933** |
| | | MUP-Sum | 0.0771 | 0.0791 | 0.0823 | 0.1056 | 0.1091 | 0.1086 |
| | 5% | MUP-UsR | 0.0358 | 0.0552 | 0.0658 | 0.0385 | 0.0457 | 0.0802 |
| | | MUP-DR | 0.0619 | 0.0761 | 0.0811 | 0.0840 | 0.0999 | 0.1272 |
| | | MUP-NNZ | **0.0818** | **0.0909** | **0.1060** | **0.1249** | **0.1475** | **0.1528** |
| | | MUP-Sum | 0.0408 | 0.0563 | 0.0663 | 0.0971 | 0.1288 | 0.1253 |
| | 10% | MUP-UsR | **0.0480** | 0.0702 | **0.0954** | 0.0231 | 0.0592 | 0.1164 |
| | | MUP-DR | 0.0379 | **0.0830** | 0.0928 | **0.0583** | 0.0649 | **0.1320** |
| | | MUP-NNZ | 0.0394 | 0.0616 | 0.0875 | 0.0573 | **0.1188** | 0.1034 |
| | | MUP-Sum | 0.0441 | 0.0597 | 0.0854 | 0.0234 | 0.1065 | 0.1072 |
| GSAGen_lAvg | 1% | MUP-UsR | 0.0275 | 0.0496 | 0.0895 | 0.0225 | 0.0486 | 0.0886 |
| | | MUP-DR | 0.1018 | 0.1033 | 0.1077 | **0.1443** | **0.1474** | **0.1505** |
| | | MUP-NNZ | 0.1005 | 0.1026 | 0.1043 | 0.0820 | 0.0840 | 0.0870 |
| | | MUP-Sum | **0.1300** | **0.1309** | **0.1341** | 0.0790 | 0.0835 | 0.0881 |
| | 5% | MUP-UsR | 0.0394 | 0.0623 | 0.0739 | 0.0405 | 0.0634 | 0.0910 |
| | | MUP-DR | 0.0610 | 0.0711 | 0.0876 | 0.0772 | 0.0948 | **0.1080** |
| | | MUP-NNZ | 0.0645 | 0.0758 | 0.0912 | 0.0637 | **0.0998** | 0.1049 |
| | | MUP-Sum | **0.0959** | **0.1072** | **0.1227** | **0.0959** | 0.0903 | 0.1061 |
| | 10% | MUP-UsR | 0.0325 | **0.0789** | 0.0928 | **0.0457** | 0.0754 | **0.1126** |
| | | MUP-DR | **0.0402** | 0.0569 | **0.0950** | 0.0239 | 0.0781 | 0.1081 |
| | | MUP-NNZ | 0.0278 | 0.0383 | 0.0859 | 0.0180 | **0.1021** | 0.0925 |
| | | MUP-Sum | 0.0249 | 0.0634 | 0.0779 | 0.0264 | 0.0758 | 0.1123 |

**Table 7**. AvgHR results of multi-criteria recommender systems on YM10 dataset for filler size 5%.

| Attack Type | Cluster Size | Target Item Selection | $S_1$ | | | $S_2$ | | |
|---|---|---|---|---|---|---|---|---|
| | | | Top-5 | Top-10 | Top-15 | Top-5 | Top-10 | Top-15 |
| $GSAGen_lRan$ | 1% | MUP-UsR | 0.0260 | 0.0462 | **0.0921** | 0.0410 | 0.0895 | 0.0958 |
| | | MUP-DR | 0.0771 | 0.0805 | 0.0845 | **0.1474** | **0.1507** | **0.1569** |
| | | MUP-NNZ | **0.0869** | **0.0881** | 0.0918 | 0.1263 | 0.1262 | 0.1267 |
| | | MUP-Sum | 0.0852 | 0.0877 | 0.0904 | 0.0470 | 0.0536 | 0.0569 |
| | 5% | MUP-UsR | 0.0339 | 0.0483 | 0.0763 | 0.0474 | 0.0664 | 0.0885 |
| | | MUP-DR | 0.0524 | 0.0695 | 0.0852 | **0.0996** | **0.1418** | **0.1449** |
| | | MUP-NNZ | 0.0546 | 0.0684 | 0.0917 | 0.0743 | 0.0813 | 0.0945 |
| | | MUP-Sum | **0.0709** | **0.0852** | **0.0952** | 0.0295 | 0.0648 | 0.0832 |
| | 10% | MUP-UsR | **0.0606** | 0.0684 | 0.0915 | 0.0499 | 0.0681 | **0.1172** |
| | | MUP-DR | 0.0528 | 0.0734 | 0.0984 | **0.0529** | 0.0713 | 0.1080 |
| | | MUP-NNZ | 0.0389 | 0.0605 | **0.1035** | 0.0147 | 0.0539 | 0.0761 |
| | | MUP-Sum | 0.0458 | **0.0750** | 0.0846 | 0.0272 | **0.0764** | 0.0852 |
| $GSAGen_lAvg$ | 1% | MUP-UsR | 0.0288 | 0.0570 | 0.0974 | 0.0352 | 0.0758 | 0.1151 |
| | | MUP-DR | **0.1023** | **0.1037** | **0.1066** | 0.1341 | 0.1364 | 0.1386 |
| | | MUP-NNZ | 0.0658 | 0.0699 | 0.0733 | **0.1436** | **0.1385** | **0.1392** |
| | | MUP-Sum | 0.0990 | 0.1005 | 0.1032 | 0.0750 | 0.0776 | 0.0802 |
| | 5% | MUP-UsR | 0.0255 | 0.0362 | 0.0690 | 0.0803 | **0.1185** | 0.0928 |
| | | MUP-DR | **0.1099** | **0.1062** | **0.1210** | **0.0983** | 0.1126 | **0.1199** |
| | | MUP-NNZ | 0.0459 | 0.0804 | 0.0915 | 0.0821 | 0.0852 | **0.1121** |
| | | MUP-Sum | 0.0812 | 0.0914 | 0.1049 | 0.0539 | 0.0771 | 0.0830 |
| | 10% | MUP-UsR | 0.0230 | **0.0782** | 0.0811 | **0.0748** | 0.0634 | 0.0939 |
| | | MUP-DR | **0.0489** | 0.0608 | 0.0946 | 0.0650 | **0.0926** | **0.1202** |
| | | MUP-NNZ | 0.0300 | 0.0534 | 0.0918 | 0.0299 | 0.0515 | 0.0729 |
| | | MUP-Sum | 0.0390 | 0.0646 | **0.1051** | 0.0385 | 0.0633 | 0.1159 |

attack penetration. This indicates that the Sum strategy, although not as extreme as NNZ, still exposes the system to significant vulnerabilities.

In conclusion, the analysis on the YM10 dataset highlights that filler size is a critical factor in determining the success of multi-criteria group shilling attacks. With a filler size of 1%, the impact remains relatively limited, whereas at 5% filler nearly 15% of Top-10 recommendations consist of fake items. Among the strategies, MUP-NNZ consistently emerges as the most vulnerable, while MUP-DR and MUP-Sum also display considerable susceptibility. By contrast, MUP-UsR proves to be comparatively more robust, though its resilience diminishes as filler size increases.

A comparison of the results for filler sizes of 1% and 5% on the YM10 dataset clearly demonstrates the substantial increase in attack effectiveness as the filler ratio grows. At a filler size of 1%, AvgHR values remain relatively limited, with the proportion of fake items in the Top-10 recommendation lists generally ranging between 6% and 12%. Under this setting, the influence of shilling profiles on recommendation outcomes is noticeable but still constrained. When the filler size is raised to 5%, the proportion of manipulated items in the Top-10 lists increases to approximately 9–15%. This rise indicates that a greater number of injected profiles directly strengthens the penetration of attacks, resulting in fake items appearing more frequently in recommendation lists. In particular, both MUP-DR and MUP-Sum strategies exceed the 10% threshold, while MUP-NNZ reaches 13–14%, confirming its position as the most vulnerable target selection approach. MUP-UsR, by contrast, remains the most robust, yielding the lowest ratios in both scenarios, with values of around 6% at 1% filler and between 8–11% at 5% filler. Overall, this comparison highlights that filler size is a decisive factor in determining the success of multi-criteria group shilling attacks. While the impact of attacks is relatively modest at 1% filler, at 5% filler nearly one out of every six items in the Top-10 lists is a fake product. These findings underscore the heightened risk posed by higher filler levels, where group shilling attacks become considerably more destructive in multi-criteria recommendation environments.

In summary, the comparative analysis of the two datasets illustrates the behavior of both attack models with respect to varying cluster and filler sizes (can see Figs 3 and 4). For each scenario, the average values of the attack models were calculated and visualized in a comparative manner. According to these results, the performance differences between attack strategies and datasets provide valuable insights into the robustness characteristics of multi-criteria recommender systems.

For the first dataset, YM10, both attack types negatively affected the recommendation accuracy. The $GSAGen_lAvg$ method produced higher AvgHR values, particularly under the $S_1$ scenario, indicating a stronger attack impact due to the more realistic user profile generation. In contrast, under the $S_2$ scenario, the $GSAGen\_Ran$ attack exhibited better performance, suggesting that random profile injection can sometimes align more effectively with user diversity in specific scenarios. Moreover, as the filler size increased, the overall AvgHR values consistently decreased, with the most

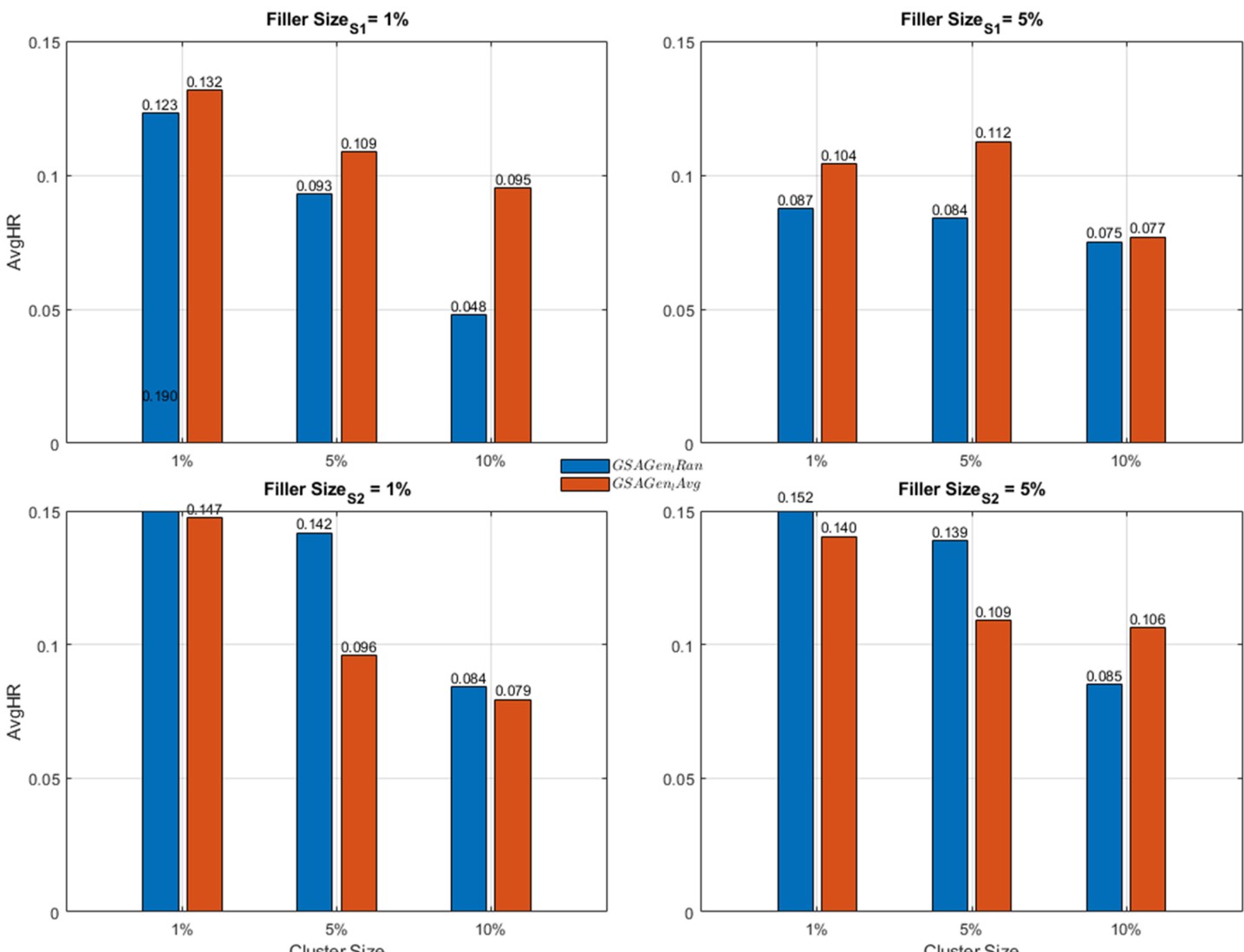

**Fig 3**. **Comparison of** $GSAGen_lRan$ **and** $GSAGen_lAvg$ **for AvgHR Metrics and YM10.**

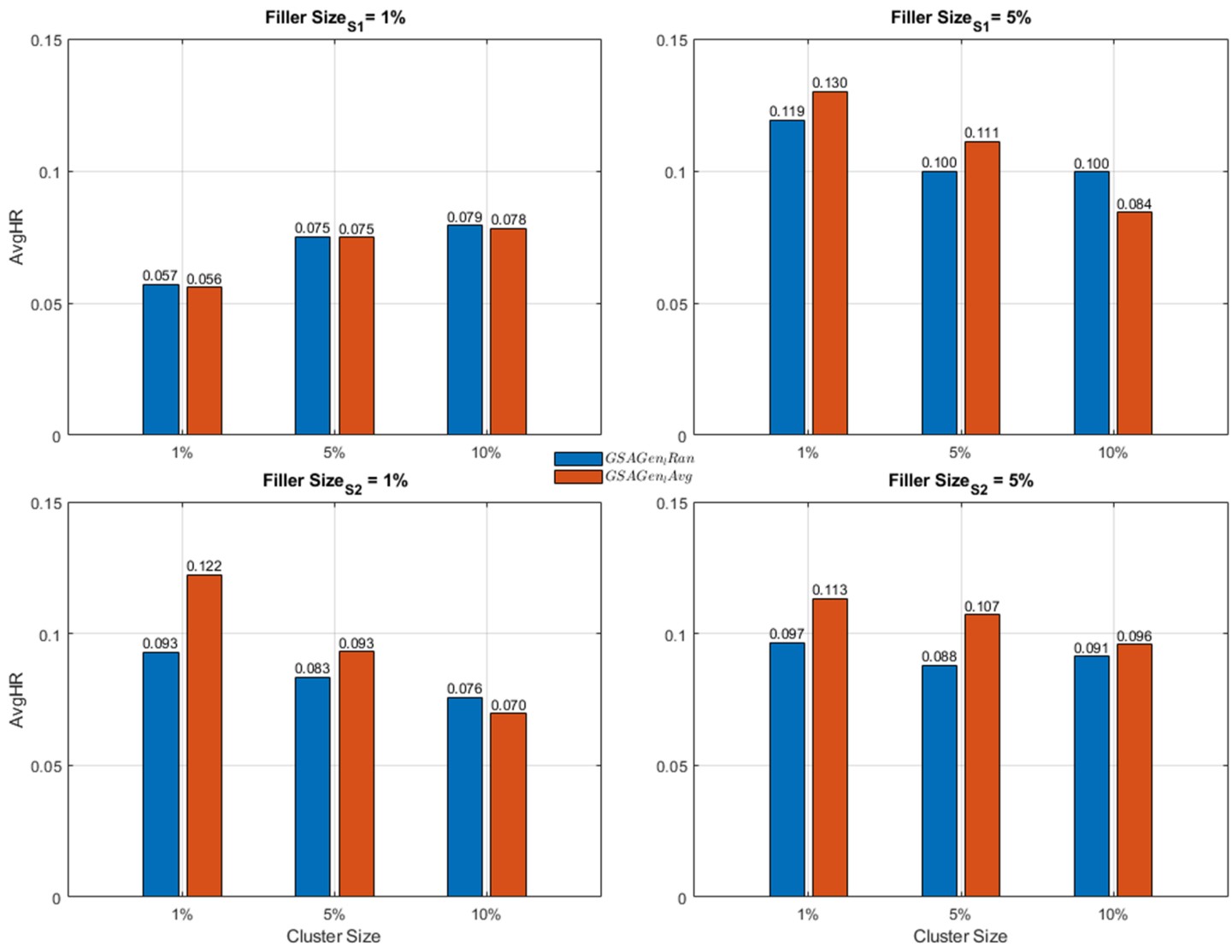

**Fig 4**. **Comparison of** *GSAGen<sub>l</sub>Ran* **and** *GSAGen<sub>l</sub>Avg* **for AvgHR Metrics and YM20.**

significant degradation observed at the 10% filler level. Additionally, increasing the cluster size (from 1% to 10%) resulted in a gradual reduction of the attack impact, implying that larger user groups tend to dilute the influence of injected profiles.

In contrast, the results obtained from the YM20 dataset revealed a relatively weaker attack influence compared to YM10. At lower filler levels, the *GSAGen<sub>l</sub>Avg* method achieved slightly higher AvgHR values; however, the difference between attack types became marginal as the filler size and cluster size increased. The AvgHR trend remained more stable across different cluster sizes, indicating that the greater diversity of users and items in YM20 made the system inherently less susceptible to manipulation. This stability highlights the role of dataset scale and heterogeneity in absorbing or mitigating artificial perturbations.

When comparing both datasets, it becomes evident that the YM10 dataset is more vulnerable to shilling attacks, whereas the YM20 dataset demonstrates higher robustness. The YM10 results exhibit stronger sensitivity to both attack type and scenario, while YM20 presents a more stable and resistant behavior. These findings collectively suggest that

dataset scale, user diversity, and item variety are crucial factors in mitigating the effects of profile injection attacks, ultimately enhancing the resilience of recommender systems operating on larger and more heterogeneous datasets.

An examination of the WRV metric results on the YM20 dataset demonstrates that increasing the filler size significantly strengthens the impact of group shilling attacks (see Tables 8 and 9). Since WRV evaluates not only whether target items appear in recommendation lists but also their ranking positions, higher values indicate that fake items are successfully promoted to the top of the lists, thereby intensifying the attack's effectiveness. At a filler size of 1%, WRV values remain relatively modest; however, certain strategies still manage to push target items toward the top positions (see Table 8). For instance, the MUP-NNZ strategy achieves 1.0507 for Top-10 and 2.9115 for Top-15, suggesting that sparsely rated items can already be placed within upper ranks even under low filler conditions. Similarly, both MUP-DR and MUP-Sum generate noticeable values, indicating that manipulated items appear in favorable positions. In contrast, MUP-UsR yields the lowest WRV scores, meaning that items selected under this strategy tend to remain in lower positions within the lists. Nevertheless, the results confirm that even with only 1% filler, recommendation outputs are not fully resistant to attack.

When the filler size is raised to 5%, the effect of the attacks becomes dramatic according to the Table 9. WRV values rise sharply, with target items being consistently promoted to the very top of the lists. Under $GSAGen_lRan$, MUP-DR reaches 3.4846 at Top-10 and 4.9753 at Top-15, while under $GSAGen_lAvg$ the MUP-NNZ strategy attains 4.8811 for Top-10 and 6.2493 for Top-15, marking the highest observed values. These findings clearly demonstrate that selecting sparsely rated items makes the system particularly vulnerable, as such items can dominate recommendation lists once manipulated. The MUP-Sum strategy also produces high WRV values, including 5.5510 at Top-15, further illustrating the ability of attackers to promote fake items into upper ranks. Although MUP-UsR remains the least effective strategy, its WRV values also increase noticeably under 5% filler, highlighting that even relatively robust approaches lose resistance under stronger attack scenarios. In summary, the WRV analysis reveals that as the filler size increases, group

**Table 8**. WRV results of multi-criteria recommender systems on YM20 dataset for filler size 1%.

| Attack Type | Cluster Size | Target Item Selection | $S_1$ | | | $S_2$ | | |
|---|---|---|---|---|---|---|---|---|
| | | | Top-5 | Top-10 | Top-15 | Top-5 | Top-10 | Top-15 |
| $GSAGen_lRan$ | 1% | MUP-UsR | 0.2134 | 0.5672 | 1.217 | 0.1392 | 0.6542 | 2.4097 |
| | | MUP-DR | 0.2495 | 0.5431 | 1.3098 | 1.7695 | 1.8715 | 2.0569 |
| | | MUP-NNZ | **0.3153** | **1.0507** | **2.9115** | 2.5749 | 2.6865 | 2.9529 |
| | | MUP-Sum | 0.2479 | 0.5544 | 1.3653 | **4.8694** | **5.1026** | **5.4480** |
| | 5% | MUP-UsR | 0.0971 | 0.7696 | 2.1302 | 0.1107 | 0.9200 | 2.4243 |
| | | MUP-DR | 0.1937 | 0.6700 | 2.0190 | 1.3252 | 1.9911 | 3.1492 |
| | | MUP-NNZ | **0.4723** | **1.8684** | **3.6577** | 1.1640 | 1.8901 | 3.0300 |
| | | MUP-Sum | 0.1545 | 0.8937 | 1.9035 | **2.6823** | **3.6299** | **5.5421** |
| | 10% | MUP-UsR | **0.4036** | 1.9905 | **5.3730** | 0.4045 | 2.2654 | **5.2544** |
| | | MUP-DR | 0.3933 | **2.3663** | 5.2527 | **0.4683** | **2.2655** | 5.0461 |
| | | MUP-NNZ | 0.3005 | 1.5058 | 3.5816 | 0.4184 | 1.9195 | 4.4890 |
| | | MUP-Sum | 0.3771 | 1.8621 | 5.0762 | 0.2704 | 1.6472 | 4.6114 |
| $GSAGen_lAvg$ | 1% | MUP-UsR | 0.1625 | 0.5259 | 1.3594 | 0.1512 | 1.0227 | 2.4887 |
| | | MUP-DR | 0.2329 | 0.4818 | 1.5515 | **4.2894** | **4.4003** | **4.6584** |
| | | MUP-NNZ | **0.3810** | **1.1999** | **2.9514** | 2.3902 | 2.6314 | 2.9825 |
| | | MUP-Sum | 0.2201 | 0.7155 | 1.8245 | 2.6087 | 2.7199 | 2.9710 |
| | 5% | MUP-UsR | 0.1517 | 0.8012 | 2.1878 | 0.1298 | 0.7767 | 2.1193 |
| | | MUP-DR | 0.1360 | 1.2131 | 1.9835 | **2.1338** | **2.6881** | **3.8778** |
| | | MUP-NNZ | **0.5473** | **2.1242** | **4.5366** | 1.3010 | 2.2744 | 3.7804 |
| | | MUP-Sum | 0.1839 | 1.0483 | 2.0244 | 1.8631 | 2.4526 | 3.8107 |
| | 10% | MUP-UsR | **0.4334** | 1.9606 | **5.9846** | 0.5885 | **2.4157** | **5.2697** |
| | | MUP-DR | 0.3096 | **2.4853** | 4.8385 | **0.6286** | 2.2697 | 5.2269 |
| | | MUP-NNZ | 0.1577 | 1.3880 | 3.2418 | 0.3257 | 1.8218 | 4.6695 |
| | | MUP-Sum | 0.3922 | 1.5178 | 4.4954 | 0.3193 | 1.9843 | 4.9644 |

**Table 9**. WRV results of multi-criteria recommender systems on YM20 dataset for filler size 5%.

| Attack Type | Cluster Size | Target Item Selection | $S_1$ | | | $S_2$ | | |
|---|---|---|---|---|---|---|---|---|
| | | | Top-5 | Top-10 | Top-15 | Top-5 | Top-10 | Top-15 |
| $GSAGen_lRan$ | 1% | MUP-UsR | 0.1993 | 0.4429 | 1.1333 | 0.1446 | 0.5753 | 2.1232 |
| | | MUP-DR | 4.3920 | 4.5294 | 4.7373 | 2.5740 | 2.6702 | 2.8793 |
| | | MUP-NNZ | 3.5288 | 3.5636 | 3.6482 | 2.3773 | 2.5291 | 2.7810 |
| | | MUP-Sum | **5.1073** | **5.2893** | **5.5933** | **4.3337** | **4.5778** | **4.9523** |
| | 5% | MUP-UsR | 0.1246 | 0.7662 | 2.6827 | 0.1568 | 0.9562 | 2.2116 |
| | | MUP-DR | **3.0170** | **3.4846** | **4.9753** | 1.4988 | 2.3274 | 3.8087 |
| | | MUP-NNZ | 2.3072 | 2.4591 | 3.0272 | 1.6448 | 2.2370 | 3.4634 |
| | | MUP-Sum | 2.1374 | 2.8459 | 4.5788 | **2.7838** | **3.9227** | **5.5510** |
| | 10% | MUP-UsR | 0.4457 | 2.3331 | **5.6031** | 0.4189 | 2.1618 | **5.0608** |
| | | MUP-DR | 0.4627 | **2.3567** | 4.6679 | **0.4716** | 2.0136 | 4.6450 |
| | | MUP-NNZ | 0.2734 | 0.8054 | 2.7291 | 0.3316 | 1.3473 | 4.1037 |
| | | MUP-Sum | **0.8949** | 1.7776 | 5.5658 | 0.3491 | **2.2345** | 5.0339 |
| $GSAGen_lAvg$ | 1% | MUP-UsR | 0.2255 | 0.5641 | 1.3569 | 0.1188 | 0.5639 | 2.3157 |
| | | MUP-DR | 2.6470 | 2.7086 | 2.8482 | 5.1879 | 5.2956 | 5.6566 |
| | | MUP-NNZ | **5.5908** | **5.8142** | **6.1550** | **6.8387** | **6.9943** | **7.2811** |
| | | MUP-Sum | 4.0610 | 4.1993 | 4.4413 | 5.2940 | 5.4651 | 5.7097 |
| | 5% | MUP-UsR | 0.1044 | 0.6791 | 2.3502 | 0.1111 | 0.9066 | 2.4187 |
| | | MUP-DR | 1.1954 | 2.1209 | 2.8742 | 3.2345 | 4.1600 | 5.8367 |
| | | MUP-NNZ | **4.0309** | **4.8811** | **6.2493** | **4.1874** | **4.9980** | **6.4182** |
| | | MUP-Sum | 3.1495 | 4.0033 | 5.2664 | 3.3979 | 4.0461 | 5.2252 |
| | 10% | MUP-UsR | **0.3908** | 1.8589 | 5.6734 | **0.5822** | 2.1707 | **5.5653** |
| | | MUP-DR | 0.3670 | **2.1098** | **5.8676** | 0.4626 | **2.2945** | 5.4614 |
| | | MUP-NNZ | 0.1856 | 1.3520 | 3.4199 | 0.2782 | 1.5483 | 3.9543 |
| | | MUP-Sum | 0.3397 | 1.4815 | 5.1646 | 0.3001 | 2.1676 | 4.7727 |

shilling attacks not only infiltrate recommendation lists but also succeed in positioning target items at the top ranks. At 1% filler, the attacks exert a moderate effect, whereas at 5% filler the upper parts of the Top-10 and Top-15 lists are heavily dominated by fake products. Among the selection strategies, MUP-NNZ consistently emerges as the most vulnerable, while MUP-Sum and MUP-DR also present significant threats. By comparison, MUP-UsR proves to be the most robust, although its resilience diminishes as the filler size grows.

In conclusion, under a filler size of 1% the impact of the attacks remains at a moderate level, whereas at 5% filler the fake items not only infiltrate the recommendation lists but also occupy the top ranks. The MUP-NNZ strategy consistently emerges as the most vulnerable, while both MUP-Sum and MUP-DR demonstrate a high level of threat. By contrast, MUP-UsR yields the lowest values in both scenarios and thus proves to be the most robust strategy against such attacks.

The analysis of the WRV results in Tables 10 and 11 for the YM10 dataset reveals several key insights regarding the effectiveness of the multi-criteria recommender systems attacks. For the filler size of 1%, it is observed that the WRV values for $GSAGen_lAvg$ are generally higher than those for $GSAGen_lRan$. This indicates that $GSAGen_lAvg$ demonstrates superior attack efficiency. Furthermore, as the cluster size increases from 1% to 10%, there is a consistent increase in WRV values, highlighting that a larger cluster size amplifies the attack's impact. When examining the target product selection strategies, MUP-NNZ and MUP-Sum exhibit notably higher WRV values compared to other strategies across both $S_1$ and $S_2$ systems. These findings underscore the effectiveness of these two strategies in ensuring the successful placement of target items in the top-$n$ recommendations. Particularly in the top-15 recommendations, MUP-NNZ and MUP-Sum consistently outperform alternative methods, establishing their dominance in attack success. A comparison of the $S_1$ and $S_2$ systems indicates that WRV values are generally higher for $S_1$, suggesting that the system is more vulnerable to attacks. The trend persists across all cluster sizes and target selection strategies. The findings suggest that system-specific characteristics can influence susceptibility to adversarial attacks.

**Table 10**. WRV results of multi-criteria recommender systems on YM10 dataset for filler size 1%.

| Attack Type | Cluster Size | Target Item Selection | $S_1$ | | | $S_2$ | | |
|---|---|---|---|---|---|---|---|---|
| | | | Top-5 | Top-10 | Top-15 | Top-5 | Top-10 | Top-15 |
| $GSAGen_lRan$ | 1% | MUP-UsR | 0.3153 | 1.7195 | 3.2196 | 0.8193 | 1.9495 | 5.8445 |
| | | MUP-DR | 4.0910 | 4.2964 | 4.5171 | 7.2963 | 7.4231 | 7.6795 |
| | | MUP-NNZ | **5.5406** | **5.6893** | **5.9104** | **10.2565** | **10.5273** | **10.9514** |
| | | MUP-Sum | 2.8052 | 2.8936 | 3.1279 | 5.7711 | 6.0142 | 6.3946 |
| | 5% | MUP-UsR | 0.3673 | 1.1189 | 2.7910 | 0.6943 | 1.6996 | 3.5924 |
| | | MUP-DR | 2.4880 | 3.5063 | 4.8666 | 4.0876 | 4.5258 | 5.5313 |
| | | MUP-NNZ | **3.8171** | **4.4251** | **5.8163** | **6.1338** | **6.6751** | **8.3782** |
| | | MUP-Sum | 1.1580 | 1.4790 | 2.2481 | 4.3529 | 5.5999 | 7.1460 |
| | 10% | MUP-UsR | **0.5501** | 1.4365 | 3.7982 | 0.2910 | 1.6633 | **8.0462** |
| | | MUP-DR | 0.3029 | **2.0945** | 3.4847 | **0.6630** | 1.9229 | 6.4735 |
| | | MUP-NNZ | 0.4110 | 1.3575 | **3.8181** | 0.5426 | **3.8118** | 5.3523 |
| | | MUP-Sum | 0.5399 | 1.7168 | 3.5006 | 0.2149 | 2.5185 | 5.4689 |
| $GSAGen_lAvg$ | 1% | MUP-UsR | 0.3631 | 1.1524 | 3.9449 | 0.2895 | 1.3986 | 4.7174 |
| | | MUP-DR | 4.4652 | 4.5372 | 4.8252 | 5.6291 | 5.7576 | 5.9286 |
| | | MUP-NNZ | 3.1940 | 3.2975 | 3.4059 | **8.8353** | **9.1212** | **9.5483** |
| | | MUP-Sum | **5.5992** | **5.6894** | **5.9125** | 2.0298 | 2.1844 | 2.5318 |
| | 5% | MUP-UsR | 0.4949 | 1.5522 | 3.2908 | 0.4413 | 1.1640 | 4.0772 |
| | | MUP-DR | 2.8493 | 3.1574 | 4.1881 | 2.6824 | 3.1773 | 3.7041 |
| | | MUP-NNZ | 1.6004 | 1.9751 | 3.0264 | **5.1849** | **6.3137** | **8.2301** |
| | | MUP-Sum | **3.7305** | **4.3290** | **5.4511** | 1.1721 | 1.8789 | 3.2115 |
| | 10% | MUP-UsR | 0.1950 | **2.3612** | 3.5769 | **0.4944** | 2.0613 | 5.5020 |
| | | MUP-DR | **0.4968** | 1.4499 | **4.2911** | 0.2320 | **3.6887** | **7.4108** |
| | | MUP-NNZ | 0.2774 | 1.0763 | 3.8373 | 0.2100 | 3.5228 | 4.0650 |
| | | MUP-Sum | 0.2829 | 1.5746 | 3.0029 | 0.2493 | 2.4428 | 6.0987 |

**Table 11**. WRV results of multi-criteria recommender systems on YM10 dataset for filler size 5%.

| Attack Type | Cluster Size | Target Item Selection | $S_1$ | | | $S_2$ | | |
|---|---|---|---|---|---|---|---|---|
| | | | Top-5 | Top-10 | Top-15 | Top-5 | Top-10 | Top-15 |
| $GSAGen_lRan$ | 1% | MUP-UsR | 0.1466 | 1.0985 | **3.7907** | 0.3688 | 2.1451 | 5.1966 |
| | | MUP-DR | 1.7014 | 1.7893 | 1.9663 | **8.8771** | **9.0072** | **9.4277** |
| | | MUP-NNZ | **2.8781** | **2.9188** | 3.1706 | 5.2753 | 5.3033 | 5.3501 |
| | | MUP-Sum | 2.3780 | 2.4742 | 2.6824 | 1.0970 | 1.2273 | 1.4930 |
| | 5% | MUP-UsR | 0.3150 | 1.2093 | 2.8158 | 0.5116 | 2.5664 | 4.6802 |
| | | MUP-DR | 1.0667 | 1.5311 | 2.7152 | **5.7030** | **6.9591** | **7.8785** |
| | | MUP-NNZ | **1.7174** | **2.0098** | **3.4439** | 2.7752 | 3.0701 | 4.2095 |
| | | MUP-Sum | 1.2016 | 1.9161 | 3.0940 | 0.7169 | 1.4490 | 2.5213 |
| | 10% | MUP-UsR | **0.7099** | 1.6067 | 4.2851 | **0.8515** | **2.5666** | **5.9402** |
| | | MUP-DR | 0.7009 | 1.9264 | 4.3387 | 0.4443 | 1.6819 | 5.0995 |
| | | MUP-NNZ | 0.5487 | 1.4235 | **4.6778** | 0.1265 | 0.9664 | 3.3508 |
| | | MUP-Sum | 0.5235 | **2.2558** | 3.7554 | 0.2186 | 2.2310 | 4.1088 |
| $GSAGen_lAvg$ | 1% | MUP-UsR | 0.2367 | 1.4820 | 3.8281 | 0.4113 | 2.4551 | 6.0410 |
| | | MUP-DR | **4.2201** | **4.3429** | **4.6188** | 4.8323 | 5.0133 | 5.2465 |
| | | MUP-NNZ | 3.9713 | 4.1382 | 4.5530 | **5.2698** | **5.3544** | **5.5443** |
| | | MUP-Sum | 2.6014 | 2.7651 | 3.0651 | 2.0570 | 2.1186 | 2.2914 |
| | 5% | MUP-UsR | 0.2625 | 0.8415 | 2.3839 | 1.0320 | **4.3152** | **5.6618** |
| | | MUP-DR | **2.9230** | **3.6774** | **5.7720** | 3.0267 | 3.7728 | 4.8721 |
| | | MUP-NNZ | 2.1369 | 3.5504 | 4.8845 | **3.0079** | 3.1666 | 5.2762 |
| | | MUP-Sum | 1.6140 | 2.4173 | 3.8091 | 1.3620 | 2.2402 | 3.0688 |
| | 10% | MUP-UsR | 0.1645 | **2.0502** | 3.1321 | **1.1332** | 1.2930 | 3.7525 |
| | | MUP-DR | **0.5704** | 1.9460 | 4.4082 | 1.0097 | **3.1527** | **5.4023** |
| | | MUP-NNZ | 0.3655 | 1.5982 | **5.1857** | 0.4093 | 1.4992 | 2.2190 |
| | | MUP-Sum | 0.3471 | 1.5653 | 4.5367 | 0.3920 | 2.0223 | 4.7813 |

For the filler size of 5% in Table 11, a significant increase in WRV values is observed compared to the 1% filler size in Table 10. This emphasizes the critical role of filler size in enhancing attack effectiveness. The impact of increased filler size is particularly evident in strategies like MUP-NNZ, where WRV values show remarkable improvements. Across both $GSAGen_lRan$ and $GSAGen_lAvg$, the trend of higher WRV values with larger cluster sizes is sustained, further reinforcing the role of cluster size in determining attack efficacy. Notably, $GSAGen_lRan$ performs well with smaller cluster sizes and lower filler ratios, while $GSAGen_lAvg$ demonstrates superior performance with larger cluster sizes and higher filler ratios. The distinction highlights the varying strengths of these attack types under different configurations. Moreover, the effectiveness of MUP-NNZ and MUP-Sum remains evident, with these strategies achieving the highest WRV values across both $S_1$ and $S_2$ systems under varying filler sizes and cluster configurations.

In summary, the results for the YM10 dataset corroborate the observations made on the YM20 dataset. Larger cluster sizes and higher filler ratios enhance attack effectiveness, while MUP-NNZ and MUP-Sum consistently emerge as the most successful target selection strategies. The $GSAGen_lAvg$ attack exhibits superior performance in most scenarios, particularly with larger configurations, while $S_1$ consistently shows greater vulnerability compared to $S_2$.

The Table 12 provides a comprehensive analysis of the AvgHR results for different attack configurations, highlighting the effectiveness of various attack types, cluster sizes, and aggregation strategies across $S_1$. The AvgHR metric is a critical indicator of the success of adversarial attacks in increasing the likelihood of targeted items being recommended and the metric reveals that higher ratios indicate a greater vulnerability of the system to shilling attacks, while lower ratios correspond to stronger robustness. Under the $GSAGen_lRan$ attack, the most vulnerable aggregation techniques are MUL, SC, and, to some extent, AU. For instance, MUL produces values of 0.1223 at Top-10 and 0.2202 at Top-15 with a cluster size of 0.1, while SC reaches 0.0843 at Top-10 and 0.0978 at Top-15 with a cluster size of 0.05. These results demonstrate that such techniques allow fake items to infiltrate the recommendation lists effectively. In contrast, AwM and MP exhibit very low AvgHR values, remaining close to zero in most scenarios, thereby proving to be the most robust approaches against $GSAGen_lRan$. A similar pattern can be observed under the $GSAGen_lAvg$ attack. MUL and SC again appear as the most fragile strategies, yielding the highest AvgHR values. For example, MUL records 0.1732 at Top-15 with a cluster size of 0.05 and 0.1715 at Top-15 with a cluster size of 0.1, whereas SC produces 0.1467 at Top-15 with a cluster size of 0.05. These results indicate that both techniques are highly susceptible to manipulation. On the other hand, AwM consistently remains the most robust method, producing either zero or very low values across all scenarios. Furthermore, MP and, to some extent, LM also show relatively low AvgHR values, suggesting better resistance to attacks.

In summary, across both attack types, MUL, SC, and AU emerge as the most vulnerable aggregation techniques, while AwM, MP, and partially LM stand out as the most robust methods. This comparison highlights the importance of selecting appropriate aggregation strategies to mitigate the impact of group shilling attacks in multi-criteria recommender systems.

The results in Table 13 show that the AvgHR values remain generally low under the S2 scenario, indicating that the attacks significantly reduce the hit ratio of the group recommender systems. As the cluster size increases, a partial improvement in AvgHR values can be observed, with more noticeable gains in the $GSAGen_lAvg$ case when the cluster size is 0.1. In contrast, the $GSAGen_lRan$ scenario consistently yields the weakest performance. Among the aggregation techniques, AU provides moderate results and performs relatively better under $GSAGen_lAvg$ with larger clusters, while MUL behaves similarly to AU but sometimes aligns with SC as one of the better methods. SC performs poorly with smaller cluster sizes but achieves considerable improvements in $GSAGen_lAvg$ at 0.1 cluster size. MP remains mostly average, contributing more at medium and large cluster sizes. MRP stands out as the most robust technique, showing the highest AvgHR values under $GSAGen_lAvg$ with cluster size 0.1, which suggests its higher resistance to attacks. Moreover, the AvgHR values consistently increase from Top-5 to Top-15 recommendations, revealing that the systems are more vulnerable to attacks with smaller recommendation lists, but partially recover as the list length increases. Overall, $GSAGen_lRan$

**Table 12. AvgHR results of multi-criteria recommender systems on YM20 dataset for each aggregation techniques and filler size 1% (S1).**

| Attack Type | Cluster Size | Aggregation Techniques | S1 MUP-Usr | | | MUP-DR | | | MUP-NNZ | | | MUP-Sum | | |
|---|---|---|---|---|---|---|---|---|---|---|---|---|---|---|
| | | | Top-5 | Top-10 | Top-15 | Top-5 | Top-10 | Top-15 | Top-5 | Top-10 | Top-15 | Top-5 | Top-10 | Top-15 |
| GSAGen_lRan | 0.01 | Avg | 0.025 | 0.05 | 0.0583 | 0.0138 | 0.0325 | 0.0675 | 0.035 | 0.07 | 0.1233 | 0.025 | 0.045 | 0.0583 |
| | | AU | 0.035 | 0.055 | 0.0567 | 0.04 | 0.0456 | 0.0708 | 0.1 | 0.095 | 0.1367 | 0.025 | 0.0375 | 0.07 |
| | | MUL | 0 | 0.0225 | 0.1017 | 0.03 | 0.0525 | 0.1063 | 0.025 | 0.08 | 0.1267 | 0.075 | 0.1 | 0.1267 |
| | | SC | 0 | 0 | 0 | 0 | 0 | 0 | 0 | 0 | 0 | 0 | 0 | 0 |
| | | AwM | 0 | 0 | 0 | 0 | 0 | 0 | 0 | 0 | 0 | 0 | 0 | 0 |
| | | Avg | 0.025 | 0 | 0 | 0.0075 | 0 | 0 | 0.025 | 0.0025 | 0.02 | 0.025 | 0 | 0 |
| | | MP | 0.03 | 0.0375 | 0.0633 | 0.0138 | 0.0325 | 0.0713 | 0.045 | 0.0525 | 0.105 | 0.025 | 0.0325 | 0.0733 |
| | | LM | 0.025 | 0.05 | 0.1 | 0.0225 | 0.0438 | 0.0933 | 0.035 | 0.0775 | 0.11 | 0.055 | 0.0475 | 0.1217 |
| | | MRP | 0.03 | 0.05 | 0.0633 | 0.0188 | 0.0338 | 0.0888 | 0.075 | 0.0725 | 0.1317 | 0.03 | 0.0475 | 0.0717 |
| | 0.05 | Avg | 0.0095 | 0.0576 | 0.086 | 0.0229 | 0.0476 | 0.0629 | 0.0695 | 0.1214 | 0.1695 | 0.0133 | 0.0395 | 0.0819 |
| | | AU | 0.0267 | 0.071 | 0.106 | 0.0276 | 0.0748 | 0.1111 | 0.0857 | 0.1419 | 0.167 | 0.0295 | 0.0995 | 0.0927 |
| | | MUL | 0.0105 | 0.0695 | 0.0908 | 0.0248 | 0.05 | 0.0813 | 0.0743 | 0.0852 | 0.1698 | 0.0238 | 0.0543 | 0.0921 |
| | | SC | 0.0219 | 0.0843 | 0.0978 | 0.0219 | 0.1138 | 0.0971 | 0.0124 | 0.089 | 0.1016 | 0.0133 | 0.1071 | 0.1124 |
| | | AwM | 0 | 0 | 0 | 0 | 0 | 0 | 0 | 0 | 0 | 0 | 0 | 0 |
| | | Avg | 0.0057 | 0 | 0 | 0.0057 | 0 | 0 | 0.0524 | 0.05 | 0.0568 | 0.0067 | 0 | 0 |
| | | MP | 0.0048 | 0.0129 | 0.0197 | 0.0048 | 0.0071 | 0.0054 | 0.0105 | 0.011 | 0.0092 | 0.0048 | 0.0057 | 0.0127 |
| | | LM | 0.0124 | 0.0895 | 0.0905 | 0.0343 | 0.0581 | 0.1095 | 0.0267 | 0.061 | 0.1403 | 0.0238 | 0.0714 | 0.0876 |
| | | MRP | 0.0076 | 0.041 | 0.1006 | 0.041 | 0.0462 | 0.1295 | 0.0524 | 0.1305 | 0.1394 | 0.019 | 0.0576 | 0.0987 |
| | 0.1 | Avg | 0.1191 | 0.1272 | 0.2335 | 0.0879 | 0.1486 | 0.2087 | 0.1516 | 0.1449 | 0.1443 | 0.0935 | 0.1479 | 0.2048 |
| | | AU | 0.0219 | 0.0798 | 0.1184 | 0.046 | 0.1198 | 0.1439 | 0.0665 | 0.1677 | 0.1361 | 0.0284 | 0.0812 | 0.131 |
| | | MUL | 0.0847 | 0.1223 | 0.2299 | 0.0526 | 0.1474 | 0.2202 | 0.1284 | 0.1295 | 0.1262 | 0.0819 | 0.13 | 0.2102 |
| | | SC | 0.0153 | 0.1005 | 0.0881 | 0.0447 | 0.0916 | 0.1253 | 0.0079 | 0.1014 | 0.0975 | 0.026 | 0.0749 | 0.1152 |
| | | AwM | 0 | 0 | 0 | 0 | 0 | 0 | 0 | 0 | 0 | 0 | 0 | 0 |
| | | Avg | 0.0028 | 0 | 0 | 0.0028 | 0 | 0 | 0.0651 | 0.06 | 0.04 | 0.0028 | 0 | 0 |
| | | MP | 0.0028 | 0 | 0 | 0.0023 | 0 | 0 | 0.0033 | 0 | 0 | 0.0023 | 0 | 0 |
| | | LM | 0.0819 | 0.0937 | 0.1833 | 0.1135 | 0.1391 | 0.1422 | 0.0549 | 0.1314 | 0.1338 | 0.0823 | 0.1021 | 0.1363 |
| | | MRP | 0.0321 | 0.1184 | 0.1873 | 0.0181 | 0.1223 | 0.1693 | 0.0781 | 0.1384 | 0.142 | 0.0409 | 0.1023 | 0.1831 |
| GSAGen_lAvg | 0.01 | Avg | 0.03 | 0.0425 | 0.0433 | 0.0088 | 0.0238 | 0.0633 | 0.065 | 0.0775 | 0.1333 | 0.03 | 0.03 | 0.0883 |
| | | AU | 0.025 | 0.045 | 0.0883 | 0.0238 | 0.025 | 0.0588 | 0.05 | 0.08 | 0.1517 | 0.045 | 0.06 | 0.0733 |
| | | MUL | 0.015 | 0.0625 | 0.0917 | 0.0125 | 0.06 | 0.105 | 0.06 | 0.115 | 0.1183 | 0.03 | 0.075 | 0.1317 |
| | | SC | 0 | 0 | 0 | 0 | 0 | 0 | 0 | 0 | 0 | 0 | 0 | 0 |
| | | AwM | 0 | 0 | 0 | 0 | 0 | 0 | 0 | 0 | 0 | 0 | 0 | 0 |
| | | Avg | 0.025 | 0 | 0 | 0.0075 | 0 | 0 | 0.025 | 0.0025 | 0.02 | 0.025 | 0 | 0 |
| | | MP | 0.025 | 0.04 | 0.0533 | 0.0163 | 0.0269 | 0.0729 | 0.05 | 0.0625 | 0.135 | 0.025 | 0.06 | 0.0783 |
| | | LM | 0.005 | 0.0375 | 0.07 | 0.0138 | 0.045 | 0.07 | 0.035 | 0.12 | 0.105 | 0.005 | 0.0725 | 0.15 |
| | | MRP | 0.005 | 0.0575 | 0.0917 | 0.0138 | 0.0463 | 0.0908 | 0.04 | 0.0525 | 0.12 | 0.05 | 0.0925 | 0.0883 |
| | 0.05 | Avg | 0.0086 | 0.0295 | 0.0778 | 0.02 | 0.0662 | 0.0616 | 0.0714 | 0.1224 | 0.1683 | 0.0181 | 0.0638 | 0.0765 |
| | | AU | 0.04 | 0.0624 | 0.1384 | 0.0238 | 0.0824 | 0.1317 | 0.0343 | 0.1143 | 0.1711 | 0.0171 | 0.0919 | 0.1121 |
| | | MUL | 0.0105 | 0.0424 | 0.0803 | 0.019 | 0.0662 | 0.0752 | 0.0952 | 0.1095 | 0.1752 | 0.0267 | 0.0738 | 0.101 |
| | | SC | 0.0619 | 0.0795 | 0.1467 | 0.0229 | 0.0895 | 0.1235 | 0.0114 | 0.0629 | 0.1 | 0.0238 | 0.1086 | 0.1229 |
| | | AwM | 0 | 0 | 0 | 0 | 0 | 0 | 0 | 0 | 0 | 0 | 0 | 0 |
| | | Avg | 0.0057 | 0 | 0 | 0.0057 | 0 | 0 | 0.0533 | 0.0567 | 0.0635 | 0.0067 | 0 | 0 |
| | | MP | 0.0048 | 0.009 | 0.0346 | 0.0048 | 0.001 | 0.0216 | 0.0048 | 0.0129 | 0.0083 | 0.0048 | 0.0033 | 0.0289 |
| | | LM | 0.0219 | 0.0724 | 0.0737 | 0.0257 | 0.0719 | 0.0876 | 0.0648 | 0.0776 | 0.1305 | 0.0352 | 0.0576 | 0.0835 |
| | | MRP | 0.0238 | 0.051 | 0.1025 | 0.019 | 0.0529 | 0.0933 | 0.0505 | 0.1257 | 0.1346 | 0.0143 | 0.0252 | 0.0848 |
| | 0.1 | Avg | 0.0451 | 0.1463 | 0.2234 | 0.047 | 0.1421 | 0.214 | 0.0726 | 0.1314 | 0.1381 | 0.074 | 0.0979 | 0.1753 |
| | | AU | 0.0623 | 0.0853 | 0.1402 | 0.0586 | 0.0965 | 0.142 | 0.0619 | 0.0947 | 0.1384 | 0.0851 | 0.1221 | 0.1116 |
| | | MUL | 0.0702 | 0.1402 | 0.1732 | 0.0428 | 0.1402 | 0.1864 | 0.0674 | 0.1277 | 0.1366 | 0.0707 | 0.09 | 0.165 |
| | | SC | 0.0902 | 0.0572 | 0.1034 | 0.027 | 0.0816 | 0.1254 | 0.0191 | 0.0642 | 0.1299 | 0.0349 | 0.106 | 0.1161 |
| | | AwM | 0 | 0 | 0 | 0 | 0 | 0 | 0 | 0 | 0 | 0 | 0 | 0 |
| | | Avg | 0.0028 | 0 | 0 | 0.0028 | 0 | 0 | 0.0535 | 0.06 | 0.04 | 0.0028 | 0 | 0 |
| | | MP | 0.0023 | 0 | 0.0005 | 0.0023 | 0 | 0 | 0.0023 | 0 | 0.0003 | 0.0023 | 0 | 0 |
| | | LM | 0.0702 | 0.1358 | 0.1811 | 0.0312 | 0.0865 | 0.1229 | 0.0474 | 0.0707 | 0.1391 | 0.0595 | 0.0688 | 0.1198 |
| | | MRP | 0.0702 | 0.0984 | 0.1715 | 0.0907 | 0.1547 | 0.1899 | 0.0744 | 0.1223 | 0.16 | 0.0786 | 0.114 | 0.1575 |

**Table 13.** AvgHR results of multi-criteria recommender systems on YM20 dataset for each aggregation techniques and filler size 1% (S2).

| Attack Type | Cluster Size | Aggregation Techniques | S2 | | | | | | | | | | | |
| --- | --- | --- | --- | --- | --- | --- | --- | --- | --- | --- | --- | --- | --- | --- |
| | | | MUP-Usr | | | MUP-DR | | | MUP-NNZ | | | MUP-Sum | | |
| | | | Top-5 | Top-10 | Top-15 | Top-5 | Top-10 | Top-15 | Top-5 | Top-10 | Top-15 | Top-5 | Top-10 | Top-15 |
| $GSAGen_lRan$ | 0.01 | Avg | 0.0113 | 0.0363 | 0.0646 | 0.0605 | 0.0612 | 0.0667 | 0.0485 | 0.0536 | 0.0586 | 0.1321 | 0.137 | 0.1433 |
| | | AU | 0.0375 | 0.0406 | 0.0713 | 0.0797 | 0.0915 | 0.0922 | 0.0853 | 0.0902 | 0.0911 | 0.086 | 0.0916 | 0.0952 |
| | | MUL | 0.03 | 0.0638 | 0.1096 | 0.0961 | 0.0968 | 0.0997 | 0.0594 | 0.0645 | 0.0695 | 0.1501 | 0.1645 | 0.162 |
| | | SC | 0 | 0 | 0 | 0.0883 | 0.1003 | 0.1034 | 0.033 | 0.0359 | 0.0379 | 0.0646 | 0.0668 | 0.0729 |
| | | AwM | 0 | 0 | 0 | 0 | 0 | 0 | 0 | 0 | 0 | 0 | 0 | 0 |
| | | Avg | 0 | 0.0006 | 0 | 0 | 0 | 0 | 0.0019 | 0.0007 | 0.0011 | 0 | 0 | 0 |
| | | MP | 0.015 | 0.0263 | 0.0767 | 0.1225 | 0.1229 | 0.1234 | 0.1147 | 0.1147 | 0.1163 | 0.1615 | 0.1749 | 0.1771 |
| | | LM | 0.0213 | 0.0531 | 0.0821 | 0.0729 | 0.0754 | 0.0753 | 0.0899 | 0.0854 | 0.0886 | 0.1628 | 0.1706 | 0.1781 |
| | | MRP | 0.0075 | 0.0375 | 0.0946 | 0.054 | 0.0621 | 0.0668 | 0.0798 | 0.0814 | 0.0883 | 0.0345 | 0.0406 | 0.0431 |
| | 0.05 | Avg | 0.0088 | 0.0264 | 0.0553 | 0.0497 | 0.0696 | 0.0913 | 0.0282 | 0.0616 | 0.0842 | 0.0908 | 0.1165 | 0.1392 |
| | | AU | 0.0279 | 0.0795 | 0.1121 | 0.0898 | 0.1256 | 0.1531 | 0.0696 | 0.0907 | 0.1044 | 0.0605 | 0.0897 | 0.1184 |
| | | MUL | 0.0057 | 0.0427 | 0.0834 | 0.0753 | 0.0865 | 0.0897 | 0.0261 | 0.0596 | 0.0805 | 0.1126 | 0.1576 | 0.1708 |
| | | SC | 0.0474 | 0.0814 | 0.1104 | 0.09 | 0.1251 | 0.144 | 0.0251 | 0.0536 | 0.0698 | 0.0462 | 0.0744 | 0.0969 |
| | | AwM | 0 | 0 | 0 | 0 | 0 | 0 | 0 | 0 | 0 | 0 | 0 | 0 |
| | | Avg | 0 | 0 | 0 | 0 | 0 | 0 | 0.0008 | 0.0006 | 0.0005 | 0 | 0 | 0 |
| | | MP | 0.0157 | 0.0776 | 0.106 | 0.0666 | 0.0731 | 0.0772 | 0.0977 | 0.0926 | 0.1008 | 0.1297 | 0.166 | 0.1936 |
| | | LM | 0.0071 | 0.0618 | 0.0955 | 0.0639 | 0.0689 | 0.0786 | 0.0901 | 0.0862 | 0.0958 | 0.0958 | 0.1207 | 0.1601 |
| | | MRP | 0.0224 | 0.0468 | 0.0797 | 0.0611 | 0.1015 | 0.1168 | 0.0541 | 0.0734 | 0.0975 | 0.0146 | 0.0324 | 0.0648 |
| | 0.1 | Avg | 0.0691 | 0.1102 | 0.14 | 0.0657 | 0.1319 | 0.1458 | 0.0293 | 0.0777 | 0.1269 | 0.0301 | 0.0919 | 0.1245 |
| | | AU | 0.0458 | 0.0958 | 0.1369 | 0.0638 | 0.092 | 0.1356 | 0.0472 | 0.0756 | 0.1206 | 0.0333 | 0.0848 | 0.1187 |
| | | MUL | 0.0641 | 0.1071 | 0.1375 | 0.0558 | 0.1283 | 0.139 | 0.0271 | 0.0791 | 0.1443 | 0.041 | 0.0901 | 0.1259 |
| | | SC | 0.0444 | 0.0974 | 0.1288 | 0.0322 | 0.0847 | 0.1398 | 0.0393 | 0.0727 | 0.115 | 0.028 | 0.0681 | 0.1245 |
| | | AwM | 0 | 0 | 0 | 0 | 0 | 0 | 0 | 0 | 0 | 0 | 0 | 0 |
| | | Avg | 0 | 0 | 0 | 0 | 0 | 0.0015 | 0.015 | 0.0003 | 0.0017 | 0 | 0 | 0 |
| | | MP | 0.0233 | 0.0746 | 0.1324 | 0.0348 | 0.0863 | 0.1202 | 0.06 | 0.091 | 0.1375 | 0.0337 | 0.0705 | 0.1307 |
| | | LM | 0.0421 | 0.0863 | 0.1152 | 0.059 | 0.1072 | 0.1364 | 0.0364 | 0.0907 | 0.1468 | 0.0285 | 0.0906 | 0.1282 |
| | | MRP | 0.0535 | 0.1127 | 0.1412 | 0.0505 | 0.1033 | 0.1354 | 0.0359 | 0.0955 | 0.1171 | 0.049 | 0.0744 | 0.128 |
| $GSAGen_lAvg$ | 0.01 | Avg | 0.0163 | 0.0356 | 0.0588 | 0.2202 | 0.2339 | 0.2356 | 0.1205 | 0.126 | 0.1345 | 0.0628 | 0.0668 | 0.0726 |
| | | AU | 0.0288 | 0.0444 | 0.0833 | 0.0622 | 0.0659 | 0.0728 | 0.1096 | 0.115 | 0.1158 | 0.0322 | 0.0412 | 0.0414 |
| | | MUL | 0.0113 | 0.0525 | 0.0954 | 0.2155 | 0.2246 | 0.2321 | 0.0595 | 0.067 | 0.0737 | 0.0752 | 0.0759 | 0.0774 |
| | | SC | 0 | 0 | 0 | 0.0591 | 0.0641 | 0.0687 | 0.066 | 0.0715 | 0.0708 | 0.0976 | 0.1025 | 0.1044 |
| | | AwM | 0 | 0 | 0 | 0 | 0 | 0 | 0 | 0 | 0 | 0 | 0 | 0 |
| | | Avg | 0 | 0.0006 | 0 | 0 | 0 | 0 | 0.0014 | 0 | 0.0002 | 0 | 0 | 0 |
| | | MP | 0.02 | 0.0644 | 0.0933 | 0.1225 | 0.1225 | 0.1288 | 0.1239 | 0.1238 | 0.126 | 0.0264 | 0.0289 | 0.0303 |
| | | LM | 0.02 | 0.0438 | 0.0858 | 0.1887 | 0.1957 | 0.1938 | 0.0955 | 0.1003 | 0.1063 | 0.0646 | 0.0685 | 0.0726 |
| | | MRP | 0.025 | 0.065 | 0.0979 | 0.1927 | 0.1994 | 0.2049 | 0.2269 | 0.2336 | 0.2359 | 0.0889 | 0.0908 | 0.095 |
| | 0.05 | Avg | 0.0095 | 0.033 | 0.064 | 0.1273 | 0.1464 | 0.1857 | 0.0887 | 0.1094 | 0.1457 | 0.0476 | 0.0658 | 0.1021 |
| | | AU | 0.0157 | 0.05 | 0.0895 | 0.0486 | 0.0658 | 0.0836 | 0.0865 | 0.1087 | 0.1196 | 0.036 | 0.0809 | 0.1046 |
| | | MUL | 0.0081 | 0.041 | 0.0938 | 0.1395 | 0.1455 | 0.1938 | 0.048 | 0.0748 | 0.1052 | 0.0474 | 0.0548 | 0.0611 |
| | | SC | 0.0286 | 0.0605 | 0.0914 | 0.0467 | 0.0684 | 0.0976 | 0.0484 | 0.0774 | 0.084 | 0.0525 | 0.0859 | 0.1072 |
| | | AwM | 0 | 0 | 0 | 0 | 0 | 0 | 0 | 0 | 0 | 0 | 0 | 0 |
| | | Avg | 0 | 0 | 0 | 0 | 0 | 0.0005 | 0.0049 | 0.0009 | 0.0008 | 0 | 0 | 0 |
| | | MP | 0.0502 | 0.0613 | 0.1089 | 0.0802 | 0.1003 | 0.1137 | 0.0736 | 0.092 | 0.105 | 0.0206 | 0.0391 | 0.056 |
| | | LM | 0.0229 | 0.0695 | 0.104 | 0.1521 | 0.1833 | 0.1746 | 0.0741 | 0.0908 | 0.1155 | 0.0436 | 0.0586 | 0.0783 |
| | | MRP | 0.0269 | 0.0401 | 0.0544 | 0.0958 | 0.118 | 0.1455 | 0.1454 | 0.1821 | 0.1886 | 0.0671 | 0.0895 | 0.1062 |
| | 0.1 | Avg | 0.1123 | 0.1215 | 0.1658 | 0.0972 | 0.1197 | 0.1593 | 0.0548 | 0.0785 | 0.116 | 0.0513 | 0.0901 | 0.1398 |
| | | AU | 0.0641 | 0.1228 | 0.1163 | 0.0647 | 0.108 | 0.1357 | 0.0395 | 0.0739 | 0.124 | 0.0481 | 0.0934 | 0.131 |
| | | MUL | 0.1137 | 0.1216 | 0.1602 | 0.1006 | 0.1174 | 0.148 | 0.0476 | 0.0733 | 0.1213 | 0.0441 | 0.0865 | 0.1424 |
| | | SC | 0.0409 | 0.0972 | 0.1252 | 0.0462 | 0.089 | 0.1328 | 0.0233 | 0.0687 | 0.1205 | 0.0256 | 0.0884 | 0.1314 |
| | | AwM | 0 | 0 | 0 | 0 | 0 | 0 | 0 | 0 | 0 | 0 | 0 | 0 |
| | | Avg | 0.0002 | 0 | 0 | 0 | 0 | 0.0014 | 0 | 0.0042 | 0.0012 | 0 | 0 | 0 |
| | | MP | 0.0621 | 0.0808 | 0.117 | 0.0263 | 0.0737 | 0.1145 | 0.0442 | 0.1003 | 0.1229 | 0.0258 | 0.094 | 0.1401 |
| | | LM | 0.0633 | 0.0886 | 0.1591 | 0.0729 | 0.0862 | 0.139 | 0.0279 | 0.0874 | 0.1332 | 0.0313 | 0.0963 | 0.1306 |
| | | MRP | 0.0579 | 0.1094 | 0.1253 | 0.0795 | 0.1086 | 0.1458 | 0.0442 | 0.0874 | 0.1247 | 0.0326 | 0.094 | 0.1339 |

with small clusters is the weakest case, while *GSAGen₁Avg* with cluster size 0.1 yields the strongest results, where MRP, SC, and MUL emerge as the most effective techniques, with MRP being the most stable one.

The results in Tables 14 and 15 show that when the filler size increases to 5%, the AvgHR values rise considerably compared to the 1% filler condition. In particular, the hit ratio values at Top-15 reach around 20–25%, indicating that injected fake profiles have a stronger ability to push target items into recommendation lists. Under the *GSAGen₁Ran* scenario, MRP achieves up to 22–23% AvgHR at Top-15, while AU and MUL also produce values in the range of 18–21%. Although SC remains weak for small cluster sizes, its performance improves with larger clusters, reaching nearly 17%. The *GSAGen₁Avg* scenario reveals an even stronger attack effect: for cluster size 0.1, MRP reaches around 23–25% AvgHR at Top-15, while MUL and AU achieve 19–22%, and SC rises to 16–18%. Among the aggregation strategies, MRP is the most vulnerable, consistently producing the highest AvgHR values under both scenarios. AU and MUL also show significant susceptibility, whereas MP and LM remain relatively less affected with values mostly in the range of 10–15%. Overall, the increase in filler size amplifies the impact of attacks, with *GSAGen₁Avg* combined with larger cluster sizes producing the strongest effects on the recommendation lists.

In the last dataset, YM10, we can see the AvgHR results of performed experimental in Tables 16–19. According to the results in Tables 16 and 17 , increasing the cluster size consistently enhances AvgHR values across both $S_1$ and $S_2$, with larger clusters proving more effective in influencing recommendations. The *GSAGen₁Avg* attack outperforms *GSAGen₁Ran* in all configurations, particularly with larger clusters and in top-10 and top-15 scenarios, demonstrating its superior ability to exploit system vulnerabilities.

Among target item selection strategies, MUP-NNZ and MUP-Sum achieve the highest AvgHR values, especially in higher-ranked recommendations, due to their effectiveness in prioritizing influential items. Other strategies, such as AU and MUL, exhibit limited impact. The vulnerability comparison shows that $S_1$ is more susceptible to attacks than $S_2$, though the difference diminishes as cluster size grows.

The results in Tables 18 and 19 demonstrate that when the filler size increases to 5%, the AvgHR values rise significantly compared to the 1% filler condition. In the *GSAGen₁Ran* scenario, AvgHR values generally range between 10–20%, with MRP reaching up to nearly 20% and AU and MUL achieving around 16–19%. Although SC remains weak under small cluster sizes, it improves to nearly 17% with larger clusters, while MP and LM remain relatively less affected at around 10–15%. In contrast, the *GSAGen₁Avg* scenario exhibits stronger attack effects: with cluster size 0.1, MRP reaches 22–25% AvgHR at Top-15, followed by AU and MUL in the range of 18–22% and SC around 15–18%. These findings indicate that average-based attacks manipulate the system more effectively than random attacks. Among the aggregation strategies, MRP is the most vulnerable, consistently showing the highest AvgHR values, while AU, MUL, and SC are also significantly influenced. MP and LM, on the other hand, appear more resilient with comparatively lower hit ratios. Overall, the increase in filler size amplifies the attack impact on YM10, with MRP emerging as the weakest technique.

Overall, the results indicate that increasing the filler size to 5% substantially amplifies the attack impact on YM10, with average-based attacks (*GSAGen₁Avg*) being more effective than random ones, and MRP emerging as the most vulnerable aggregation strategy, while MP and LM remain relatively more robust.

In the final stage, a statistical analysis was conducted to evaluate the robustness of the system against adversarial attacks, focusing on the relationship between *GSAGen₁Avg* and *GSAGen₁Ran* (see Figs 5 and 6). According to the results of the paired t-test, the differences between *GSAGen₁Avg* and *GSAGen₁Ran* methods were generally limited.

**Table 14. AvgHR results of multi-criteria recommender systems on YM20 dataset for each aggregation techniques and filler size 5% (S1).**

| Attack Type | Cluster Size | Aggregation Techniques | S1 MUP-Usr | | | MUP-DR | | | MUP-NNZ | | | MUP-Sum | | |
|---|---|---|---|---|---|---|---|---|---|---|---|---|---|---|
| | | | Top-5 | Top-10 | Top-15 | Top-5 | Top-10 | Top-15 | Top-5 | Top-10 | Top-15 | Top-5 | Top-10 | Top-15 |
| $GSAGen_lRan$ | 0.01 | Avg | 0.025 | 0.045 | 0.045 | 0.1721 | 0.1702 | 0.1721 | 0.1538 | 0.1557 | 0.1588 | 0.1287 | 0.1457 | 0.1527 |
| | | AU | 0.03 | 0.0475 | 0.0583 | 0.0388 | 0.0416 | 0.0465 | 0.0896 | 0.0898 | 0.0918 | 0.1938 | 0.2043 | 0.2084 |
| | | MUL | 0.005 | 0.035 | 0.0833 | 0.1597 | 0.1541 | 0.1566 | 0.1476 | 0.1495 | 0.1526 | 0.1147 | 0.1252 | 0.1352 |
| | | SC | 0 | 0 | 0 | 0.1488 | 0.1505 | 0.156 | 0.1392 | 0.1443 | 0.1464 | 0.2047 | 0.2165 | 0.2177 |
| | | AwM | 0 | 0 | 0 | 0 | 0 | 0 | 0 | 0 | 0 | 0 | 0 | 0 |
| | | Avg | 0.025 | 0 | 0 | 0 | 0 | 0 | 0 | 0 | 0 | 0 | 0 | 0 |
| | | MP | 0.03 | 0.0675 | 0.0517 | 0 | 0 | 0 | 0 | 0 | 0 | 0 | 0 | 0 |
| | | LM | 0.015 | 0.055 | 0.0717 | 0.1628 | 0.1572 | 0.1566 | 0.0915 | 0.0943 | 0.0952 | 0.1023 | 0.117 | 0.1243 |
| | | MRP | 0.035 | 0.0425 | 0.08 | 0.1519 | 0.1626 | 0.1662 | 0.0828 | 0.0823 | 0.086 | 0.262 | 0.2813 | 0.2848 |
| | 0.05 | Avg | 0.0514 | 0.0395 | 0.0905 | 0.1307 | 0.1619 | 0.215 | 0.093 | 0.0953 | 0.1166 | 0.0837 | 0.1188 | 0.1659 |
| | | AU | 0.021 | 0.0733 | 0.1298 | 0.0448 | 0.0467 | 0.084 | 0.0631 | 0.075 | 0.1045 | 0.1456 | 0.1621 | 0.2188 |
| | | MUL | 0.0295 | 0.0543 | 0.1168 | 0.1301 | 0.1373 | 0.1628 | 0.0682 | 0.075 | 0.0964 | 0.0817 | 0.1236 | 0.1552 |
| | | SC | 0.021 | 0.0862 | 0.1171 | 0.1126 | 0.127 | 0.1561 | 0.0967 | 0.114 | 0.1341 | 0.1481 | 0.1811 | 0.2322 |
| | | AwM | 0 | 0 | 0 | 0 | 0 | 0 | 0 | 0 | 0 | 0 | 0 | 0 |
| | | Avg | 0.001 | 0 | 0 | 0 | 0 | 0 | 0.0042 | 0 | 0 | 0 | 0 | 0 |
| | | MP | 0 | 0.001 | 0.0194 | 0 | 0 | 0 | 0 | 0 | 0 | 0 | 0 | 0 |
| | | LM | 0.0286 | 0.0724 | 0.1302 | 0.1194 | 0.1352 | 0.1715 | 0.0491 | 0.0547 | 0.0665 | 0.0798 | 0.0959 | 0.1268 |
| | | MRP | 0.0219 | 0.0676 | 0.119 | 0.1095 | 0.1171 | 0.136 | 0.0654 | 0.0698 | 0.111 | 0.1504 | 0.1978 | 0.2264 |
| | 0.1 | Avg | 0.0619 | 0.1321 | 0.213 | 0.087 | 0.1667 | 0.202 | 0.0549 | 0.0602 | 0.0901 | 0.1042 | 0.1277 | 0.1792 |
| | | AU | 0.0837 | 0.1307 | 0.1443 | 0.0428 | 0.1026 | 0.1347 | 0.0367 | 0.0428 | 0.1112 | 0.0479 | 0.0658 | 0.1346 |
| | | MUL | 0.073 | 0.1447 | 0.1907 | 0.0795 | 0.157 | 0.225 | 0.0419 | 0.0642 | 0.1164 | 0.1516 | 0.1174 | 0.1726 |
| | | SC | 0.0395 | 0.1086 | 0.1285 | 0.0265 | 0.0735 | 0.1126 | 0.0298 | 0.0484 | 0.1085 | 0.0288 | 0.0663 | 0.1346 |
| | | AwM | 0 | 0 | 0 | 0 | 0 | 0 | 0 | 0 | 0 | 0 | 0 | 0 |
| | | Avg | 0 | 0 | 0 | 0 | 0 | 0 | 0.0149 | 0 | 0 | 0 | 0 | 0 |
| | | MP | 0 | 0 | 0 | 0 | 0 | 0 | 0 | 0 | 0 | 0 | 0 | 0 |
| | | LM | 0.067 | 0.0998 | 0.1414 | 0.0288 | 0.11 | 0.1691 | 0.0488 | 0.0867 | 0.1206 | 0.1288 | 0.1158 | 0.1391 |
| | | MRP | 0.0684 | 0.1363 | 0.2225 | 0.0712 | 0.127 | 0.1789 | 0.0753 | 0.0586 | 0.1062 | 0.12 | 0.0812 | 0.1705 |
| $GSAGen_lAvg$ | 0.01 | Avg | 0.025 | 0.025 | 0.0583 | 0.1922 | 0.1997 | 0.2084 | 0.2428 | 0.2512 | 0.2515 | 0.1628 | 0.1693 | 0.1771 |
| | | AU | 0.03 | 0.0375 | 0.0683 | 0.1008 | 0.1073 | 0.1088 | 0.1048 | 0.1111 | 0.1126 | 0.0729 | 0.0815 | 0.0878 |
| | | MUL | 0.025 | 0.0475 | 0.0967 | 0.1287 | 0.1296 | 0.1367 | 0.2009 | 0.2065 | 0.2102 | 0.2031 | 0.2171 | 0.2192 |
| | | SC | 0 | 0 | 0 | 0.0915 | 0.0915 | 0.0935 | 0.0919 | 0.1003 | 0.1053 | 0.0698 | 0.0723 | 0.0747 |
| | | AwM | 0 | 0 | 0 | 0 | 0 | 0 | 0 | 0 | 0 | 0 | 0 | 0 |
| | | Avg | 0.025 | 0 | 0 | 0 | 0 | 0 | 0 | 0 | 0 | 0 | 0 | 0 |
| | | MP | 0.03 | 0.045 | 0.0933 | 0 | 0 | 0 | 0 | 0 | 0 | 0 | 0 | 0 |
| | | LM | 0.015 | 0.0425 | 0.0983 | 0.1572 | 0.1535 | 0.1584 | 0.1575 | 0.1631 | 0.1664 | 0.1457 | 0.1606 | 0.1619 |
| | | MRP | 0.045 | 0.0625 | 0.085 | 0.0605 | 0.067 | 0.0685 | 0.3426 | 0.3457 | 0.3488 | 0.1237 | 0.1316 | 0.1344 |
| | 0.05 | Avg | 0.0048 | 0.021 | 0.0921 | 0.1478 | 0.1754 | 0.2146 | 0.1628 | 0.21 | 0.2202 | 0.1631 | 0.1578 | 0.2019 |
| | | AU | 0.0552 | 0.081 | 0.1279 | 0.0592 | 0.065 | 0.091 | 0.0876 | 0.1285 | 0.1535 | 0.06 | 0.1005 | 0.118 |
| | | MUL | 0.0162 | 0.0329 | 0.1006 | 0.1078 | 0.1561 | 0.1856 | 0.1315 | 0.1761 | 0.1735 | 0.1532 | 0.1611 | 0.1791 |
| | | SC | 0.0257 | 0.0743 | 0.1044 | 0.0527 | 0.055 | 0.0633 | 0.0891 | 0.1131 | 0.1416 | 0.0389 | 0.0359 | 0.0612 |
| | | AwM | 0 | 0 | 0 | 0 | 0 | 0 | 0 | 0 | 0 | 0 | 0 | 0 |
| | | Avg | 0.0019 | 0 | 0 | 0 | 0 | 0 | 0.0042 | 0 | 0 | 0 | 0 | 0 |
| | | MP | 0.001 | 0.0038 | 0.0149 | 0 | 0 | 0 | 0 | 0 | 0 | 0 | 0 | 0 |
| | | LM | 0.0114 | 0.041 | 0.1121 | 0.16 | 0.1679 | 0.1518 | 0.1188 | 0.1562 | 0.1578 | 0.114 | 0.1607 | 0.1794 |
| | | MRP | 0.0152 | 0.0529 | 0.1175 | 0.0681 | 0.0843 | 0.0899 | 0.2243 | 0.2664 | 0.2857 | 0.1898 | 0.206 | 0.2101 |
| | 0.1 | Avg | 0.0833 | 0.126 | 0.2198 | 0.0512 | 0.16 | 0.2239 | 0.0377 | 0.0837 | 0.1178 | 0.086 | 0.1233 | 0.1983 |
| | | AU | 0.0391 | 0.0884 | 0.1391 | 0.0433 | 0.1223 | 0.1403 | 0.0349 | 0.094 | 0.1082 | 0.0623 | 0.0942 | 0.1634 |
| | | MUL | 0.053 | 0.1067 | 0.1843 | 0.0386 | 0.1491 | 0.1933 | 0.0502 | 0.0802 | 0.1088 | 0.074 | 0.1172 | 0.2059 |
| | | SC | 0.0567 | 0.0677 | 0.1005 | 0.0428 | 0.0991 | 0.1188 | 0.0205 | 0.0791 | 0.1039 | 0.0228 | 0.0688 | 0.136 |
| | | AwM | 0 | 0 | 0 | 0 | 0 | 0 | 0 | 0 | 0 | 0 | 0 | 0 |
| | | Avg | 0 | 0 | 0 | 0 | 0 | 0 | 0.0177 | 0 | 0 | 0 | 0 | 0 |
| | | MP | 0 | 0 | 0 | 0 | 0 | 0 | 0 | 0 | 0 | 0 | 0 | 0 |
| | | LM | 0.0558 | 0.1002 | 0.1512 | 0.0377 | 0.0893 | 0.1512 | 0.0381 | 0.0712 | 0.1257 | 0.027 | 0.1079 | 0.146 |
| | | MRP | 0.1051 | 0.1114 | 0.1781 | 0.0651 | 0.1402 | 0.1981 | 0.0367 | 0.1007 | 0.1184 | 0.0842 | 0.1316 | 0.2113 |

**Table 15. AvgHR results of multi-criteria recommender systems on YM20 dataset for each aggregation techniques and filler size 5% (S2).**

| Attack Type | Cluster Size | Aggregation Techniques | S2 | | | | | | | | | | | |
| --- | --- | --- | --- | --- | --- | --- | --- | --- | --- | --- | --- | --- | --- | --- |
| | | | MUP-Usr | | | MUP-DR | | | MUP-NNZ | | | MUP-Sum | | |
| | | | Top-5 | Top-10 | Top-15 | Top-5 | Top-10 | Top-15 | Top-5 | Top-10 | Top-15 | Top-5 | Top-10 | Top-15 |
| $GSAGen_lRan$ | 0.01 | Avg | 0.015 | 0.035 | 0.0417 | 0.0558 | 0.056 | 0.0634 | 0.0967 | 0.1014 | 0.1036 | 0.0757 | 0.0789 | 0.0788 |
| | | AU | 0.0438 | 0.0356 | 0.0713 | 0.0797 | 0.0856 | 0.0876 | 0.0801 | 0.0862 | 0.0882 | 0.2379 | 0.2426 | 0.2468 |
| | | MUL | 0.0138 | 0.0638 | 0.0996 | 0.0341 | 0.0343 | 0.039 | 0.1169 | 0.1234 | 0.1269 | 0.0147 | 0.018 | 0.0198 |
| | | SC | 0 | 0 | 0 | 0.1024 | 0.1107 | 0.1191 | 0.036 | 0.0388 | 0.0429 | 0.2368 | 0.2416 | 0.2464 |
| | | AwM | 0 | 0 | 0 | 0 | 0 | 0 | 0 | 0 | 0 | 0 | 0 | 0 |
| | | Avg | 0 | 0 | 0 | 0 | 0 | 0 | 0.0019 | 0 | 0.0003 | 0 | 0 | 0 |
| | | MP | 0.0225 | 0.0313 | 0.0783 | 0.062 | 0.0629 | 0.0666 | 0.2223 | 0.2252 | 0.2289 | 0.0352 | 0.0399 | 0.0445 |
| | | LM | 0.0238 | 0.0563 | 0.0738 | 0.0667 | 0.0668 | 0.0755 | 0.1978 | 0.2016 | 0.2047 | 0.0457 | 0.0481 | 0.0481 |
| | | MRP | 0.0175 | 0.0369 | 0.0817 | 0.0511 | 0.0566 | 0.0588 | 0.0581 | 0.0639 | 0.0631 | 0.1977 | 0.2018 | 0.2055 |
| | 0.05 | Avg | 0.0167 | 0.0343 | 0.0548 | 0.0562 | 0.0765 | 0.1291 | 0.0636 | 0.1015 | 0.1139 | 0.0745 | 0.0941 | 0.0949 |
| | | AU | 0.0431 | 0.072 | 0.1213 | 0.0811 | 0.1104 | 0.125 | 0.0581 | 0.0809 | 0.0961 | 0.1554 | 0.1793 | 0.2051 |
| | | MUL | 0.0136 | 0.0546 | 0.0843 | 0.0232 | 0.0463 | 0.087 | 0.0795 | 0.1093 | 0.1339 | 0.0395 | 0.0564 | 0.0551 |
| | | SC | 0.0271 | 0.0768 | 0.1189 | 0.0782 | 0.1053 | 0.1402 | 0.0474 | 0.053 | 0.0748 | 0.162 | 0.1722 | 0.1923 |
| | | AwM | 0 | 0 | 0 | 0 | 0 | 0 | 0 | 0 | 0 | 0 | 0 | 0 |
| | | Avg | 0 | 0 | 0 | 0 | 0 | 0 | 0.0049 | 0 | 0.0008 | 0 | 0 | 0 |
| | | MP | 0.0333 | 0.0675 | 0.0787 | 0.041 | 0.0562 | 0.0838 | 0.1503 | 0.1715 | 0.1966 | 0.0437 | 0.0644 | 0.0884 |
| | | LM | 0.0233 | 0.0512 | 0.0876 | 0.0437 | 0.0724 | 0.1054 | 0.1303 | 0.1624 | 0.1853 | 0.0533 | 0.0672 | 0.0665 |
| | | MRP | 0.0283 | 0.0365 | 0.0825 | 0.0533 | 0.0915 | 0.1034 | 0.0719 | 0.09 | 0.0925 | 0.1516 | 0.1793 | 0.1785 |
| | 0.1 | Avg | 0.0774 | 0.114 | 0.1539 | 0.0777 | 0.1126 | 0.1413 | 0.0413 | 0.0716 | 0.1187 | 0.0458 | 0.1135 | 0.1423 |
| | | AU | 0.0479 | 0.0683 | 0.1162 | 0.0265 | 0.0889 | 0.141 | 0.0349 | 0.0787 | 0.1374 | 0.0471 | 0.0869 | 0.139 |
| | | MUL | 0.0673 | 0.1148 | 0.1495 | 0.0773 | 0.1077 | 0.1442 | 0.0409 | 0.0626 | 0.1207 | 0.0492 | 0.1122 | 0.1407 |
| | | SC | 0.0306 | 0.0689 | 0.1174 | 0.0266 | 0.0753 | 0.1197 | 0.0312 | 0.0638 | 0.1119 | 0.0417 | 0.083 | 0.1198 |
| | | AwM | 0 | 0 | 0 | 0 | 0 | 0 | 0 | 0 | 0 | 0 | 0 | 0 |
| | | Avg | 0 | 0 | 0 | 0 | 0 | 0 | 0.005 | 0 | 0 | 0 | 0 | 0 |
| | | MP | 0.0387 | 0.0774 | 0.1152 | 0.0515 | 0.0753 | 0.1175 | 0.0605 | 0.0679 | 0.1338 | 0.0536 | 0.0968 | 0.1357 |
| | | LM | 0.0485 | 0.0896 | 0.1337 | 0.0623 | 0.1019 | 0.1395 | 0.0367 | 0.0647 | 0.1175 | 0.0373 | 0.0943 | 0.1411 |
| | | MRP | 0.0484 | 0.1049 | 0.136 | 0.0599 | 0.0951 | 0.1407 | 0.0431 | 0.0672 | 0.1278 | 0.0502 | 0.1034 | 0.1345 |
| $GSAGen_lAvg$ | 0.01 | Avg | 0.015 | 0.0356 | 0.0633 | 0.0967 | 0.1007 | 0.1043 | 0.1662 | 0.1736 | 0.183 | 0.1095 | 0.1125 | 0.1132 |
| | | AU | 0.0388 | 0.0375 | 0.0763 | 0.0977 | 0.106 | 0.1113 | 0.1901 | 0.1942 | 0.1979 | 0.0782 | 0.0806 | 0.0815 |
| | | MUL | 0.0163 | 0.06 | 0.1146 | 0.1029 | 0.1069 | 0.1133 | 0.1385 | 0.1441 | 0.1505 | 0.1056 | 0.1123 | 0.1155 |
| | | SC | 0 | 0 | 0 | 0.1414 | 0.1487 | 0.1572 | 0.1331 | 0.1381 | 0.1415 | 0.1142 | 0.1225 | 0.1237 |
| | | AwM | 0 | 0 | 0 | 0 | 0 | 0 | 0 | 0 | 0 | 0 | 0 | 0 |
| | | Avg | 0 | 0 | 0 | 0 | 0 | 0 | 0.0005 | 0.0007 | 0 | 0 | 0 | 0 |
| | | MP | 0.0188 | 0.0356 | 0.0796 | 0.093 | 0.1045 | 0.1157 | 0.1087 | 0.1159 | 0.1241 | 0.1503 | 0.1634 | 0.1683 |
| | | LM | 0.015 | 0.0431 | 0.0879 | 0.0578 | 0.0697 | 0.0727 | 0.0607 | 0.0684 | 0.0727 | 0.1504 | 0.1563 | 0.1607 |
| | | MRP | 0.0338 | 0.0563 | 0.0979 | 0.107 | 0.1089 | 0.1209 | 0.1827 | 0.1868 | 0.19 | 0.1548 | 0.1573 | 0.1605 |
| | 0.05 | Avg | 0.004 | 0.0295 | 0.0638 | 0.1046 | 0.1309 | 0.1391 | 0.1162 | 0.14 | 0.1841 | 0.0884 | 0.1095 | 0.1145 |
| | | AU | 0.0248 | 0.07 | 0.1045 | 0.07 | 0.0881 | 0.1141 | 0.1519 | 0.1764 | 0.195 | 0.0429 | 0.0545 | 0.0725 |
| | | MUL | 0.0043 | 0.0377 | 0.0947 | 0.1025 | 0.1325 | 0.1425 | 0.1128 | 0.1428 | 0.168 | 0.0841 | 0.1078 | 0.1195 |
| | | SC | 0.0281 | 0.066 | 0.0969 | 0.0802 | 0.1119 | 0.1334 | 0.098 | 0.1229 | 0.1545 | 0.0883 | 0.0995 | 0.1171 |
| | | AwM | 0 | 0 | 0 | 0 | 0 | 0 | 0 | 0 | 0 | 0 | 0 | 0 |
| | | Avg | 0 | 0 | 0 | 0 | 0 | 0.0007 | 0 | 0.0012 | 0.0016 | 0 | 0 | 0 |
| | | MP | 0.0238 | 0.0793 | 0.1017 | 0.1094 | 0.1678 | 0.2031 | 0.0957 | 0.1392 | 0.1691 | 0.1124 | 0.1482 | 0.1683 |
| | | LM | 0.0131 | 0.0542 | 0.1059 | 0.0642 | 0.1211 | 0.1367 | 0.0419 | 0.0769 | 0.1011 | 0.0957 | 0.13 | 0.1595 |
| | | MRP | 0.0179 | 0.051 | 0.086 | 0.0803 | 0.1085 | 0.1433 | 0.1479 | 0.1737 | 0.1869 | 0.0912 | 0.1125 | 0.1247 |
| | 0.1 | Avg | 0.0744 | 0.1105 | 0.1534 | 0.0728 | 0.1202 | 0.1618 | 0.0403 | 0.0684 | 0.1057 | 0.0615 | 0.1187 | 0.1443 |
| | | AU | 0.061 | 0.1094 | 0.137 | 0.0403 | 0.107 | 0.1314 | 0.0217 | 0.0801 | 0.1229 | 0.0244 | 0.081 | 0.1159 |
| | | MUL | 0.0871 | 0.1064 | 0.1542 | 0.0805 | 0.1176 | 0.1694 | 0.0481 | 0.0752 | 0.1135 | 0.0621 | 0.1205 | 0.141 |
| | | SC | 0.046 | 0.0978 | 0.1242 | 0.0367 | 0.1009 | 0.1267 | 0.0207 | 0.072 | 0.1119 | 0.0165 | 0.0804 | 0.1115 |
| | | AwM | 0 | 0 | 0 | 0 | 0 | 0 | 0 | 0 | 0 | 0 | 0 | 0 |
| | | Avg | 0 | 0 | 0 | 0 | 0 | 0 | 0 | 0 | 0 | 0 | 0 | 0 |
| | | MP | 0.0498 | 0.0791 | 0.1161 | 0.0417 | 0.0877 | 0.1209 | 0.0385 | 0.0772 | 0.125 | 0.0264 | 0.078 | 0.1248 |
| | | LM | 0.0714 | 0.0769 | 0.1341 | 0.064 | 0.1023 | 0.1439 | 0.0353 | 0.0863 | 0.132 | 0.0274 | 0.0881 | 0.144 |
| | | MRP | 0.0572 | 0.1041 | 0.1484 | 0.0487 | 0.0976 | 0.1383 | 0.0299 | 0.0886 | 0.1312 | 0.027 | 0.1153 | 0.1273 |

**Table 16.** AvgHR results of multi-criteria recommender systems on YM10 dataset for each aggregation techniques and filler size 1% (S1).

| Attack Type | Cluster Size | Aggregation Techniques | S1 | | | | | | | | | | | |
|---|---|---|---|---|---|---|---|---|---|---|---|---|---|---|
| | | | MUP-Usr | | | MUP-DR | | | MUP-NNZ | | | MUP-Sum | | |
| | | | Top-5 | Top-10 | Top-15 | Top-5 | Top-10 | Top-15 | Top-5 | Top-10 | Top-15 | Top-5 | Top-10 | Top-15 |
| $GSAGen_lRan$ | 0.01 | Avg | 0.0667 | 0.1339 | 0.14 | 0.071 | 0.0712 | 0.0737 | 0.21 | 0.2144 | 0.2159 | 0.1688 | 0.1718 | 0.1747 |
| | | AU | 0.0622 | 0.0994 | 0.1204 | 0.1681 | 0.1731 | 0.1706 | 0.2134 | 0.2149 | 0.2225 | 0.064 | 0.0687 | 0.0737 |
| | | MUL | 0.0622 | 0.0872 | 0.1444 | 0.071 | 0.0712 | 0.0716 | 0.2023 | 0.2075 | 0.2089 | 0.1149 | 0.1179 | 0.1208 |
| | | SC | 0.0011 | 0.0894 | 0.0696 | 0.166 | 0.17 | 0.1726 | 0.1419 | 0.1413 | 0.1443 | 0.0487 | 0.0485 | 0.0562 |
| | | AwM | 0 | 0 | 0 | 0 | 0 | 0 | 0 | 0 | 0 | 0 | 0 | 0 |
| | | Avg | 0.0067 | 0 | 0.0059 | 0 | 0 | 0 | 0 | 0 | 0 | 0 | 0 | 0 |
| | | MP | 0.01 | 0.0194 | 0.0204 | 0 | 0 | 0 | 0 | 0 | 0 | 0 | 0 | 0 |
| | | LM | 0.0033 | 0.0578 | 0.1319 | 0.102 | 0.1022 | 0.1026 | 0.1166 | 0.1234 | 0.1235 | 0.1328 | 0.1368 | 0.1407 |
| | | MRP | 0.0267 | 0.0911 | 0.1119 | 0.1584 | 0.1667 | 0.169 | 0.2043 | 0.2063 | 0.2093 | 0.1647 | 0.1683 | 0.1743 |
| | 0.05 | Avg | 0.0325 | 0.0471 | 0.0927 | 0.0301 | 0.0363 | 0.0566 | 0.1298 | 0.1486 | 0.1803 | 0.0718 | 0.0915 | 0.1045 |
| | | AU | 0.0697 | 0.1176 | 0.0988 | 0.1659 | 0.1946 | 0.1895 | 0.1623 | 0.1674 | 0.185 | 0.0698 | 0.0877 | 0.1097 |
| | | MUL | 0.0519 | 0.0525 | 0.0931 | 0.0262 | 0.0372 | 0.0505 | 0.1121 | 0.1319 | 0.1649 | 0.0408 | 0.0671 | 0.0766 |
| | | SC | 0.0629 | 0.1156 | 0.105 | 0.1832 | 0.204 | 0.1922 | 0.1199 | 0.1125 | 0.1211 | 0.0585 | 0.0734 | 0.0839 |
| | | AwM | 0 | 0 | 0 | 0 | 0 | 0 | 0 | 0 | 0 | 0 | 0 | 0 |
| | | Avg | 0 | 0 | 0.003 | 0 | 0 | 0 | 0 | 0 | 0 | 0 | 0 | 0 |
| | | MP | 0 | 0 | 0 | 0 | 0 | 0 | 0 | 0 | 0 | 0 | 0 | 0 |
| | | LM | 0.0486 | 0.048 | 0.1081 | 0.0528 | 0.0644 | 0.0716 | 0.0679 | 0.0942 | 0.1174 | 0.0456 | 0.0724 | 0.0895 |
| | | MRP | 0.0563 | 0.1159 | 0.0916 | 0.0988 | 0.1485 | 0.169 | 0.1446 | 0.1633 | 0.1857 | 0.0803 | 0.1146 | 0.1329 |
| | 0.1 | Avg | 0.0726 | 0.1063 | 0.1592 | 0.1046 | 0.1465 | 0.1706 | 0.0557 | 0.0911 | 0.1398 | 0.0768 | 0.1053 | 0.105 |
| | | AU | 0.0661 | 0.1094 | 0.1402 | 0.0257 | 0.1226 | 0.1328 | 0.0361 | 0.0757 | 0.1334 | 0.0459 | 0.0737 | 0.1479 |
| | | MUL | 0.0842 | 0.107 | 0.1605 | 0.1046 | 0.1439 | 0.1626 | 0.0611 | 0.0967 | 0.1412 | 0.0783 | 0.1108 | 0.1164 |
| | | SC | 0.0812 | 0.0917 | 0.1291 | 0.02 | 0.089 | 0.0995 | 0.0518 | 0.0669 | 0.1066 | 0.0526 | 0.081 | 0.1406 |
| | | AwM | 0 | 0 | 0 | 0 | 0 | 0 | 0 | 0 | 0 | 0 | 0 | 0 |
| | | Avg | 0 | 0 | 0 | 0 | 0 | 0 | 0 | 0 | 0 | 0 | 0 | 0 |
| | | MP | 0 | 0 | 0 | 0 | 0 | 0 | 0 | 0 | 0 | 0 | 0 | 0 |
| | | LM | 0.0651 | 0.0862 | 0.1157 | 0.036 | 0.1055 | 0.1487 | 0.0916 | 0.1345 | 0.1381 | 0.0879 | 0.095 | 0.1157 |
| | | MRP | 0.0632 | 0.1313 | 0.1536 | 0.0507 | 0.1393 | 0.1213 | 0.0581 | 0.0895 | 0.1283 | 0.0555 | 0.0718 | 0.1427 |
| $GSAGen_lAvg$ | 0.01 | Avg | 0.0522 | 0.1083 | 0.1448 | 0.1843 | 0.1863 | 0.1929 | 0.2357 | 0.2367 | 0.2432 | 0.2713 | 0.2734 | 0.2765 |
| | | AU | 0.0544 | 0.06 | 0.1474 | 0.1572 | 0.1589 | 0.1684 | 0.0825 | 0.0878 | 0.0866 | 0.1405 | 0.1394 | 0.1464 |
| | | MUL | 0.01 | 0.1122 | 0.1219 | 0.1475 | 0.1495 | 0.1536 | 0.2128 | 0.2137 | 0.2213 | 0.1985 | 0.2014 | 0.2085 |
| | | SC | 0.0156 | 0.0417 | 0.1 | 0.197 | 0.1989 | 0.2093 | 0.0977 | 0.1024 | 0.1005 | 0.1274 | 0.1268 | 0.1304 |
| | | AwM | 0 | 0 | 0 | 0 | 0 | 0 | 0 | 0 | 0 | 0 | 0 | 0 |
| | | Avg | 0.0056 | 0.0044 | 0.0037 | 0 | 0 | 0 | 0 | 0 | 0 | 0 | 0 | 0 |
| | | MP | 0.0078 | 0.0022 | 0.047 | 0 | 0 | 0 | 0 | 0 | 0 | 0 | 0 | 0 |
| | | LM | 0.02 | 0.0478 | 0.1044 | 0.0775 | 0.0777 | 0.0804 | 0.1578 | 0.1588 | 0.1634 | 0.2699 | 0.2701 | 0.2741 |
| | | MRP | 0.0822 | 0.07 | 0.1359 | 0.1523 | 0.1579 | 0.1645 | 0.1178 | 0.124 | 0.1238 | 0.1626 | 0.1672 | 0.1711 |
| | 0.05 | Avg | 0.0769 | 0.0788 | 0.1031 | 0.108 | 0.1178 | 0.1345 | 0.1209 | 0.1336 | 0.1578 | 0.186 | 0.2079 | 0.2348 |
| | | AU | 0.0908 | 0.109 | 0.1191 | 0.0858 | 0.1186 | 0.1563 | 0.0868 | 0.1184 | 0.1387 | 0.1056 | 0.126 | 0.1458 |
| | | MUL | 0.0646 | 0.0642 | 0.1103 | 0.086 | 0.0928 | 0.1137 | 0.1031 | 0.1196 | 0.1409 | 0.1311 | 0.1572 | 0.1783 |
| | | SC | 0.0512 | 0.0888 | 0.1331 | 0.1058 | 0.1177 | 0.1509 | 0.1105 | 0.1364 | 0.1583 | 0.1218 | 0.1358 | 0.154 |
| | | AwM | 0 | 0 | 0 | 0 | 0 | 0 | 0 | 0 | 0 | 0 | 0 | 0 |
| | | Avg | 0 | 0 | 0 | 0 | 0 | 0 | 0 | 0 | 0 | 0.0004 | 0.0048 | 0 |
| | | MP | 0 | 0 | 0 | 0 | 0 | 0 | 0 | 0 | 0 | 0 | 0 | 0 |
| | | LM | 0.0237 | 0.1279 | 0.1001 | 0.0599 | 0.0693 | 0.0785 | 0.0872 | 0.0917 | 0.1099 | 0.1715 | 0.1823 | 0.2093 |
| | | MRP | 0.0473 | 0.0921 | 0.0995 | 0.1034 | 0.1237 | 0.1542 | 0.0719 | 0.0829 | 0.1153 | 0.1467 | 0.1513 | 0.1821 |
| | 0.1 | Avg | 0.0852 | 0.1282 | 0.1631 | 0.0647 | 0.0827 | 0.1531 | 0.019 | 0.0407 | 0.1345 | 0.0283 | 0.0894 | 0.1109 |
| | | AU | 0.0294 | 0.1149 | 0.1147 | 0.0805 | 0.1104 | 0.1322 | 0.0674 | 0.0802 | 0.127 | 0.0238 | 0.1039 | 0.1321 |
| | | MUL | 0.0943 | 0.1398 | 0.166 | 0.0662 | 0.0723 | 0.1507 | 0.019 | 0.0373 | 0.1274 | 0.0345 | 0.1014 | 0.1141 |
| | | SC | 0.0222 | 0.1055 | 0.1211 | 0.0243 | 0.0678 | 0.1301 | 0.0691 | 0.0651 | 0.1046 | 0.0507 | 0.0774 | 0.1156 |
| | | AwM | 0 | 0 | 0 | 0 | 0 | 0 | 0 | 0 | 0 | 0 | 0 | 0 |
| | | Avg | 0.0002 | 0.0099 | 0 | 0 | 0 | 0 | 0 | 0 | 0 | 0 | 0 | 0 |
| | | MP | 0 | 0 | 0 | 0 | 0 | 0 | 0 | 0 | 0 | 0 | 0 | 0 |
| | | LM | 0.0438 | 0.1036 | 0.1382 | 0.0057 | 0.086 | 0.1375 | 0.041 | 0.0425 | 0.1444 | 0.0354 | 0.1147 | 0.1023 |
| | | MRP | 0.0177 | 0.108 | 0.1319 | 0.1205 | 0.0925 | 0.1511 | 0.0345 | 0.0791 | 0.1347 | 0.0516 | 0.084 | 0.1261 |

**Table 17.** AvgHR results of multi-criteria recommender systems on YM10 dataset for each aggregation techniques and filler size 1% (S2).

| Attack Type | Cluster Size | Aggregation Techniques | S2 | | | | | | | | | | | |
| --- | --- | --- | --- | --- | --- | --- | --- | --- | --- | --- | --- | --- | --- | --- |
| | | | MUP-Usr | | | MUP-DR | | | MUP-NNZ | | | MUP-Sum | | |
| | | | Top-5 | Top-10 | Top-15 | Top-5 | Top-10 | Top-15 | Top-5 | Top-10 | Top-15 | Top-5 | Top-10 | Top-15 |
| GSAGen$_l$Ran | 0.01 | Avg | 0.0972 | 0.1333 | 0.2204 | 0.1224 | 0.1322 | 0.1355 | 0.2754 | 0.2877 | 0.2894 | 0.0763 | 0.0788 | 0.0796 |
| | | AU | 0.1333 | 0.1111 | 0.1111 | 0.1945 | 0.1982 | 0.2077 | 0.1942 | 0.1867 | 0.1902 | 0.1959 | 0.1967 | 0.1947 |
| | | MUL | 0.0583 | 0.0986 | 0.1981 | 0.1231 | 0.133 | 0.1415 | 0.2754 | 0.2859 | 0.289 | 0.1 | 0.1025 | 0.1 |
| | | SC | 0.0389 | 0.1014 | 0.1185 | 0.1689 | 0.1766 | 0.183 | 0.1632 | 0.1534 | 0.1572 | 0.1855 | 0.1908 | 0.1839 |
| | | AwM | 0 | 0 | 0 | 0 | 0 | 0 | 0 | 0 | 0 | 0 | 0 | 0 |
| | | Avg | 0 | 0 | 0 | 0 | 0 | 0 | 0 | 0 | 0 | 0 | 0 | 0 |
| | | MP | 0.0694 | 0.075 | 0.1463 | 0.1016 | 0.11 | 0.1131 | 0.2532 | 0.2637 | 0.2675 | 0.0947 | 0.1002 | 0.1067 |
| | | LM | 0.0056 | 0.0528 | 0.1185 | 0.1311 | 0.1291 | 0.1279 | 0.2124 | 0.2222 | 0.2255 | 0.0947 | 0.1021 | 0.1029 |
| | | MRP | 0.125 | 0.0708 | 0.1435 | 0.1623 | 0.1654 | 0.1778 | 0.2936 | 0.3132 | 0.3205 | 0.2029 | 0.2106 | 0.2095 |
| | 0.05 | Avg | 0.039 | 0.0492 | 0.1284 | 0.0692 | 0.1237 | 0.1524 | 0.1643 | 0.2259 | 0.2198 | 0.1286 | 0.141 | 0.1044 |
| | | AU | 0.1055 | 0.0615 | 0.0982 | 0.1668 | 0.1597 | 0.189 | 0.1527 | 0.1213 | 0.1413 | 0.1216 | 0.178 | 0.2163 |
| | | MUL | 0.0374 | 0.0591 | 0.1253 | 0.0699 | 0.1336 | 0.166 | 0.1643 | 0.227 | 0.2148 | 0.1355 | 0.1516 | 0.1052 |
| | | SC | 0.061 | 0.067 | 0.0894 | 0.1368 | 0.1323 | 0.1763 | 0.1349 | 0.0876 | 0.1235 | 0.1481 | 0.1985 | 0.2109 |
| | | AwM | 0 | 0 | 0 | 0 | 0 | 0 | 0 | 0 | 0 | 0 | 0 | 0 |
| | | Avg | 0 | 0 | 0 | 0 | 0 | 0 | 0 | 0 | 0 | 0 | 0 | 0 |
| | | MP | 0.0126 | 0.0206 | 0.0511 | 0.0485 | 0.0765 | 0.1276 | 0.1432 | 0.2036 | 0.2005 | 0.074 | 0.1097 | 0.1139 |
| | | LM | 0.0286 | 0.0558 | 0.098 | 0.1577 | 0.1378 | 0.1689 | 0.1494 | 0.1994 | 0.2025 | 0.067 | 0.149 | 0.1242 |
| | | MRP | 0.0626 | 0.0978 | 0.1317 | 0.1068 | 0.1358 | 0.1647 | 0.2149 | 0.2629 | 0.2726 | 0.1995 | 0.2313 | 0.2529 |
| | 0.1 | Avg | 0.0033 | 0.0944 | 0.16 | 0.0828 | 0.0877 | 0.1679 | 0.0899 | 0.1525 | 0.0762 | 0.0246 | 0.2156 | 0.1392 |
| | | AU | 0.0536 | 0.0654 | 0.1782 | 0.0265 | 0.0865 | 0.1588 | 0.0639 | 0.138 | 0.158 | 0.0579 | 0.0568 | 0.1255 |
| | | MUL | 0.0044 | 0.1051 | 0.1308 | 0.0495 | 0.0728 | 0.1923 | 0.0893 | 0.1519 | 0.1048 | 0.0478 | 0.2157 | 0.1709 |
| | | SC | 0.112 | 0.0624 | 0.1466 | 0.047 | 0.0898 | 0.1663 | 0.0639 | 0.0784 | 0.1799 | 0.038 | 0.0225 | 0.1216 |
| | | AwM | 0 | 0 | 0 | 0 | 0 | 0 | 0 | 0 | 0 | 0 | 0 | 0 |
| | | Avg | 0.0003 | 0.0003 | 0.0165 | 0 | 0 | 0 | 0 | 0 | 0 | 0 | 0 | 0 |
| | | MP | 0.0085 | 0.0443 | 0.126 | 0.0847 | 0.0934 | 0.1327 | 0.0481 | 0.1496 | 0.1072 | 0.0284 | 0.1919 | 0.1441 |
| | | LM | 0.0161 | 0.0497 | 0.1088 | 0.1331 | 0.0437 | 0.1543 | 0.106 | 0.1844 | 0.0984 | 0.0085 | 0.1328 | 0.0902 |
| | | MRP | 0.0098 | 0.1111 | 0.1804 | 0.1008 | 0.1105 | 0.2158 | 0.0549 | 0.2142 | 0.2061 | 0.0057 | 0.1231 | 0.173 |
| GSAGen$_l$Avg | 0.01 | Avg | 0.0722 | 0.0722 | 0.1454 | 0.1781 | 0.1781 | 0.1851 | 0.1144 | 0.1046 | 0.1036 | 0.0882 | 0.1069 | 0.116 |
| | | AU | 0.0083 | 0.0694 | 0.075 | 0.1719 | 0.1862 | 0.1863 | 0.1188 | 0.1339 | 0.1415 | 0.0924 | 0.0898 | 0.0923 |
| | | MUL | 0.0639 | 0.0875 | 0.1454 | 0.1792 | 0.1792 | 0.1911 | 0.1126 | 0.1027 | 0.1133 | 0.1427 | 0.1441 | 0.1457 |
| | | SC | 0 | 0.0431 | 0.0731 | 0.1199 | 0.1294 | 0.1358 | 0.0587 | 0.072 | 0.0767 | 0.0554 | 0.0548 | 0.06 |
| | | AwM | 0 | 0 | 0 | 0 | 0 | 0 | 0 | 0 | 0 | 0 | 0 | 0 |
| | | Avg | 0.0028 | 0 | 0.0093 | 0 | 0 | 0 | 0.0587 | 0.0587 | 0.0587 | 0 | 0 | 0 |
| | | MP | 0.0056 | 0.0542 | 0.1278 | 0.3035 | 0.3052 | 0.3031 | 0.0747 | 0.0863 | 0.0923 | 0.094 | 0.0964 | 0.1051 |
| | | LM | 0.0333 | 0.0542 | 0.1037 | 0.2007 | 0.2007 | 0.1978 | 0.0958 | 0.0936 | 0.0897 | 0.1264 | 0.1264 | 0.1455 |
| | | MRP | 0.0167 | 0.0569 | 0.1176 | 0.1457 | 0.1475 | 0.1556 | 0.1042 | 0.1038 | 0.1076 | 0.1119 | 0.1327 | 0.1278 |
| | 0.05 | Avg | 0.1093 | 0.0843 | 0.0641 | 0.0984 | 0.1179 | 0.1329 | 0.1144 | 0.1414 | 0.1422 | 0.0601 | 0.1639 | 0.173 |
| | | AU | 0.0352 | 0.0934 | 0.1392 | 0.1006 | 0.1289 | 0.1232 | 0.0648 | 0.1194 | 0.1479 | 0.0379 | 0.0878 | 0.0985 |
| | | MUL | 0.0945 | 0.0651 | 0.0813 | 0.0984 | 0.119 | 0.1451 | 0.1158 | 0.1541 | 0.1468 | 0.0989 | 0.1803 | 0.1636 |
| | | SC | 0.0198 | 0.0885 | 0.1899 | 0.0613 | 0.07 | 0.0971 | 0.0378 | 0.0908 | 0.0984 | 0.02 | 0.0286 | 0.0764 |
| | | AwM | 0 | 0 | 0 | 0 | 0 | 0 | 0 | 0 | 0 | 0 | 0 | 0 |
| | | Avg | 0 | 0 | 0 | 0 | 0 | 0 | 0.0321 | 0.0321 | 0.0321 | 0 | 0 | 0 |
| | | MP | 0.0297 | 0.0335 | 0.115 | 0.1654 | 0.2058 | 0.2111 | 0.0481 | 0.1025 | 0.1209 | 0.067 | 0.0752 | 0.1154 |
| | | LM | 0.0126 | 0.1382 | 0.146 | 0.1049 | 0.1179 | 0.1361 | 0.1086 | 0.1254 | 0.1355 | 0.0692 | 0.0952 | 0.1425 |
| | | MRP | 0.0637 | 0.0679 | 0.0837 | 0.0659 | 0.0939 | 0.1266 | 0.0517 | 0.1323 | 0.1204 | 0.1318 | 0.1815 | 0.1853 |
| | 0.1 | Avg | 0.053 | 0.0989 | 0.1255 | 0.0066 | 0.0997 | 0.1523 | 0.0139 | 0.171 | 0.1227 | 0.018 | 0.1053 | 0.1774 |
| | | AU | 0.1057 | 0.115 | 0.1446 | 0.0631 | 0.0978 | 0.173 | 0.0525 | 0.1607 | 0.1313 | 0.041 | 0.0728 | 0.126 |
| | | MUL | 0.0533 | 0.0929 | 0.1291 | 0.0066 | 0.0995 | 0.129 | 0.0139 | 0.1724 | 0.1258 | 0.044 | 0.1034 | 0.174 |
| | | SC | 0.1016 | 0.0929 | 0.1367 | 0.0462 | 0.0429 | 0.1134 | 0.0279 | 0.1072 | 0.1188 | 0.05 | 0.0891 | 0.0708 |
| | | AwM | 0 | 0 | 0 | 0 | 0 | 0 | 0 | 0 | 0 | 0 | 0 | 0 |
| | | Avg | 0.0003 | 0 | 0 | 0.0005 | 0.0253 | 0.0163 | 0 | 0 | 0 | 0 | 0 | 0.0015 |
| | | MP | 0.0301 | 0.0653 | 0.1505 | 0.0027 | 0.1184 | 0.0949 | 0 | 0.0458 | 0.0813 | 0 | 0.0523 | 0.14 |
| | | LM | 0.012 | 0.0577 | 0.1547 | 0.038 | 0.0974 | 0.1068 | 0.0139 | 0.0802 | 0.0915 | 0.05 | 0.1448 | 0.1478 |
| | | MRP | 0.0549 | 0.156 | 0.1723 | 0.0511 | 0.122 | 0.1869 | 0.0396 | 0.1817 | 0.1611 | 0.0342 | 0.1145 | 0.1734 |

**Table 18. AvgHR results of multi-criteria recommender systems on YM10 dataset for each aggregation techniques and filler size 5% (S1).**

| Attack Type | Cluster Size | Aggregation Techniques | S1 MUP-Usr Top-5 | Top-10 | Top-15 | MUP-DR Top-5 | Top-10 | Top-15 | MUP-NNZ Top-5 | Top-10 | Top-15 | MUP-Sum Top-5 | Top-10 | Top-15 |
|---|---|---|---|---|---|---|---|---|---|---|---|---|---|---|
| GSAGen_lRan | 0.01 | Avg | 0.0644 | 0.0772 | 0.1556 | 0.0776 | 0.0825 | 0.0915 | 0.1244 | 0.1266 | 0.1322 | 0.1942 | 0.1997 | 0.203 |
| | | AU | 0.0367 | 0.0883 | 0.1319 | 0.1394 | 0.1387 | 0.1439 | 0.0755 | 0.0799 | 0.0847 | 0.0444 | 0.0473 | 0.0522 |
| | | MUL | 0.06 | 0.0672 | 0.1363 | 0.1096 | 0.1177 | 0.1258 | 0.1204 | 0.1209 | 0.1281 | 0.1928 | 0.1999 | 0.2035 |
| | | SC | 0 | 0.045 | 0.1 | 0.0693 | 0.068 | 0.0696 | 0.1192 | 0.1207 | 0.1284 | 0.0829 | 0.0872 | 0.0908 |
| | | AwM | 0 | 0 | 0 | 0 | 0 | 0 | 0 | 0 | 0 | 0 | 0 | 0 |
| | | Avg | 0.0056 | 0.0122 | 0.0107 | 0.0002 | 0.0009 | 0 | 0.001 | 0.0005 | 0 | 0 | 0 | 0 |
| | | MP | 0.01 | 0.0178 | 0.0452 | 0 | 0 | 0 | 0 | 0 | 0 | 0 | 0 | 0 |
| | | LM | 0.0144 | 0.0511 | 0.127 | 0.2141 | 0.2315 | 0.2426 | 0.1681 | 0.1714 | 0.1739 | 0.1749 | 0.1749 | 0.1781 |
| | | MRP | 0.0433 | 0.0572 | 0.1219 | 0.0837 | 0.0856 | 0.087 | 0.174 | 0.1728 | 0.1793 | 0.0772 | 0.0802 | 0.0864 |
| | 0.05 | Avg | 0.0156 | 0.0615 | 0.1062 | 0.0384 | 0.0522 | 0.091 | 0.0741 | 0.0859 | 0.13 | 0.1436 | 0.1779 | 0.1855 |
| | | AU | 0.0767 | 0.0833 | 0.1081 | 0.1155 | 0.1239 | 0.1535 | 0.0533 | 0.0893 | 0.1119 | 0.0413 | 0.064 | 0.0881 |
| | | MUL | 0.0134 | 0.0607 | 0.1225 | 0.0558 | 0.0893 | 0.1132 | 0.0701 | 0.0952 | 0.1344 | 0.151 | 0.1849 | 0.1877 |
| | | SC | 0.0886 | 0.0756 | 0.094 | 0.0731 | 0.0803 | 0.0985 | 0.0968 | 0.1123 | 0.154 | 0.0847 | 0.1007 | 0.116 |
| | | AwM | 0 | 0 | 0 | 0 | 0 | 0 | 0 | 0 | 0 | 0 | 0 | 0 |
| | | Avg | 0.0015 | 0.0055 | 0 | 0 | 0 | 0.0025 | 0 | 0 | 0 | 0.0098 | 0.001 | 0 |
| | | MP | 0 | 0 | 0 | 0 | 0 | 0 | 0 | 0 | 0 | 0 | 0 | 0 |
| | | LM | 0.064 | 0.0907 | 0.1623 | 0.1267 | 0.203 | 0.2076 | 0.1031 | 0.119 | 0.1471 | 0.1327 | 0.1492 | 0.1626 |
| | | MRP | 0.0451 | 0.057 | 0.094 | 0.062 | 0.0766 | 0.101 | 0.0942 | 0.1135 | 0.1482 | 0.075 | 0.0888 | 0.1172 |
| | 0.1 | Avg | 0.1232 | 0.1256 | 0.1394 | 0.089 | 0.111 | 0.1678 | 0.0595 | 0.0905 | 0.1806 | 0.1094 | 0.1046 | 0.1254 |
| | | AU | 0.0648 | 0.0751 | 0.1401 | 0.0682 | 0.1268 | 0.1367 | 0.0642 | 0.1019 | 0.152 | 0.0557 | 0.1036 | 0.1314 |
| | | MUL | 0.1036 | 0.116 | 0.1495 | 0.098 | 0.1031 | 0.1494 | 0.0628 | 0.0854 | 0.1588 | 0.0862 | 0.129 | 0.1317 |
| | | SC | 0.0769 | 0.0592 | 0.1166 | 0.0553 | 0.1063 | 0.1186 | 0.0526 | 0.0862 | 0.1296 | 0.0315 | 0.092 | 0.1063 |
| | | AwM | 0 | 0 | 0 | 0 | 0 | 0 | 0 | 0 | 0 | 0 | 0 | 0 |
| | | Avg | 0.0168 | 0.0017 | 0 | 0 | 0 | 0 | 0.0183 | 0.0009 | 0.0066 | 0 | 0.0202 | 0.0004 |
| | | MP | 0 | 0 | 0 | 0 | 0 | 0 | 0 | 0 | 0 | 0 | 0 | 0 |
| | | LM | 0.0638 | 0.1089 | 0.1279 | 0.0613 | 0.0828 | 0.1647 | 0.0318 | 0.0744 | 0.1365 | 0.0477 | 0.0971 | 0.1366 |
| | | MRP | 0.096 | 0.129 | 0.1499 | 0.1038 | 0.1309 | 0.1487 | 0.0611 | 0.1054 | 0.1671 | 0.082 | 0.1283 | 0.13 |
| GSAGen_lAvg | 0.01 | Avg | 0.0667 | 0.1117 | 0.1819 | 0.1919 | 0.2027 | 0.2051 | 0.0812 | 0.0847 | 0.0878 | 0.0929 | 0.0905 | 0.0929 |
| | | AU | 0.0656 | 0.095 | 0.1559 | 0.0508 | 0.0473 | 0.0502 | 0.1274 | 0.1404 | 0.1424 | 0.2805 | 0.2885 | 0.2941 |
| | | MUL | 0.0444 | 0.1017 | 0.1315 | 0.1979 | 0.2046 | 0.2079 | 0.0541 | 0.0557 | 0.0623 | 0.0615 | 0.0591 | 0.0629 |
| | | SC | 0.0078 | 0.05 | 0.1059 | 0.1111 | 0.112 | 0.1158 | 0.1044 | 0.1083 | 0.1186 | 0.27 | 0.2774 | 0.282 |
| | | AwM | 0 | 0 | 0 | 0 | 0 | 0 | 0 | 0 | 0 | 0 | 0 | 0 |
| | | Avg | 0.0122 | 0.0067 | 0 | 0 | 0 | 0 | 0 | 0 | 0.0007 | 0 | 0 | 0 |
| | | MP | 0.0078 | 0.0022 | 0.0189 | 0 | 0 | 0 | 0 | 0 | 0 | 0 | 0 | 0 |
| | | LM | 0.0167 | 0.0656 | 0.13 | 0.2148 | 0.2158 | 0.222 | 0.0821 | 0.0836 | 0.0903 | 0.0929 | 0.0926 | 0.0938 |
| | | MRP | 0.0378 | 0.08 | 0.153 | 0.1541 | 0.1511 | 0.1587 | 0.1432 | 0.1562 | 0.1579 | 0.0935 | 0.0962 | 0.103 |
| | 0.05 | Avg | 0.0301 | 0.0454 | 0.1147 | 0.2642 | 0.2351 | 0.2417 | 0.0452 | 0.0601 | 0.0927 | 0.0987 | 0.1037 | 0.1146 |
| | | AU | 0.0407 | 0.0627 | 0.0875 | 0.0445 | 0.056 | 0.0853 | 0.0918 | 0.1943 | 0.1886 | 0.1581 | 0.1928 | 0.2236 |
| | | MUL | 0.0325 | 0.0319 | 0.1009 | 0.2658 | 0.237 | 0.24 | 0.0351 | 0.0478 | 0.075 | 0.0783 | 0.0918 | 0.1063 |
| | | SC | 0.0367 | 0.0693 | 0.1021 | 0.086 | 0.0943 | 0.1283 | 0.0927 | 0.143 | 0.1775 | 0.1758 | 0.1923 | 0.2092 |
| | | AwM | 0 | 0 | 0 | 0 | 0 | 0 | 0 | 0 | 0 | 0 | 0 | 0 |
| | | Avg | 0.0022 | 0 | 0 | 0 | 0 | 0 | 0 | 0.0044 | 0.0015 | 0.009 | 0.0011 | 0 |
| | | MP | 0 | 0 | 0 | 0 | 0 | 0 | 0 | 0 | 0 | 0 | 0 | 0 |
| | | LM | 0.0402 | 0.0512 | 0.1304 | 0.2485 | 0.2312 | 0.2514 | 0.0468 | 0.0639 | 0.0941 | 0.1152 | 0.132 | 0.1534 |
| | | MRP | 0.0473 | 0.0652 | 0.085 | 0.0803 | 0.1023 | 0.1427 | 0.1014 | 0.2104 | 0.1945 | 0.0954 | 0.1086 | 0.1371 |
| | 0.1 | Avg | 0.0262 | 0.1353 | 0.135 | 0.1062 | 0.0975 | 0.1562 | 0.035 | 0.0951 | 0.1149 | 0.0835 | 0.1072 | 0.1739 |
| | | AU | 0.0339 | 0.1134 | 0.105 | 0.0442 | 0.1023 | 0.1462 | 0.036 | 0.0621 | 0.161 | 0.0684 | 0.0986 | 0.1653 |
| | | MUL | 0.0365 | 0.1257 | 0.1448 | 0.0997 | 0.1034 | 0.1523 | 0.0369 | 0.1034 | 0.1256 | 0.0689 | 0.0864 | 0.1481 |
| | | SC | 0.0336 | 0.0798 | 0.1134 | 0.0234 | 0.0982 | 0.1446 | 0.0632 | 0.0404 | 0.1594 | 0.0414 | 0.0778 | 0.134 |
| | | AwM | 0 | 0 | 0 | 0 | 0 | 0 | 0 | 0 | 0 | 0 | 0 | 0 |
| | | Avg | 0.0002 | 0 | 0 | 0.0183 | 0.001 | 0.0066 | 0.0151 | 0.0025 | 0 | 0 | 0 | 0 |
| | | MP | 0 | 0 | 0 | 0 | 0 | 0 | 0 | 0 | 0 | 0 | 0 | 0 |
| | | LM | 0.0415 | 0.1089 | 0.1233 | 0.074 | 0.0533 | 0.1227 | 0.0438 | 0.0881 | 0.1052 | 0.0395 | 0.0879 | 0.1391 |
| | | MRP | 0.035 | 0.1404 | 0.1085 | 0.0743 | 0.0914 | 0.1234 | 0.0401 | 0.0891 | 0.1601 | 0.0497 | 0.1233 | 0.1857 |

**Table 19. AvgHR results of multi-criteria recommender systems on YM10 dataset for each aggregation techniques and filler size 5% (S2).**

| Attack Type | Cluster Size | Aggregation Techniques | S2 MUP-Usr | | | MUP-DR | | | MUP-NNZ | | | MUP-Sum | | |
|---|---|---|---|---|---|---|---|---|---|---|---|---|---|---|
| | | | Top-5 | Top-10 | Top-15 | Top-5 | Top-10 | Top-15 | Top-5 | Top-10 | Top-15 | Top-5 | Top-10 | Top-15 |
| GSAGen$_l$Ran | 0.01 | Avg | 0.0778 | 0.1639 | 0.1528 | 0.1921 | 0.2033 | 0.2046 | 0.1639 | 0.1639 | 0.1639 | 0.0441 | 0.0531 | 0.0556 |
| | | AU | 0.075 | 0.0583 | 0.1213 | 0.228 | 0.2248 | 0.2405 | 0.118 | 0.1275 | 0.1248 | 0.0058 | 0.0138 | 0.0146 |
| | | MUL | 0.0639 | 0.1653 | 0.1093 | 0.1921 | 0.2074 | 0.2055 | 0.1716 | 0.1716 | 0.1716 | 0.0597 | 0.0679 | 0.0712 |
| | | SC | 0.0333 | 0.0792 | 0.1019 | 0.1781 | 0.1719 | 0.1819 | 0.1421 | 0.1322 | 0.1292 | 0.1191 | 0.1257 | 0.1315 |
| | | AwM | 0.0139 | 0 | 0 | 0 | 0 | 0 | 0 | 0 | 0 | 0 | 0 | 0 |
| | | Avg | 0.0139 | 0.0556 | 0.0139 | 0 | 0 | 0 | 0 | 0 | 0 | 0 | 0 | 0 |
| | | MP | 0.0167 | 0.0972 | 0.1241 | 0.1228 | 0.1277 | 0.1407 | 0.1876 | 0.1876 | 0.192 | 0.0565 | 0.0638 | 0.0729 |
| | | LM | 0.0306 | 0.0819 | 0.0972 | 0.1231 | 0.1283 | 0.1321 | 0.1996 | 0.1996 | 0.1996 | 0.1326 | 0.1474 | 0.1556 |
| | | MRP | 0.0444 | 0.1042 | 0.1417 | 0.2902 | 0.2925 | 0.3068 | 0.1537 | 0.1537 | 0.1591 | 0.0051 | 0.0103 | 0.0111 |
| | 0.05 | Avg | 0.0714 | 0.0714 | 0.0934 | 0.141 | 0.2108 | 0.2033 | 0.0947 | 0.0914 | 0.1006 | 0.027 | 0.0745 | 0.0885 |
| | | AU | 0.0522 | 0.1137 | 0.1222 | 0.1432 | 0.1969 | 0.2168 | 0.0692 | 0.09 | 0.1128 | 0.0003 | 0.0503 | 0.06 |
| | | MUL | 0.0159 | 0.0846 | 0.1214 | 0.1516 | 0.2102 | 0.1822 | 0.0923 | 0.107 | 0.131 | 0.0332 | 0.0807 | 0.1007 |
| | | SC | 0.0137 | 0.0945 | 0.1203 | 0.1202 | 0.1568 | 0.1834 | 0.0933 | 0.1032 | 0.1272 | 0.0695 | 0.096 | 0.1237 |
| | | AwM | 0 | 0 | 0 | 0 | 0 | 0 | 0 | 0 | 0 | 0 | 0 | 0 |
| | | Avg | 0.0385 | 0 | 0.0075 | 0 | 0 | 0 | 0 | 0 | 0 | 0 | 0 | 0 |
| | | MP | 0.106 | 0.1107 | 0.1315 | 0.0754 | 0.1262 | 0.1474 | 0.1163 | 0.1341 | 0.1384 | 0.0485 | 0.0824 | 0.1213 |
| | | LM | 0.0511 | 0.0396 | 0.0945 | 0.0656 | 0.0992 | 0.0826 | 0.1239 | 0.1102 | 0.1329 | 0.0867 | 0.1543 | 0.168 |
| | | MRP | 0.078 | 0.0827 | 0.1053 | 0.1998 | 0.2765 | 0.2881 | 0.0787 | 0.096 | 0.1076 | 0 | 0.0447 | 0.0866 |
| | 0.1 | Avg | 0.103 | 0.0985 | 0.163 | 0.1311 | 0.1216 | 0.1497 | 0 | 0.0492 | 0.0831 | 0 | 0.0679 | 0.112 |
| | | AU | 0.0306 | 0.0732 | 0.1474 | 0.0456 | 0.0321 | 0.1233 | 0.0361 | 0.0895 | 0.0807 | 0.05 | 0.1276 | 0.1127 |
| | | MUL | 0.0945 | 0.1287 | 0.1699 | 0.126 | 0.1214 | 0.1688 | 0.0068 | 0.0378 | 0.1077 | 0 | 0.0656 | 0.1107 |
| | | SC | 0.0107 | 0.0728 | 0.1406 | 0.0579 | 0.0365 | 0.1007 | 0.0822 | 0.1146 | 0.1394 | 0.0475 | 0.1272 | 0.1022 |
| | | AwM | 0 | 0 | 0 | 0 | 0 | 0 | 0 | 0 | 0 | 0 | 0 | 0 |
| | | Avg | 0.0014 | 0.0007 | 0 | 0 | 0 | 0 | 0 | 0 | 0 | 0.0475 | 0.0012 | 0 |
| | | MP | 0.0555 | 0.0518 | 0.1446 | 0.0208 | 0.1344 | 0.1474 | 0 | 0.0791 | 0.1072 | 0.0475 | 0.1223 | 0.1021 |
| | | LM | 0.0664 | 0.0964 | 0.1427 | 0.0208 | 0.138 | 0.1346 | 0 | 0.101 | 0.1148 | 0 | 0.0825 | 0.1283 |
| | | MRP | 0.0869 | 0.0904 | 0.1462 | 0.0735 | 0.0577 | 0.1474 | 0.0068 | 0.0141 | 0.0515 | 0.0519 | 0.093 | 0.099 |
| GSAGen$_l$Avg | 0.01 | Avg | 0.0556 | 0.0903 | 0.1796 | 0.2413 | 0.2463 | 0.2495 | 0.2328 | 0.223 | 0.2212 | 0.035 | 0.035 | 0.035 |
| | | AU | 0.1139 | 0.1694 | 0.1972 | 0.0757 | 0.0765 | 0.0775 | 0.0787 | 0.0787 | 0.0819 | 0.1118 | 0.1172 | 0.1221 |
| | | MUL | 0.0194 | 0.025 | 0.0944 | 0.2413 | 0.2463 | 0.25 | 0.2339 | 0.224 | 0.2223 | 0.0374 | 0.035 | 0.0415 |
| | | SC | 0.0056 | 0.0875 | 0.1269 | 0.095 | 0.0991 | 0.0995 | 0.1412 | 0.1352 | 0.1362 | 0.1445 | 0.1488 | 0.1538 |
| | | AwM | 0 | 0 | 0 | 0 | 0 | 0 | 0 | 0 | 0 | 0 | 0 | 0 |
| | | Avg | 0.0111 | 0.0694 | 0.0231 | 0.0036 | 0.0007 | 0 | 0 | 0 | 0 | 0 | 0 | 0 |
| | | MP | 0.0472 | 0.0903 | 0.138 | 0.1566 | 0.1555 | 0.1621 | 0.2335 | 0.2237 | 0.2225 | 0.1752 | 0.1851 | 0.1826 |
| | | LM | 0.0167 | 0.0208 | 0.1185 | 0.2629 | 0.2689 | 0.2777 | 0.2153 | 0.2055 | 0.2053 | 0.1508 | 0.157 | 0.1656 |
| | | MRP | 0.0472 | 0.1292 | 0.1583 | 0.1301 | 0.1347 | 0.1313 | 0.1566 | 0.1566 | 0.1637 | 0.0199 | 0.0204 | 0.0211 |
| | 0.05 | Avg | 0.0643 | 0.1143 | 0.0976 | 0.1878 | 0.2006 | 0.1938 | 0.1212 | 0.1278 | 0.1404 | 0.0321 | 0.0603 | 0.0663 |
| | | AU | 0.1346 | 0.1805 | 0.1679 | 0.0721 | 0.0796 | 0.0932 | 0.0473 | 0.0471 | 0.1056 | 0.0882 | 0.1117 | 0.1192 |
| | | MUL | 0.078 | 0.1404 | 0.0969 | 0.1878 | 0.1989 | 0.1974 | 0.1223 | 0.1289 | 0.1459 | 0.0321 | 0.0603 | 0.0737 |
| | | SC | 0.1088 | 0.1511 | 0.1427 | 0.0762 | 0.0941 | 0.12 | 0.093 | 0.0948 | 0.1398 | 0.098 | 0.1354 | 0.127 |
| | | AwM | 0 | 0 | 0 | 0 | 0 | 0 | 0 | 0 | 0 | 0 | 0 | 0 |
| | | Avg | 0.0527 | 0.0011 | 0 | 0 | 0 | 0 | 0 | 0 | 0 | 0 | 0 | 0 |
| | | MP | 0.0929 | 0.1459 | 0.1192 | 0.0823 | 0.1072 | 0.1349 | 0.1286 | 0.1179 | 0.1567 | 0.1161 | 0.1447 | 0.1321 |
| | | LM | 0.0429 | 0.1376 | 0.0727 | 0.1833 | 0.213 | 0.2326 | 0.1325 | 0.1422 | 0.1542 | 0.1024 | 0.1469 | 0.1816 |
| | | MRP | 0.1489 | 0.1953 | 0.1379 | 0.095 | 0.1196 | 0.1074 | 0.094 | 0.1083 | 0.1661 | 0.0167 | 0.0346 | 0.0476 |
| | 0.1 | Avg | 0.0145 | 0.0678 | 0.0972 | 0.0016 | 0.1078 | 0.1954 | 0.0377 | 0.0413 | 0.0546 | 0.0593 | 0.0739 | 0.1588 |
| | | AU | 0.2005 | 0.1008 | 0.1133 | 0.1497 | 0.1344 | 0.152 | 0.0128 | 0.0631 | 0.0942 | 0.0085 | 0.0667 | 0.127 |
| | | MUL | 0.0038 | 0.068 | 0.1008 | 0.0426 | 0.0936 | 0.157 | 0.0377 | 0.044 | 0.0551 | 0.0623 | 0.0997 | 0.186 |
| | | SC | 0.124 | 0.1101 | 0.1435 | 0.0833 | 0.1504 | 0.1232 | 0.0128 | 0.1023 | 0.1214 | 0.0085 | 0.0731 | 0.1115 |
| | | AwM | 0 | 0 | 0 | 0 | 0 | 0 | 0 | 0 | 0 | 0 | 0 | 0 |
| | | Avg | 0.0475 | 0.0022 | 0 | 0.0415 | 0.0042 | 0 | 0 | 0 | 0 | 0 | 0 | 0 |
| | | MP | 0.0978 | 0.0702 | 0.1273 | 0.0443 | 0.0889 | 0.1329 | 0.0984 | 0.0843 | 0.1612 | 0.0566 | 0.0977 | 0.15 |
| | | LM | 0.003 | 0.0745 | 0.1535 | 0.0533 | 0.0683 | 0.1658 | 0.0694 | 0.1056 | 0.1423 | 0.0699 | 0.0877 | 0.1683 |
| | | MRP | 0.182 | 0.0769 | 0.1091 | 0.1689 | 0.1859 | 0.1554 | 0 | 0.0231 | 0.0275 | 0.0814 | 0.071 | 0.1418 |

In the S1 scenario for YM10 dataset, the average performance of $GSAGen_lAvg$ was slightly higher than that of $GSAGen_lRan$ across the Top-5, Top-10, and Top-15 metrics (e.g., 0.0615 vs. 0.0580 for $S1_{Top5}$); however, these differences were not statistically significant, as all p-values were above 0.05 (Fig 5). Therefore, both methods yielded comparable outcomes in the S1 scenario, and the superiority of $GSAGen_lAvg$ could not be statistically confirmed. In contrast, more distinct differences were observed in the S2 scenario. For the $S2_{Top5}$ metric, the $GSAGen_lRan$ method achieved a significantly higher mean value compared to $GSAGen_lAvg$ (p = 0.0129, t = –2.961, Cohen's dz = –0.893). Similarly, this difference was also statistically significant for the $S2_{Top15}$ metric (p = 0.0160, t = –2.841, Cohen's dz = –0.857). In both cases, the Cohen's dz values indicated a large effect size, revealing a strong superiority of $GSAGen_lRan$ over $GSAGen_lAvg$. For the $S2_{Top10}$ metric, the difference was marginally significant (p = 0.0523), suggesting a trend favoring $GSAGen_lRan$, though without strong statistical support. Overall, while no significant performance difference was detected between $GSAGen_lAvg$ and $GSAGen_lRan$ in the S1 scenario, $GSAGen_lRan$ demonstrated a statistically significant and large effect size advantage in the S2 scenario, particularly in the Top-5 and Top-15 metrics. These results suggest that the performance of the $GSAGen_lAvg$ approach may vary depending on the data structure and scenario characteristics, indicating that in certain contexts, even random selection-based strategies can yield relatively higher performance.

When the filler size ratio was increased to 5%, the YM10 results exhibited a partially different pattern compared to the previous 1% filler ratio experiments (Fig 5). In the S1 scenario, the differences between $GSAGen_lAvg$ and $GSAGen_lRan$ remained small and statistically insignificant (e.g., p = 0.610 for $S1_{Top5}$ and p = 0.272 for $S1_{Top10}$). However, a statistically significant difference was observed in the $S1_{Top15}$ metric ($p = 2.27e{-}5$, t = 6.997, Cohen's dz = 2.109). The remarkably high t-statistic and effect size values indicate a clear superiority of $GSAGen_lAvg$ over the random baseline. This finding implies that at larger recommendation list sizes (Top-15), the performance advantage of $GSAGen_lAvg$ becomes more pronounced. In the S2 scenario, however, this tendency weakened considerably. For all Top-5, Top-10, and Top-15 metrics, p-values remained above 0.05, indicating no statistically significant difference between the two methods. Moreover, the Cohen's dz values ranged between –0.06 and 0.35, corresponding to small effect sizes. This outcome

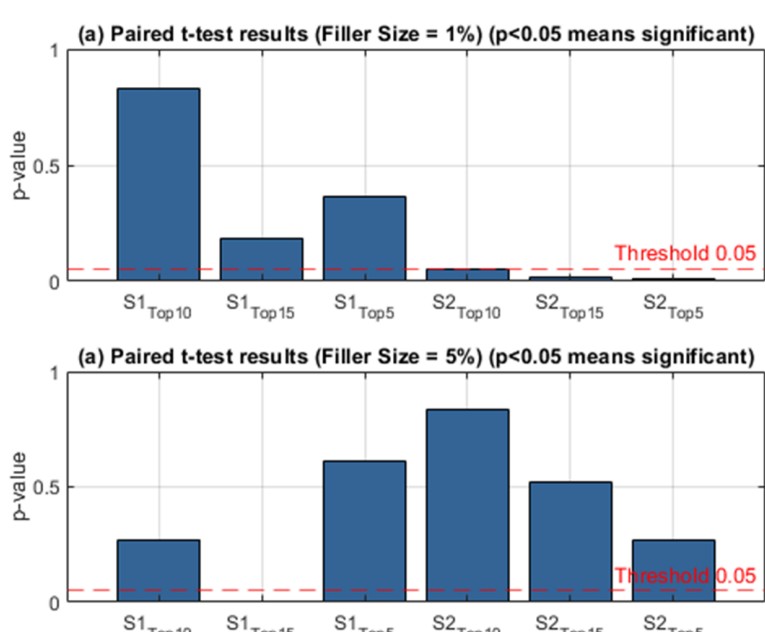

**Fig 5**. Statistical analysis for YM10 dataset according to AvgHR.

suggests that increasing the filler ratio in the S2 scenario diminishes the statistical advantage of *GSAGen$_l$Avg*, implying that the method's performance is sensitive to both data volume and sampling density. In summary, while the *GSAGen$_l$Ran* method exhibited significant superiority under the 1% filler condition for specific scenarios, increasing the filler ratio to 5% led to statistically significant improvements in favor of *GSAGen$_l$Avg*, particularly within the S1 scenario. These findings indicate that the recommendation performance depends not only on the employed method but also on factors such as sampling density and data distribution homogeneity. The increase in filler ratio appears to enhance the stability of the *GSAGen$_l$Avg* approach while reducing the comparative advantage of the *GSAGen$_l$Ran* method.

For the other dataset, YM20, the corresponding results are presented as Fig 6. When the filler size was set to 1%, the results indicated that although *GSAGen$_l$Avg* produced slightly higher average values overall, most differences were not statistically significant (e.g., $p > 0.1$). However, significant differences were observed for $S_1$–Top15 ($p = 2.27 \times 10^{-5}$) and for $S_2$ metrics, particularly Top-5 and Top-15 ($p < 0.05$). This suggests that at lower injection levels, the average-based attack tends to adapt more effectively to user similarity patterns, thereby producing a stronger impact. Furthermore, the high Cohen's $d_z$ values (ranging between 1.15 and 1.76) confirm a strong effect size in these cases.

When the filler size increased to 5%, the structure of attack impact changed (see Fig 6). Although *GSAGen$_l$Avg* continued to yield higher mean results, the statistical significance of the differences decreased (e.g., $p > 0.2$). Only in the $S2$ scenario (particularly for Top-5, Top-10, and Top-15; $p < 0.05$) were significant effects detected, indicating that the system became more resistant to perturbations as the filler ratio increased. This behavior implies that at higher filler levels, the influence of injected profiles diminishes due to the system's ability to absorb and stabilize rating fluctuations.

Briefly, the YM20 dataset demonstrated relatively high robustness against both *GSAGen$_l$Avg* and *GSAGen$_l$Ran* attacks. While the $S1$ scenario (attacks targeting popular items) exhibited higher vulnerability at lower filler levels, the $S2$ scenario (attacks focusing on less popular or sparsely rated items) continued to show statistically significant effects even under higher filler sizes. These findings highlight that dataset scale and diversity are critical factors in determining system resilience, as they shape how recommendation models respond to adversarial manipulations under different attack conditions.

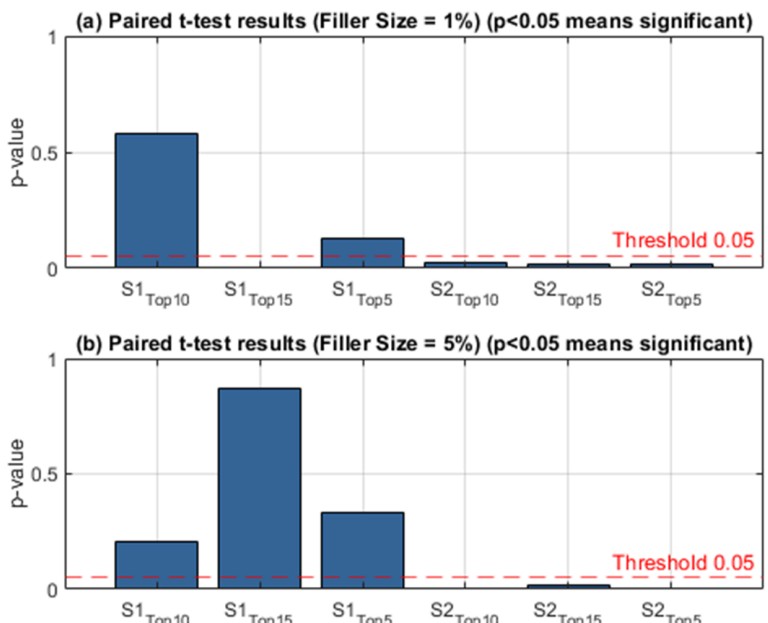

**Fig 6**. Statistical analysis for YM20 dataset according to AvgHR.

Overall, the results highlight that the combination of larger cluster sizes, the *GSAGen₁Avg* attack, and advanced target item selection strategies such as MUP-NNZ and MUP-Sum significantly increases the AvgHR values, making them critical factors in the success of adversarial attacks. The higher vulnerability observed under the MUP-NNZ and MUP-Sum selection strategies can be explained by their focus on items with limited rating information. In both cases, the targeted items are characterized by low popularity and sparse user interactions, which makes their estimated rating profiles less stable and more susceptible to distortion. Consequently, even a small number of injected fake ratings can cause substantial shifts in the predicted scores of these items. This heightened sensitivity amplifies the influence of adversarial users, leading to higher AvgHR values and demonstrating that sparsely rated items are inherently more prone to manipulation. Conversely, items with richer rating histories tend to exhibit greater robustness, as the diversity and volume of genuine feedback dilute the relative effect of artificially introduced profiles. Collectively, these findings underscore the importance of developing defense mechanisms that are specifically tailored to the system's structural vulnerabilities and data sparsity characteristics.

A comparison of the YM10 and YM20 datasets in terms of the AvgHR metric reveals notable differences in their susceptibility to group shilling attacks. While both datasets exhibit the general trend that higher filler sizes amplify the impact of attacks, the overall intensity and sensitivity to target item selection strategies vary across datasets. For the YM10 dataset, AvgHR values are generally higher. Even at a filler size of 1%, the proportion of fake items in the Top-10 lists ranges between 6% and 12%, and at 5% filler this ratio increases further to 9–15%. This indicates that, due to the structural characteristics of YM10, fake profiles can more easily penetrate the recommendation lists. In particular, the MUP-NNZ strategy consistently produces the highest values, with nearly 13–14% of Top-10 recommendations being manipulated items. MUP-DR and MUP-Sum also yield strong attack success rates, while MUP-UsR remains comparatively more resilient. In contrast, the YM20 dataset exhibits lower AvgHR values, suggesting stronger robustness against attacks. At 1% filler, the proportion of fake items in Top-10 lists typically falls within the 3–6% range, while at 5% filler this ratio increases to 8–12%. Although YM20 is less vulnerable at low filler levels, the effect of attacks becomes more pronounced as the filler size increases. Specifically, both MUP-Sum and MUP-DR exceed the 10% threshold at 5% filler, demonstrating that these strategies can still achieve substantial influence under more aggressive attack scenarios. In summary, the AvgHR results highlight that YM10 is more fragile, with fake items infiltrating recommendation lists even at low filler levels, whereas YM20 shows greater resistance but remains vulnerable as the filler ratio grows. This comparison underscores the critical role of dataset characteristics in shaping the effectiveness of group shilling attacks, as well as the importance of tailoring defense mechanisms to the specific vulnerabilities of each dataset.

## 7 Conclusion and future work

This study presents a comprehensive analysis of the vulnerabilities of multi-criteria recommender systems to group shilling attacks. The existing literature predominantly focuses on such attacks in single-criteria systems, leaving multi-criteria systems largely unexamined. In this context, our study addresses a critical gap by investigating, modeling, and analyzing shilling attacks specifically in multi-criteria group recommender systems. The proposed attack strategy reveals the susceptibility of multi-criteria recommender systems to contemporary threats. Our findings demonstrate that while these systems offer significant advantages in terms of personalization and detailed evaluations, they can be easily manipulated by malicious user groups. Such manipulations not only decrease the recommendation accuracy but also negatively impact user satisfaction and trust, posing a serious threat to the long-term success and sustainability of recommender systems. The empirical results further show that the impact of attacks increases as the filler size grows. For instance, when the filler size rises from 1% to 5%, the AvgHR values significantly increase, reaching up to 20–25% under average-based attacks. In particular, the MRP aggregation strategy consistently emerged as the most vulnerable, while AU, MUL, and SC were also notably affected. On the other hand, MP and LM demonstrated comparatively greater robustness. These findings emphasize that attack severity is highly dependent on both the attack model and the aggregation

technique employed. Also, it is important to note that the robustness observed in this study is specific to the employed datasets and attack models; therefore, the findings should be interpreted within the context of these experimental conditions.

In conclusion, the complexity and user-centric nature of multi-criteria recommender systems bring both advantages and vulnerabilities. The present study not only addresses these vulnerabilities but also provides a foundation for future research, aiming to contribute to the development of secure, efficient, and user-friendly recommender systems. Future studies could focus on developing more sophisticated detection algorithms that leverage machine learning and anomaly detection techniques to identify group shilling attacks more effectively. Additionally, investigating how dynamic user behavior and changing preferences influence the susceptibility of multi-criteria systems to group shilling attacks could enhance the adaptability and resilience of these systems. Moreover, future work could extend this research to different application domains and datasets to examine whether the observed patterns of robustness and vulnerability generalize across various contexts. Such cross-domain analyses would provide deeper insights into the transferability of attack dynamics and the scalability of proposed defense mechanisms.

## Author contributions

**Conceptualization:** Tugba Turkoglu Kaya.

**Formal analysis:** Tugba Turkoglu Kaya.

**Investigation:** Tugba Turkoglu Kaya.

**Methodology:** Tugba Turkoglu Kaya.

**Writing – original draft:** Tugba Turkoglu Kaya.

**Writing – review & editing:** Tugba Turkoglu Kaya.

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
