## [Decision Letter · Decision Letter 0]

23 Oct 2025

PONE-D-25-46783Multi-Criteria Group Shilling AttacksPLOS ONE

Dear Dr. Turkoglu Kaya,

Thank you for submitting your manuscript to PLOS ONE. After careful consideration, we feel that it has merit but does not fully meet PLOS ONE’s publication criteria as it currently stands. Therefore, we invite you to submit a revised version of the manuscript that addresses the points raised during the review process.

Please find the reviewer comments below.

We look forward to receiving your revised manuscript.

Kind regards,

Joanna Tindall, PhD

Staff Editor

PLOS ONE

Reviewers' comments:

Reviewer's Responses to Questions

**Comments to the Author**

1. Is the manuscript technically sound, and do the data support the conclusions?

Reviewer #1: Yes

Reviewer #2: Yes

2. Has the statistical analysis been performed appropriately and rigorously?

Reviewer #1: Yes

Reviewer #2: Yes

3. Have the authors made all data underlying the findings in their manuscript fully available?

Reviewer #1: Yes

Reviewer #2: No

4. Is the manuscript presented in an intelligible fashion and written in standard English?

Reviewer #1: Yes

Reviewer #2: Yes

5. Review Comments to the Author

Reviewer #1: This paper presents a valuable and timely study on the robustness of multi-criteria group recommender systems under coordinated adversarial manipulation, known as group shilling attacks. It introduces a novel adaptation of the GSAGen model to multi-criteria settings (MC-GSA) and provides an extensive experimental evaluation on Yahoo! Movies datasets (YM10 and YM20). The topic is original and relevant for the recommender systems community, addressing a clear research gap. The manuscript is well structured, and the methodology is sound. However, several sections could benefit from clarification, deeper discussion, and minor language and presentation improvements before publication.

• The abstract is informative but should briefly state the main quantitative findings (e.g., percentage impact or robustness level) to better convey the results.

• Clarify the novelty claim: while this is presented as the “first comprehensive robustness analysis,” include a brief comparative statement positioning this work against recent studies (2022–2024) in multi-criteria RS or attack detection.

• Some sentences in the introduction are repetitive (lines 5–25); they could be condensed to improve flow.

• The literature review is comprehensive but should reference a few recent studies (post-2020) on adversarial or poisoning attacks in recommender systems beyond shilling (e.g., GNN-based adversarial robustness).

• Clarify the distinction between group shilling and coordinated fake review attacks, as these are conceptually close but differ in structure.

• The explanation of the proposed MC-GSA model is clear, yet the flow could be improved with a diagram legend for Figure 2.

• Table 1 mixes notation and textual explanation; please format consistently (e.g., italicize symbols, align descriptions).

• Algorithm 1 is detailed but lacks complexity analysis or runtime comment—add a brief note on computational cost or scalability.

• Tables 3–6 contain rich data but are difficult to read. Summarize key results visually (e.g., line charts or comparative plots for AvgHR and WRV metrics).

• Add a short paragraph interpreting why certain selection strategies (e.g., MUP-NNZ) cause higher vulnerability—this would enhance insight beyond numerical reporting.

• Include a statistical significance statement or standard deviation where applicable to support the reliability of findings.

• Minor grammatical corrections are needed throughout (e.g., “the another study” → “another study”; “criteria-based is intuitively” → “criteria-based scenario is intuitively”).

• Check reference formatting for consistency (e.g., bracketed style [1], [2], [3]) and ensure recent works are included.

• Figures and tables should include full captions that explain abbreviations (e.g., MUP-DR, AvgHR).

The manuscript makes a significant contribution to the understanding of group shilling vulnerabilities in multi-criteria recommender systems. It requires only minor revisions—primarily editorial polishing, slight restructuring for clarity, and enhancement of the discussion and related work. Once these are addressed, the paper will be suitable for publication in PLOS ONE.

Reviewer #2: The manuscript presents a study on the robustness of multi-criteria group recommender systems against shilling attacks. The research addresses a relevant and timely issue, given the increasing use of group recommendation systems and the potential for malicious manipulation. The study proposes a novel group shilling attack strategy adapted for multi-criteria settings and evaluates its impact. While the study has merit, some aspects of the presentation and interpretation of the results need improvement to ensure accuracy and clarity.

1. Figures and tables should be referenced explicitly in the text, and the key findings from each should be highlighted. It is currently difficult to understand the exact impact of the proposed attack and the comparative performance of the algorithms without carefully scrutinizing the figures.

2. The conclusion that multi-criteria systems are "quite robust" needs to be tempered. While the results might indicate a degree of robustness compared to certain benchmarks, it's crucial to avoid overgeneralization. The conclusion should be rewritten to reflect the specific findings of this study, acknowledging the limitations of the attack scenarios and datasets used.

3. Avoid strong claims about the generalizability of the findings. The study is based on specific datasets and attack models. The conclusions should emphasize that the robustness observed is within the context of these specific experimental conditions.

4. When reviewing your “Related Work” section, I observed that it currently includes only classical shilling attacks (random/average bots), group shilling, and a few referenced studies. While this section provides a good summary of the topic, organizing the literature in a more structured manner or supporting it with a comparative table would make the work stronger and more coherent.

5. Since the dataset used in this study is a movie dataset, it would be valuable to note in the Conclusion section that similar analyses could be extended to other domains and datasets in future research to validate the generalizability of the findings.

The study has potential, but the clarity and accuracy of the results reporting need improvement. The authors should address the comments above before the manuscript is considered for publication.

6. PLOS authors have the option to publish the peer review history of their article (what does this mean?). If published, this will include your full peer review and any attached files.

Reviewer #1: No

Reviewer #2: No

---

## [Author Response · Author response to Decision Letter 1]

29 Oct 2025

Dear Editor and Reviewers,

We would like to thank you for your valuable comments and constructive feedback on our manuscript.

We have carefully revised the paper according to all reviewer and editor suggestions. All changes are highlighted in the revised manuscript (uploaded as “Revised Manuscript with Track Changes”), and detailed responses to each comment are provided in the file titled “Response to Reviewers.”

We believe that the revisions have significantly improved the quality and clarity of the manuscript.

Sincerely,

---

## [Decision Letter · Decision Letter 1]

21 Nov 2025

Multi-Criteria Group Shilling Attacks

PONE-D-25-46783R1

Dear Dr. Turkoglu Kaya,

We’re pleased to inform you that your manuscript has been judged scientifically suitable for publication and will be formally accepted for publication once it meets all outstanding technical requirements.

Kind regards,

Qinglin Meng, Ph.D.

Academic Editor

PLOS ONE

Additional Editor Comments (optional):

The revised version is acceptable for publication.

Reviewers' comments:

Reviewer's Responses to Questions

**Comments to the Author**

1. If the authors have adequately addressed your comments raised in a previous round of review and you feel that this manuscript is now acceptable for publication, you may indicate that here to bypass the “Comments to the Author” section, enter your conflict of interest statement in the “Confidential to Editor” section, and submit your "Accept" recommendation.

Reviewer #1: All comments have been addressed

Reviewer #2: All comments have been addressed

2. Is the manuscript technically sound, and do the data support the conclusions?

Reviewer #1: Yes

Reviewer #2: Yes

3. Has the statistical analysis been performed appropriately and rigorously?

Reviewer #1: Yes

Reviewer #2: Yes

4. Have the authors made all data underlying the findings in their manuscript fully available?

Reviewer #1: Yes

Reviewer #2: No

5. Is the manuscript presented in an intelligible fashion and written in standard English?

Reviewer #1: Yes

Reviewer #2: Yes

6. Review Comments to the Author

Reviewer #1: The authors have properly addressed all of my concerns in the revised manuscript; therefore, I recommend acceptance of the paper for publication.

Reviewer #2: I have carefully reviewed the revised version of the manuscript. The authors have thoroughly addressed all the comments raised in the previous review. The figures and tables are now clearly referenced and discussed in the text, the conclusions have been rewritten to avoid overgeneralization, and the related work section has been substantially improved with relevant literature and comparative organization. Additionally, the clarification regarding dataset dependency and generalizability has been well integrated into the conclusion.

7. PLOS authors have the option to publish the peer review history of their article (what does this mean?). If published, this will include your full peer review and any attached files.

Reviewer #1: No

Reviewer #2: No

---

## [Editor Report · Acceptance letter]

PONE-D-25-46783R1

PLOS ONE

Dear Dr. Turkoglu Kaya,

I'm pleased to inform you that your manuscript has been deemed suitable for publication in PLOS ONE. Congratulations! Your manuscript is now being handed over to our production team.

Kind regards,

on behalf of

Prof. Qinglin Meng

Academic Editor

PLOS ONE